# DNAJC9 prevents CENP-A mislocalization and chromosomal instability by maintaining the fidelity of histone supply chains

Vinutha Balachandra[1], Roshan L Shrestha[1], Colin M Hammond [2,3,4]✉, Shinjen Lin[5], Ivo A Hendriks [2], Subhash Chandra Sethi[1], Lu Chen[5], Samantha Sevilla [6,7], Natasha J Caplen[8], Raj Chari [9], Tatiana S Karpova [10], Katherine McKinnon[11], Matthew AM Todd [2,3,12], Vishal Koparde [6,7], Ken Chih-Chien Cheng[5], Michael L Nielsen[2], Anja Groth[2,3,12] & Munira A Basrai [1]✉

## Abstract

**The centromeric histone H3 variant CENP-A is overexpressed in many cancers. The mislocalization of CENP-A to noncentromeric regions contributes to chromosomal instability (CIN), a hallmark of cancer. However, pathways that promote or prevent CENP-A mislocalization remain poorly defined. Here, we performed a genome-wide RNAi screen for regulators of CENP-A localization which identified DNAJC9, a J-domain protein implicated in histone H3–H4 protein folding, as a factor restricting CENP-A mislocalization. Cells lacking DNAJC9 exhibit mislocalization of CENP-A throughout the genome, and CIN phenotypes. Global interactome analysis showed that DNAJC9 depletion promotes the interaction of CENP-A with the DNA-replication-associated histone chaperone MCM2. CENP-A mislocalization upon DNAJC9 depletion was dependent on MCM2, defining MCM2 as a driver of CENP-A deposition at ectopic sites when H3–H4 supply chains are disrupted. Cells depleted for histone H3.3, also exhibit CENP-A mislocalization. In summary, we have defined novel factors that prevent mislocalization of CENP-A, and demonstrated that the integrity of H3–H4 supply chains regulated by histone chaperones such as DNAJC9 restrict CENP-A mislocalization and CIN.**

**Keywords** CENP-A; DNAJC9; Chromosomal instability; Histone chaperone
**Subject Categories** Cell Cycle; Chromatin, Transcription & Genomics

## Introduction

The centromere is a specialized genomic region that directs the stable transmission of chromosomes during cell division. Defects in the integrity of centromeric chromatin or kinetochore structure can cause chromosomal instability (CIN) through erroneous chromosome segregation and thereby drive aneuploidy, a recurrent feature of cancers and developmental disorders (Barra and Fachinetti, 2018; Gemble et al, 2019; Hassold and Hunt, 2001; Holland and Cleveland, 2009; Thompson and Compton, 2008; Zhang et al, 2016). The histone H3 variant CENP-A (Cse4 in budding yeast, Cnp1 in fission yeast, CID in fruit fly) defines centromeric chromatin and is one of the key components for establishing kinetochore assembly and kinetochore-microtubule attachments (Cheeseman and Desai, 2008; Westhorpe and Straight, 2013).

Restricting the localization of CENP-A to centromeric chromatin is essential for chromosomal stability. The recruitment of CENP-A to the centromeres is uncoupled from canonical histone H3 loading to the chromatin and involves multiple regulatory mechanisms. For example, the dedicated chaperone HJURP deposits CENP-A at centromeres in late telophase/early G1 through interaction with the Mis18 complex which is dynamically regulated by PLK1 and CDK1/2 (Dunleavy et al, 2009; Foltz et al, 2009; Fujita et al, 2007; Jansen et al, 2007; McKinley and Cheeseman, 2014; Silva et al, 2012). During DNA replication, the recycling of pre-existing CENP-A and its reloading at the centromere by HJURP occurs in concert with the helicase MCM2 (Zasadzinska et al, 2018). Regulation at the transcriptional level precisely controls the expression of CENP-A to ensure maximal levels at G2 (Shelby et al, 1997) and recent studies have implicated specific post-translational modifications (PTMs) of CENP-A such as phosphorylation of Ser68

[1]Yeast Genome Stability Section, Genetics Branch, Center for Cancer Research, National Cancer Institute, National Institutes of Health, Bethesda, MD, USA. [2]Novo Nordisk Foundation Center for Protein Research (CPR), Faculty of Health and Medical Sciences, University of Copenhagen, Copenhagen, Denmark. [3]Biotech Research and Innovation Centre (BRIC), Faculty of Health and Medical Sciences, University of Copenhagen, Copenhagen, Denmark. [4]Department of Molecular and Clinical Cancer Medicine, Institute of Systems, Molecular and Integrative Biology, University of Liverpool, Liverpool, UK. [5]Functional Genomics Laboratory, National Center for Advancing Translational Sciences, National Institutes of Health, Bethesda, MD, USA. [6]Collaborative Bioinformatics Resource, Center for Cancer Research, National Cancer Institute, National Institutes of Health, Bethesda, MD, USA. [7]Advanced Biomedical Computational Science, Frederick National Laboratory for Cancer Research, Frederick, MD, USA. [8]Functional Genetics Section, Genetics Branch, Center for Cancer Research, National Cancer Institute, National Institutes of Health, Bethesda, MD, USA. [9]Genome Modification Core (GMC), Frederick National Lab for Cancer Research, Frederick, MD, USA. [10]Optical Microscopy Core, Laboratory of Receptor Biology and Gene Expression, Center for Cancer Research, National Cancer Institute, National Institutes of Health, Bethesda, MD, USA. [11]Flow Cytometry Core, Vaccine Branch, Center for Cancer Research, National Cancer Institute, National Institutes of Health, Bethesda, MD, USA. [12]Department of Cellular and Molecular Medicine, Faculty of Health and Medical Sciences, University of Copenhagen, Copenhagen, Denmark. ✉E-mail: colin.hammond@liverpool.ac.uk; basraim@nih.gov

and ubiquitination of Lys49 and Lys124 in cell cycle dependent regulation of CENP-A protein levels (Wang et al, 2021).

Mislocalization of CENP-A (Cse4, Cnp1, and CID) to noncentromeric regions, contributes to CIN in budding yeast, fission yeast, and flies (Au et al, 2008; Choi et al, 2012; Gonzalez et al, 2014; Heun et al, 2006). Previously, we examined the consequence of CENP-A mislocalization in HeLa, RPE1, and DLD1 human cell lines, and provided the first evidence that the mislocalization of overexpressed CENP-A contributes to CIN by weakening the integrity of the native kinetochore (Shrestha et al, 2017; Shrestha et al, 2021). Furthermore, we have reported that CENP-A overexpressing DLD1 cells exhibited increased invasiveness, and aneuploidy with karyotypic heterogeneity (Shrestha et al, 2021). The mislocalization of overexpressed CENP-A can be prevented by depleting the H3.3–H4 chaperone DAXX (Lacoste et al, 2014) and this reduced the invasiveness of cancer cells and rescued CIN phenotypes (Shrestha et al, 2017; Shrestha et al, 2021). Depletion of another H3.3–H4 chaperone HIRA contributes to CENP-A mislocalization in colorectal (Nye et al, 2018) and cervical cancer cell lines (Lacoste et al, 2014; Shrestha et al, 2023) as well as in budding yeast (Ciftci-Yilmaz et al, 2018). Cancer cells overexpressing CENP-A have a proliferative advantage when treated with ionizing radiation and DNA-damaging agents in the context of defective p53 (Jeffery et al, 2021; Lacoste et al, 2014). In the presence of functional p53, CENP-A overexpression confers sensitivity to ionizing radiation (Jeffery et al, 2021). These findings have direct clinical relevance as CENP-A is reported to be overexpressed in various cancer types, including prostate, ovarian, lung, breast, and gastric cancers, and its overexpression correlates with poor prognosis, disease stage, genomic instability, and altered response to therapeutic interventions (Ma et al, 2003; Qiu et al, 2013; Saha et al, 2020; Sun et al, 2016; Wu et al, 2012; Xu et al, 2020; Zhang et al, 2016). Recent evidence also revealed that alterations in the subnuclear localization of CENP-A are predictive of therapeutic response in head and neck squamous cell carcinoma (Verrelle et al, 2021). Thus, identifying factors that promote or prevent CENP-A mislocalization to the noncentromeric regions may aid in the prognosis and treatment of CENP-A overexpressing cancers. Despite understanding the consequence of CENP-A mislocalization on CIN, we lack a comprehensive analysis of the pathways that promote or prevent CENP-A mislocalization.

We recently described an image-based analysis to quantify nuclear CENP-A levels in interphase cells using YFP-tagged CENP-A as a reporter (Shrestha et al, 2023). Using this assay and a focused RNAi screen of 521 genes encoding chromatin modifiers, we identified a novel role for the H3.1/2 chaperone CHAF1B in preventing mislocalization of CENP-A and CIN (Shrestha et al, 2023). Here, we report the employment of the same assay in a genome-wide RNAi screen to generate the first comprehensive identification of factors that prevent the mislocalization of CENP-A in human cells. The screen identified DNAJC9 (DnaJ homolog, subfamily C, member 9) as a critical regulator of proper CENP-A localization. DNAJC9 is a member of the J-domain containing heat-shock protein HSP40 family recently discovered as a co-chaperone for histone H3.1/2/3–H4 complex (Hammond et al, 2021; Han et al, 2007; Piette et al, 2021). The J domain of HSP40 proteins binds and stimulates the ATPase activity of the HSP70 molecular chaperones that regulate the proper folding of diverse proteins (Kampinga and Craig, 2010). In the present study, we

uncovered a hitherto uncharacterized role for DNAJC9 in preventing CENP-A mislocalization and CIN. Cells depleted for DNAJC9 showed enrichment of CENP-A in chromatin, mislocalization of CENP-A to noncentromeric regions, and CIN phenotypes. Interactome analysis of CENP-A in DNAJC9-depleted cells led to the identification of MCM2 as a top-ranked interactor and further studies revealed that MCM2 contributes to CENP-A mislocalization in DNAJC9-depleted cells. Overall, our studies define novel regulators of CENP-A localization and highlight the multifaceted roles of H3–H4 chaperones in restricting CENP-A localization and CIN.

# Results

## Genome-wide RNAi screen identifies factors that alter the levels of nuclear CENP-A

Several studies have established a correlation between increased nuclear CENP-A levels and mislocalization of CENP-A to noncentromeric regions (Athwal et al, 2015; Shrestha et al, 2017; Shrestha et al, 2023; Shrestha et al, 2021; Van Hooser et al, 2001) and hence, we used nuclear CENP-A levels as a readout for mislocalized CENP-A. To comprehensively analyze gene depletions that show altered nuclear CENP-A levels, we performed an image-based genome-wide RNAi screen using an siRNA library targeting 21,405 genes (three siRNAs per gene). The screen was performed using HeLa YFP-CENP-A$^{High}$ cells expressing 17-fold higher exogenous CENP-A levels than endogenous CENP-A in unmodified HeLa cells (Appendix Fig. S1A,B). Seventy-two hours post-siRNA transfection, we measured nuclear YFP fluorescence intensity as a reporter for YFP-CENP-A levels using a high-throughput imaging platform in 384-well plates. Using the average nuclear YFP-CENP-A intensity for each well, we determined a median YFP-CENP-A Z-score relative to a negative siRNA (siNeg) control and ranked candidate genes based on YFP-CENP-A Z-scores computed at the gene level (described in "Methods"). We used an siRNA targeting *CHAF1B*, identified previously (Shrestha et al, 2023) as a positive control.

The genome-wide RNAi screen identified 176 gene depletions that altered nuclear YFP-CENP-A intensity by more than twofold, of which 108 gene depletions showed increased nuclear YFP-CENP-A intensity (Fig. 1A; Dataset EV1). Consistent with our previous findings, depletion of CHAF1A or CHAF1B significantly enhanced nuclear YFP-CENP-A intensity (Shrestha et al, 2023) (Fig. 1A). Among the lead candidates whose depletions increased nuclear YFP-CENP-A intensity were genes regulating histone deposition, transcriptional activation, proteasome functions, and RNA processing (Dataset EV1). Gene depletions that reduced nuclear YFP-CENP-A intensity were associated with the general control of gene expression, including members of the mediator complex, RNA polymerase complex, or transcription elongation factors (Dataset EV1). For the validation screen, we selected gene candidates from the primary screen that when depleted altered nuclear YFP-CENP-A intensity by more than twofold, excluding regulatory factors associated with RNA or protein turnover. In addition, we included genes that encode components of the same complex, known regulators of CENP-A PTMs and centromeric localization, and histone chaperones, as part of the validation screen. The validation screen of 199 genes employed the same workflow as the

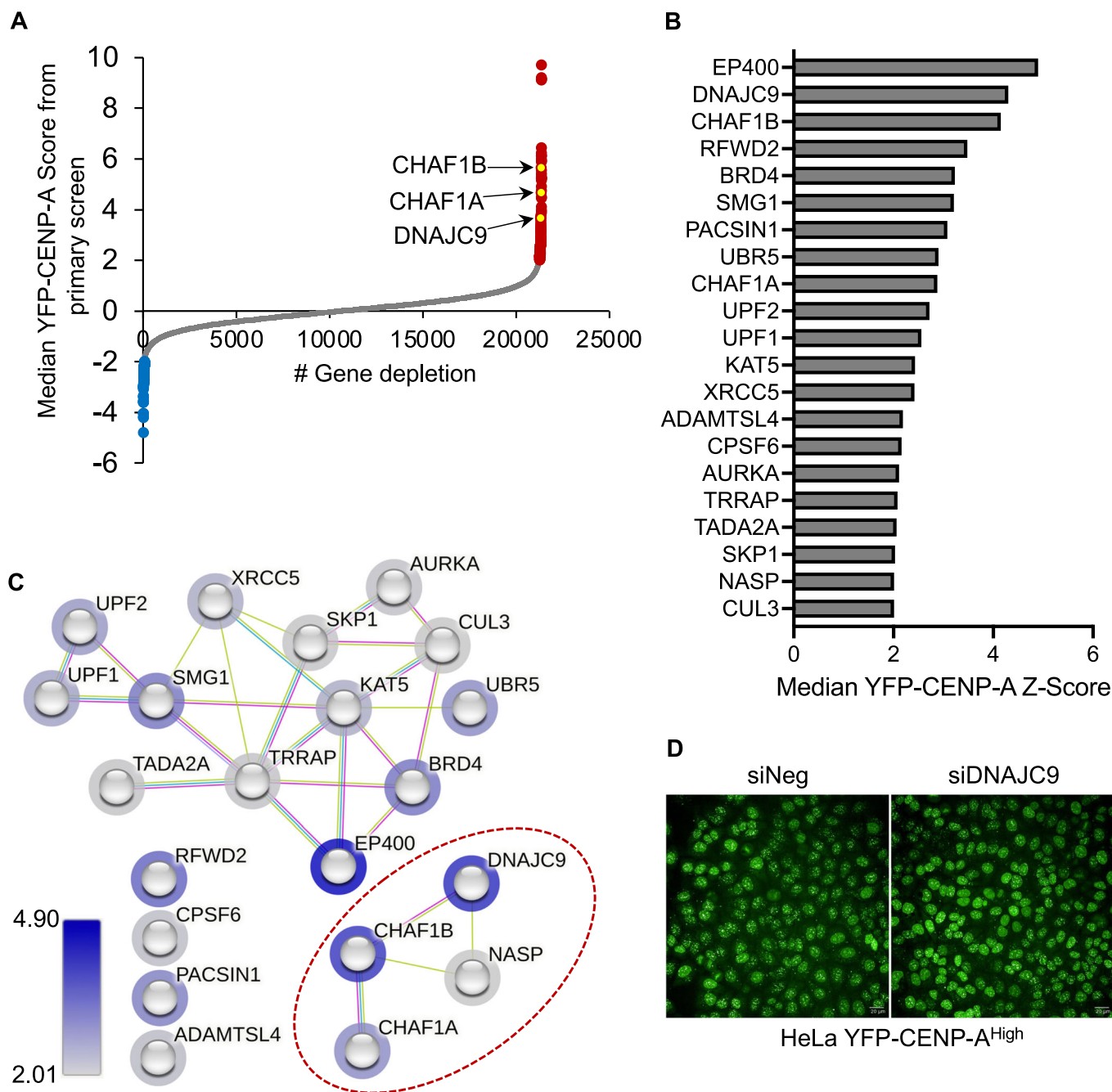

**Figure 1. Genome-wide image-based RNAi screen identifies factors that show altered nuclear CENP-A levels.**

(A) Scatter plot showing median YFP-CENP-A levels (Z-score) in HeLa YFP-CENP-A^High cells treated with siRNA library targeting 21,320 genes (three siRNAs per gene). Gene-level data is shown, with those that when targeted resulted in higher or lower nuclear YFP-CENP-A levels highlighted as red or blue dots, respectively. (B) Bar chart showing results of primary and secondary validation screen in HeLa YFP-CENP-A^High. siRNA treatments resulting in higher than twofold median YFP-CENP-A levels compared to the siNeg control are shown. (C) Functional analysis of candidates identified in (B) using the STRING database. Network from STRING analysis (https://string-db.org/) based on evidence from interaction sources that include "Experiments", "Databases" and "Textmining". The border of the node is colored based on the YFP-CENP-A Z-score values as shown in (B) and color grade. (D) Representative image from the high-throughput primary screen showing YFP-CENP-A signal in HeLa YFP-CENP-A^High cell line with negative control or DNAJC9 siRNA transfection. The images were acquired at a magnification of ×40 using the same imaging parameters. Scale bar: 20 μm. Source data are available online for this figure.

primary screen but utilized three siRNAs different than those used in the primary screen (Dataset EV2). We validated several gene depletions identified in the primary screen, including CHAF1A and CHAF1B amongst top candidates that promoted increased nuclear YFP-CENP-A levels as reported previously (Dataset EV2; Fig. 1B) (Shrestha et al, 2023). Critically, except for CHAF1A and CHAF1B, the molecular role of other candidates in regulating CENP-A levels or localization has not been defined.

## DNAJC9 depletion increases nuclear CENP-A levels throughout the cell cycle

DnaJ homolog, subfamily C, member 9 (DNAJC9) was amongst the top candidates identified in both our primary and secondary validation screens (Fig. 1A,B). Network analysis of candidates with nuclear YFP-CENP-A Z-scores above 2 in the primary and validation screens (Fig. 1B), identified a cluster of functional interactions among histone H3 chaperones comprising CHAF1A, CHAF1B, NASP, and DNAJC9 (Hammond et al, 2021; Hammond et al, 2017) (Fig. 1C, circled in red). Representative images from the primary screen showed increased nuclear YFP-CENP-A signal in cells treated with siDNAJC9 compared to the control (Fig. 1D). DNAJC9 is a member of the HSP40 family of heat-shock proteins (Han et al, 2007) and was recently demonstrated to have histone chaperone functionality towards histones H3–H4 (Hammond et al, 2021; Piette et al, 2021). Strikingly, DNAJC9 binds H3.1/2/3–H4 but not CENP-A–H4 (Hammond et al, 2021), and thus our results suggest that defective H3–H4 supply chains can contribute to CENP-A mislocalization (Fig. 1B,C). This notion is further supported by the identification of NASP, another histone chaperone required for maintaining soluble H3–H4 pool (Bao et al, 2022; Cook et al, 2011), amongst the top hits whose depletion leads to increased nuclear YFP-CENP-A intensity (Fig. 1B,C).

High levels of CENP-A expression alone can contribute to its mislocalization to noncentromeric regions and CIN phenotypes in a dose-dependent manner (Shrestha et al, 2017). Hence, we performed in-depth studies with DNAJC9 using a HeLa YFP-CENP-A^Low cell line which expresses around threefold higher levels of exogenous CENP-A compared to endogenous CENP-A in parental HeLa (Appendix Fig. S1A,B) but does not show constitutive CENP-A mislocalization or CIN phenotypes (Shrestha et al, 2017). First, we identified three siRNAs targeting DNAJC9 (siDNAJC9.1, siDNAJC9.2, and siDNAJC9.3) (Dataset EV3) that were effective for DNAJC9 depletion in comparison with negative control siRNA (siNeg) (Appendix Fig. S2). Since all siRNAs had comparable depletion efficiencies, we selected siDNAJC9.3 for further studies. To control for siRNA off-target effects, we further modified the HeLa YFP-CENP-A^Low cell line to express a doxycycline-inducible DNAJC9 cDNA resistant to siDNAJC9.3 (HeLa YFP-CENP-A^Low Tet-siRNA-resistant DNAJC9). Using these cells, we assessed the nuclear localization of CENP-A by immunofluorescence (IF) following transfection with siNeg or siDNAJC9.3, with or without doxycycline (DOX) treatment (Fig. 2A). As expected, DNAJC9-depleted cells showed significantly higher nuclear YFP-CENP-A levels compared to siNeg-transfected control cells without DOX treatment (Fig. 2A,B). In contrast, siDNAJC9.3-transfected cells with DOX-induced overexpression of the siRNA-resistant DNAJC9 cDNA showed nuclear YFP-CENP-A levels that did not differ from that of control cells, confirming that

the increased nuclear level of CENP-A is a specific consequence of DNAJC9 depletion (Fig. 2A,B).

We next tested whether the increased nuclear level of CENP-A observed in DNAJC9-depleted cells is restricted to a particular cell cycle stage, given that noncentromeric CENP-A has been previously observed in the G1 phase of the cell cycle which is cleared during DNA replication (Nechemia-Arbely et al, 2019). Using the nucleotide analog 5-Ethynyl-2′-deoxyuridine (EdU) to mark DNA synthesis and total 4′,6-diamidino-2-phenylindole (DAPI) intensity to gate on different cell cycle populations (Appendix Fig. S3A–C), we were able to observe increased nuclear levels of CENP-A at all cell cycle stages upon DNAJC9 depletion relative to the control conditions (Fig. 2C,D). Importantly, DNAJC9 depletion did not significantly alter the cell cycle profile (Appendix Fig. S3C–E). Taken together, our results show that increased nuclear CENP-A level upon DNAJC9 depletion is independent of the cell cycle.

## DNAJC9-depleted cells exhibit mislocalization of CENP-A and CIN phenotypes

We analyzed metaphase chromosome spreads from control and DNAJC9-depleted HeLa YFP-CENP-A^Low cells to examine the localization of CENP-A on mitotic chromosomes using anti-CENP-A antibody (Fig. 3A). Our results showed significantly increased levels of CENP-A at noncentromeric regions in DNAJC9-depleted cells compared to control cells (5.6-fold change in median value; Fig. 3A,B). We also observed a moderate but significant increase in centromeric CENP-A levels in DNAJC9-depleted cells (1.3-fold; Fig. 3B).

To determine whether the exogenous YFP-CENP-A is stably bound to chromatin, YFP-CENP-A localization was analyzed using IF of chromosome spreads after pre-treating cells with a high salt (500 mM NaCl) buffer. Similar conditions have been employed previously, demonstrating that CENP-A stably incorporated in chromatin resists high salt extraction (Bobkov et al, 2018). In control cells, YFP-CENP-A was mainly localized at the centromere, as defined by co-immunostaining for the CENP-C subunit of the Constitutive Centromere Associated Network (CCAN) complex (Carroll et al, 2010) (Fig. EV1A,B). Depletion of DNAJC9 resulted in significantly higher YFP-CENP-A levels at the noncentromeric regions (4.3-fold increase) relative to control siNeg cells (Fig. EV1A,B). These results show that mislocalized exogenous CENP-A is stably associated with noncentromeric chromatin. To exclude the effect of overexpression or tagging of CENP-A on its localization, we examined the localization of endogenous CENP-A in parental HeLa cells depleted of DNAJC9. Our results showed increased CENP-A levels upon DNAJC9 depletion, primarily at noncentromeric regions (2.7-fold change) (Fig. EV1C,D). Thus, DNAJC9 depletion promotes CENP-A mislocalization and the extent of this is influenced by CENP-A expression levels.

Studies from our laboratory and those of others have shown that mislocalization of overexpressed CENP-A leads to mislocalization of CENP-C (Shrestha et al, 2017; Shrestha et al, 2021; Van Hooser et al, 2001). In addition, cells with CENP-A mislocalization exhibit reduced levels of the outer kinetochore protein NUF2 at the native kinetochore, which is linked to mitotic defects (Shrestha et al, 2017; Shrestha et al, 2021). Consistent with these results, we observed that CENP-A mislocalization upon depletion of DNAJC9 (Fig. 3A,B) is

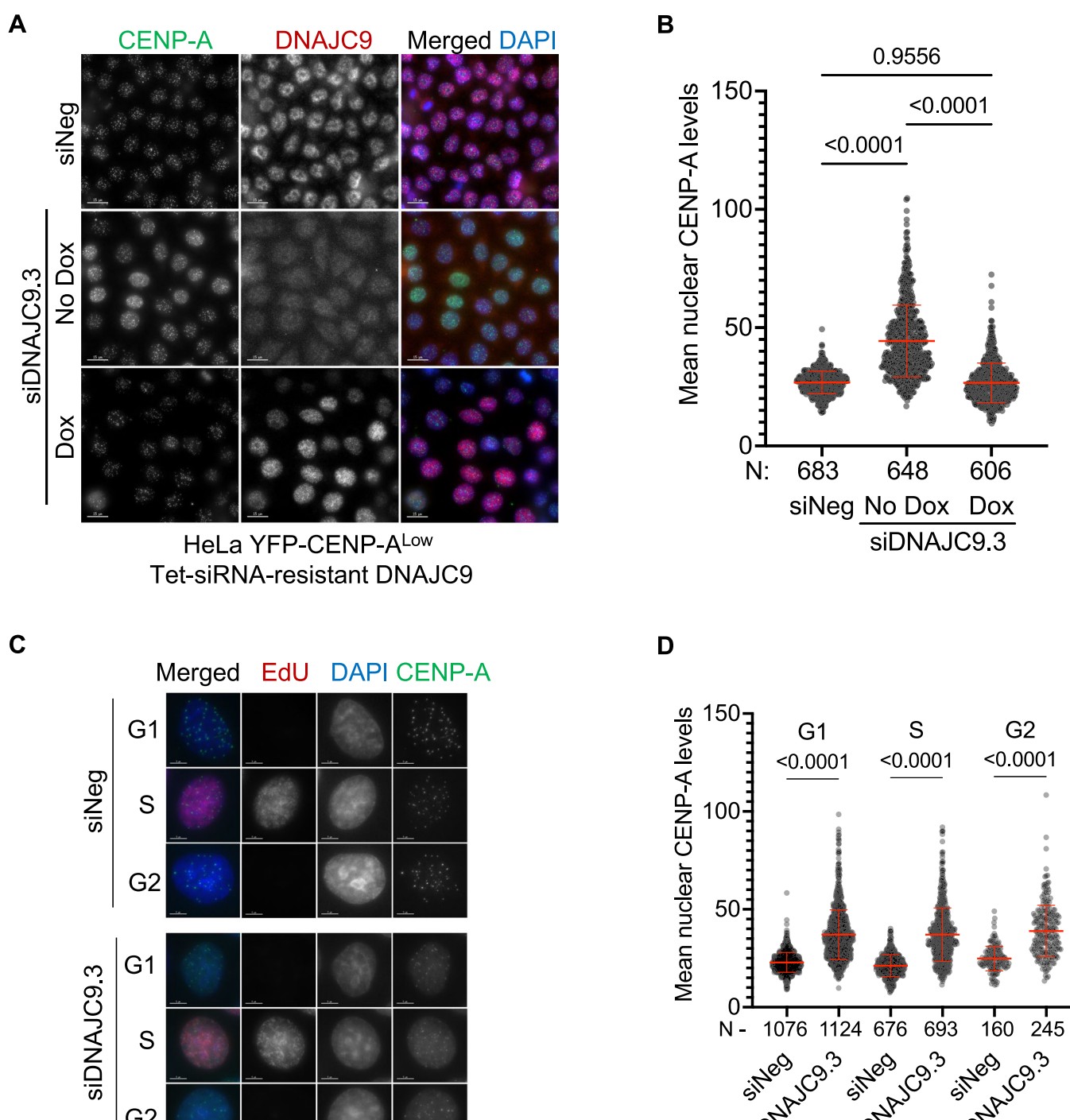

**A**

CENP-A DNAJC9 Merged DAPI

siNeg

siDNAJC9.3 — No Dox — Dox

HeLa YFP-CENP-A$^{Low}$
Tet-siRNA-resistant DNAJC9

**B**

Mean nuclear CENP-A levels

0.9556

<0.0001

<0.0001

N: 683 648 606
siNeg No Dox Dox
siDNAJC9.3

**C**

Merged EdU DAPI CENP-A

siNeg — G1 — S — G2

siDNAJC9.3 — G1 — S — G2

**D**

Mean nuclear CENP-A levels

G1    S    G2
<0.0001 <0.0001 <0.0001

N - 1076 1124 676 693 160 245
siNeg siDNAJC9.3 siNeg siDNAJC9.3 siNeg siDNAJC9.3

accompanied by the mislocalization of CENP-C to noncentromeric regions in HeLa YFP-CENP-A$^{Low}$ cells (twofold; Fig. 3C,D), without significant effects at the centromere. Thus, the moderate increase in CENP-A levels at the centromeric region observed upon DNAJC9 depletion (Fig. 3B) are insufficient to significantly affect the centromeric accumulation of CENP-C (Fig. 3D). This can be explained by a non-stoichiometric association between CENP-A and CENP-C expression levels, which would lead to the preferential localization of CENP-C to regions with significantly higher levels of

CENP-A. The mislocalization of CENP-A and CENP-C we observe upon DNAJC9 depletion is also accompanied by a reduction of NUF2 levels at the native kinetochore (Fig. 3E), as reported in our previous studies (Shrestha et al, 2017; Shrestha et al, 2021).

We next examined the physiological consequences of DNAJC9's regulation of CENP-A, CENP-C, and NUF2 localization on chromosome segregation. Our results showed that DNAJC9-depleted HeLa YFP-CENP-A$^{Low}$ cells exhibit CIN phenotypes linked to mitosis, including a higher incidence of micronuclei

 

**Figure 2. Increased nuclear CENP-A levels upon depletion of DNAJC9 in HeLa YFP-CENP-A^Low.**

(**A**) Nuclear CENP-A levels are increased in DNAJC9-depleted cells. Immunofluorescence images showing nuclear CENP-A levels in HeLa YFP-CENP-A^Low cells expressing DOX-inducible siRNA-resistant DNAJC9. Cells were transfected with either siNeg or siDNAJC9.3. DOX-induced expression of the siRNA-resistant DNAJC9 restored the increased nuclear CENP-A levels in siDNAJC9.3-transfected cells. Scale bar: 15 μm. (**B**) Scatter plot showing nuclear CENP-A intensities for conditions as described in (**A**). N represents the total number of cells analyzed per condition from three biological replicates and mean with standard deviation (SD) is shown. *P* values were derived from one-way ANOVA with Tukey's ad hoc test. (**C**) Increased nuclear CENP-A levels are observed throughout the cell cycle in DNAJC9-depleted cells. Representative IF images showing interphase nuclei from control or DNAJC9.3 siRNA-transfected cells immunostained for CENP-A and EdU. Cells from three biological replicates were sorted to G1, S, or G2 stages based on EdU and DAPI signal intensities as described in Appendix Fig. S3A,B. Scale bar: 5 μm. (**D**) Scatter plot showing nuclear CENP-A intensities for G1, S, or G2 cells as described in (**C**). N represents the total number of cells analyzed per condition from three biological replicates and mean with standard deviation (SD) is shown. *P* values were derived from one-way ANOVA with Tukey's ad hoc test. Source data are available online for this figure.

and defective chromosome segregation (Fig. 3F,G). These phenotypes were not evident in parental HeLa cells depleted of DNAJC9 (Fig. EV1E), which display lower levels of CENP-A mislocalization (Fig. EV1C,D). However, depletion of DNAJC9 in HeLa YFP-CENP-A^High cells, which display higher CENP-A mislocalization and CIN (Shrestha et al, 2017), led to a further increase in CENP-A mislocalization and chromosome segregation defects (Fig. EV1F–H). These results show that CENP-A expression and mislocalization above a certain threshold make CIN phenotypes more penetrant in DNAJC9-depleted cells. We conclude that disruption of DNAJC9 function causes mislocalization of CENP-A and CIN phenotypes.

## DNAJC9 depletion stabilizes CENP-A and contributes to its enrichment in chromatin

The mislocalization of CENP-A to noncentromeric regions in DNAJC9-depleted cells led us to examine whether DNAJC9 regulates CENP-A expression. To this end, we analyzed *CENP-A* transcript levels and demonstrated that the effective depletion of *DNAJC9* does not affect the levels of *CENP-A* mRNA (Fig. 4A). We reasoned that CENP-A mislocalization could be due to the increased stability of CENP-A. Cycloheximide (CHX) chase experiments were performed to assess YFP-CENP-A protein stability after inhibiting protein synthesis using whole-cell extracts prepared from control or DNAJC9-depleted cells. Our results demonstrated that YFP-CENP-A is more stable in DNAJC9-depleted cells (Fig. 4B,C). We next examined if the higher CENP-A protein stability correlates with its increased chromatin enrichment. Western blot analysis of soluble and chromatin fractions from control and DNAJC9-depleted cells showed significantly higher enrichment of both YFP-tagged and endogenous CENP-A in the chromatin fraction of DNAJC9-depleted cells (Fig. 4D,E). Based on these observations, we conclude that CENP-A protein is more stable and exhibits a higher association with chromatin in DNAJC9-depleted cells.

## Loss of DNAJC9 promotes the association of CENP-A with H3–H4 nucleosome assembly pathways

To define the mechanisms driving the mislocalization of CENP-A, we employed a proteomics-based approach assessing how depletion of DNAJC9 affects the interactome of overexpressed CENP-A purified from soluble or chromatin fractions, using immunoprecipitation coupled to label-free mass spectrometry (IP-MS) analysis (Figs. 5, EV2, EV3, and EV4; Dataset EV4). For these experiments, we used the DOX-inducible CENP-A FLAG-HA HeLa S3 cell line (Hammond

et al, 2021), and in addition to DNAJC9 depletion, we included CAF1B (CHAF1B) depletion as this also promotes CENP-A mislocalization (Shrestha et al, 2023). Western blot analysis confirmed the efficiency of siRNA silencing; and, as expected, ectopic CENP-A levels were higher in siDNAJC9 and siCAF1B compared to siCTRL (siNeg) transfected control conditions (Fig. 5A). Furthermore, the consistency of MNase digestion levels was confirmed across all chromatin extracts (Fig. 5A). Control purifications without DOX-induced CENP-A expression (-DOX) were performed to set statistical thresholds allowing the identification of factors specifically enriched in CENP-A purifications in both cellular fractions (Figs. EV2A,B and EV3A,B; Dataset EV4). Of the significantly enriched proteins in soluble (955) and chromatin (1045) CENP-A IP-MS experiments, about half (486) were enriched with CENP-A in both cellular fractions (Fig. EV3G; Dataset EV4). To provide an overview of the biological processes enriched with soluble and chromatin-bound CENP-A, we performed gene ontology (GO) analysis using the STRING-db ranked list functional enrichment analysis, assessing the redundancy of enriched terms using clustering analysis (Fig. EV4). This analysis revealed an enrichment of GO-terms related to nucleosome assembly factors for CENP-A and DNA-replication-dependent and -independent nucleosome assembly, transcription, nucleotide excision repair, and isoprenoid metabolism (Fig. EV4; Dataset EV4).

To identify potential mechanisms contributing to CENP-A mislocalization upon depletion of DNAJC9 or CAF1B, we compared the intensity of proteins identified as specifically enriched in soluble and chromatin CENP-A purifications (Figs. 5B, EV2B, and EV3B) across our siRNA conditions (Fig. 5C), using bait normalization to account for differences in CENP-A levels (Figs. 5C, EV2C–F, and EV3C–F). For the soluble CENP-A purifications, we observed an enrichment of 19 proteins in siCAF1B, 56 proteins in siDNAJC9, and 10 proteins (MCM2, SP16H/SPT16, NSL1, KIF11, FACD2, RPTOR, ESYT2, MELK, CSRP2, PTPM1; Fig. EV2F; Dataset EV4) in both siDNAJC9 and siCAF1B conditions compared to the control. Notably, depletion of DNAJC9 revealed a striking enrichment of histone chaperones functionally linked to DNAJC9 (Hammond et al, 2021) involved in DNA replication (MCM2, TONSL), transcription (SPT2) or both (SPT16/SP16H and SSRP1: the FACT complex) (Figs. 5C and EV2D). Intriguingly NSL1, which is part of the Mis12 complex required for kinetochore assembly (Kline et al, 2006) was highly enriched in both DNAJC9 and CAF1B-depleted soluble CENP-A datasets, despite the lack of other Mis12 complex members in these purifications (Figs. 5C and EV2E).

In the chromatin fraction, depletion of DNAJC9 or CAF1B enhanced the interaction of CENP-A with 140 and 35 proteins, respectively, and 30 of these were enriched in both conditions

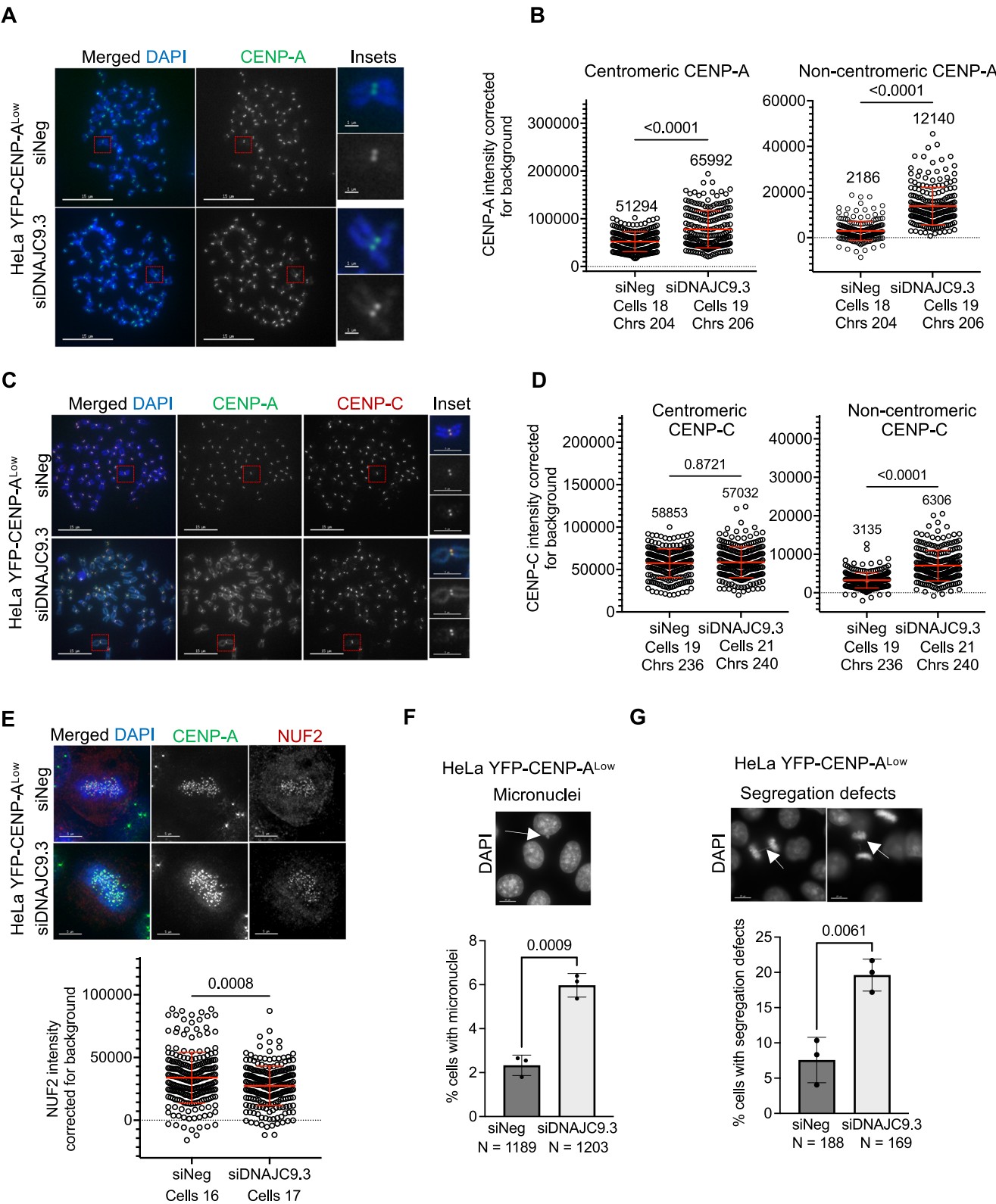

**Figure 3. DNAJC9 depletion contributes to mislocalization of CENP-A and CIN in HeLa YFP-CENP-A^Low cells.**

(A) CENP-A is mislocalized in DNAJC9-depleted HeLa YFP-CENP-A^Low cells. Representative images of metaphase chromosome spreads immunostained for CENP-A in siNeg or siDNAJC9.3-transfected HeLa YFP-CENP-A^Low cells. Scale bar: 15 μm. Scale bar for insets: 1 μm. (B) Scatter plots showing CENP-A intensities corrected for background at the centromeric or noncentromeric regions as described in (A), from three biological replicates. Median values are shown above graph. Each circle represents the value from an individual chromosome. Chrs refers to the total number of chromosomes measured per condition from three biological replicates. The mean with standard deviation is shown across all measurements from three biological replicates. *P* values were calculated using Mann–Whitney *U* test. (C) CENP-C is mislocalized in DNAJC9-depleted HeLa YFP-CENP-A^Low cells. Representative images of metaphase chromosome spread immunostained for CENP-C and CENP-A in siNeg or siDNAJC9.3-transfected cells. Scale bar: 15 μm. Scale bar for insets: 1 μm. (D) Scatter plots showing CENP-C intensities corrected for background at the centromeric or noncentromeric regions as described in (C), from three biological replicates. Median values are shown above graph. Each circle represents the value from an individual chromosome. Chrs refers to the total number of chromosomes measured per condition from three biological replicates. Mean with standard deviation is shown across all measurements from three biological replicates. *P* values were calculated using Mann–Whitney *U* test. (E) NUF2 levels are reduced at the kinetochores in DNAJC9-depleted HeLa YFP-CENP-A^Low cells. Top panel: Representative images of fixed cell immunofluorescence images showing the localization of NUF2 and CENP-A in metaphase cells transfected with siNeg or siDNAJC9.3. Scale bar: 5 μm. Bottom panel: Scatter plots showing NUF2 intensities at the kinetochore corrected for background. Indicated number of cells were analyzed from three biological replicates and each circle represents the value from an individual kinetochore. KTs refers to the total number of kinetochores measured per condition from three biological replicates. Mean with standard deviation is shown across all measurements. *P* values were calculated using Mann–Whitney *U* test. (F, G) Increased incidence of CIN phenotypes in HeLa YFP-CENP-A^Low cells. Representative IF images showing micronuclei (F) and defects in chromosome segregation (G) indicated by white arrowheads in DNAJC9-depleted HeLa YFP-CENP-A^Low cells. Scale bar: 15 μm. Bar graphs show percent cells with micronuclei (F) and defective chromosome segregation (G) in cells transfected with control (siNeg) or siDNAJC9.3. Mean with SD was plotted from three biological replicates and *P* value was calculated using unpaired *t* test. *N* denotes the total number of cells analyzed per condition. Source data are available online for this figure.

compared to the control (Fig. EV3F; Dataset EV4). Consistent with the mislocalization of CENP-A to noncentromeric regions, we observed a reduced association of centromeric proteins with chromatin-bound CENP-A upon depletion of DNAJC9 or CAF1B (Figs. 5C and EV3D,E). This included members of the CCAN complex, which regulate kinetochore function (Foltz et al, 2006; Izuta et al, 2006; Obuse et al, 2004; Okada et al, 2006), and CENP-A deposition factors (HJURP and Mis18 complex members MIS18B and MIS18BP) (Dunleavy et al, 2009; Foltz et al, 2009; Fujita et al, 2007; Hayashi et al, 2004). The Mis18 complex member MIS18A was not reliably detected in all IP-MS conditions analyzed and therefore, fold changes could not be determined. In contrast, a variety of noncentromeric chromatin regulators (e.g., the NuRD complex and DNMT3A) were enriched with CENP-A upon DNAJC9 or CAF1B depletion, consistent with enhanced deposition outside centromeres. Chromosomal Passenger Complex members (AURKB/Aurora kinase B, BOREA/borealin, BIRC5/survivin, INCE/INCENP) were also consistently enriched with CENP-A in the chromatin upon DNAJC9 depletion, and of these proteins, only BIRC5 was significantly enriched in the CAF1B-depleted cells; however, the possible consequence of this enhanced interaction is not clear.

Notably, we observed higher levels of histone H3–H4 chaperones implicated in DNA replication (Hammond et al, 2017) including MCM2, TONSL, and FACT associated with CENP-A when DNAJC9 was depleted (Fig. 5C). This enhanced association was particularly strong in soluble CENP-A IP-MS datasets, suggestive of a plausible link between CENP-A mislocalization and DNA-replication-coupled nucleosome assembly pathways. Taken together, our interactome analysis reveals specific and overlapping factors that are enriched with CENP-A upon depletion of CAF1B or DNAJC9. This suggests that perturbation of histone chaperone levels has a major effect on the histone supply chain equilibrium and may be a key driving force for the mislocalization of CENP-A.

## MCM2 contributes to CENP-A mislocalization in DNAJC9-depleted cells

Given the enrichment of replication-coupled H3–H4 chaperones in our CENP-A interactome analyses (Fig. 5C), and the histone co-

chaperone relationship of MCM2 and DNAJC9 (Hammond et al, 2021), we decided to characterize whether MCM2 plays a role in promoting the mislocalization of CENP-A in cells depleted for DNAJC9. This is plausible since MCM2 has been implicated in recycling CENP-A–H4 during DNA replication at the centromere, and the histone-binding mode of MCM2 is compatible with CENP-A–H4 association (Huang et al, 2015; Zasadzinska et al, 2018).

To examine whether the enhanced interaction of CENP-A with MCM2 contributes to CENP-A mislocalization in DNAJC9-depleted cells, we co-depleted MCM2 and DNAJC9 in HeLa YFP-CENP-A^Low cells. RT-qPCR analysis confirmed reduced expression of *DNAJC9* and *MCM2* in cells transfected with the siRNAs targeting the respective transcripts without a significant effect on the expression of *CENP-A* when compared to control cells (Appendix Fig. S4A–C). Western blot analysis showed efficient depletion of DNAJC9, however, depletion of MCM2 was partial (Fig. 6A). We next analyzed CENP-A localization using mitotic chromosome spreads prepared from control cells and DNAJC9 and/or MCM2-depleted cells. As expected, we observed increased CENP-A levels in siDNAJC9.3-transfected cells at both centromeric and noncentromeric regions when compared to siNeg-transfected cells (Fig. 6B,C). MCM2 depletion alone resulted in reduced centromeric CENP-A levels relative to control cells, consistent with previous observation for a role of MCM2 in the centromeric association of CENP-A (Zasadzinska et al, 2018). Cells co-depleted for both MCM2 and DNAJC9 showed reduced levels of CENP-A at both centromeric and noncentromeric regions (Fig. 6B,C). Based on these results, we conclude that enhanced interaction of MCM2 with CENP-A contributes to CENP-A mislocalization in DNAJC9-depleted cells.

## Catalytically inactive DNAJC9 and mis-regulated H3–H4 supply promote CENP-A mislocalization

DNAJC9 has two domains, the catalytic J domain that recruits HSP70 activity and a histone H3–H4 binding domain (Hammond et al, 2021). This combined functionality serves to maintain the supply of properly folded H3–H4 dimers for deposition on chromatin during DNA replication and transcription (Hammond et al, 2021). To establish which domains of DNAJC9 are essential to

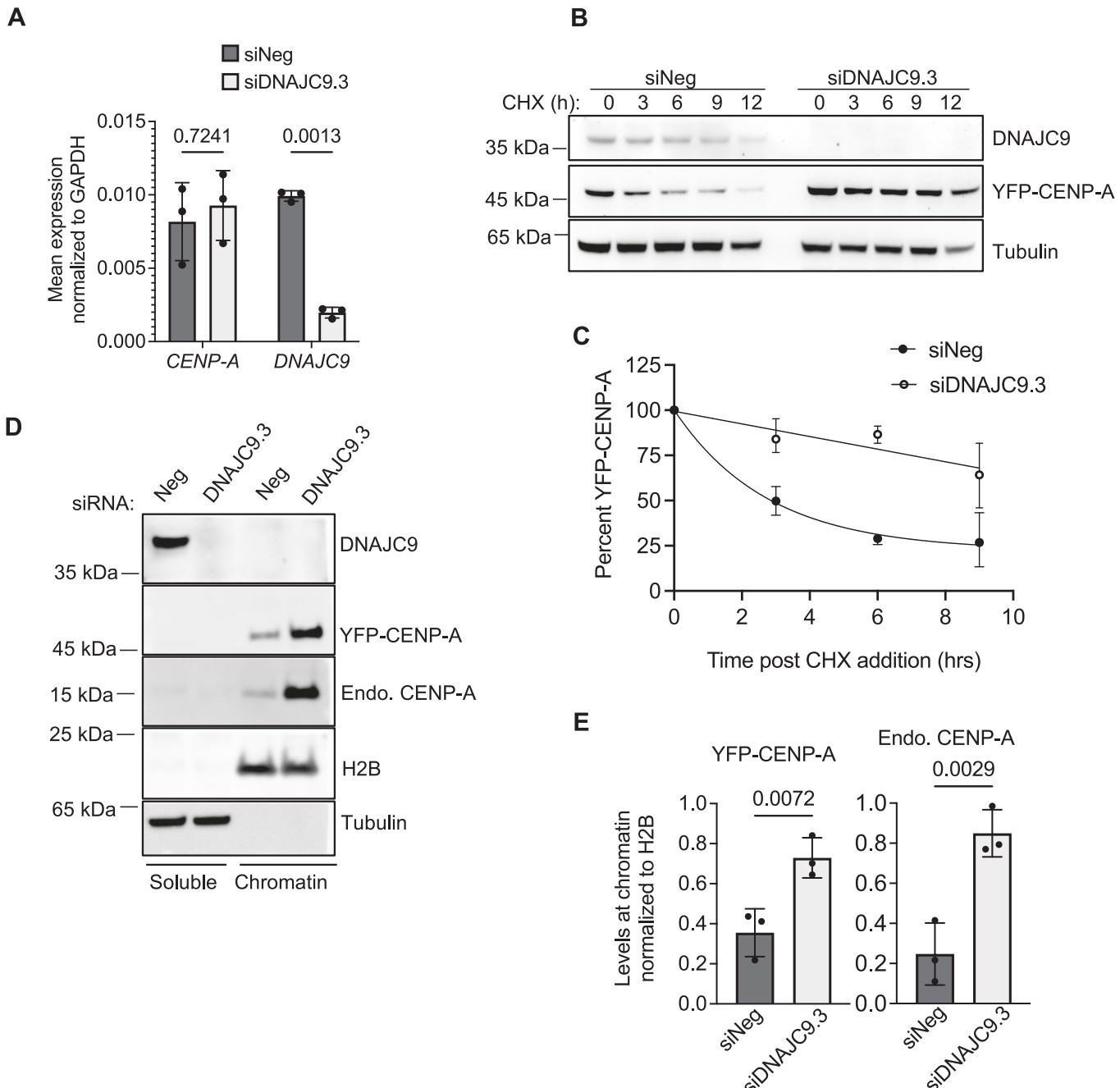

**Figure 4. DNAJC9 depletion stabilizes CENP-A and contributes to its enrichment in chromatin.**

(A) Transcript levels of *CENP-A* are not altered in DNAJC9-depleted HeLa YFP-CENP-A^Low cells. Bar graphs showing mean RNA levels of *CENP-A* and *DNAJC9* normalized to *GAPDH* in control and DNAJC9-depleted cells using RT-qPCR. Mean values with standard deviation from three biological replicates were plotted and *P* values were calculated from two-way ANOVA with Sidak's multiple correction test. (B) Stability of YFP-CENP-A is higher in DNAJC9-depleted HeLa YFP-CENP-A^Low cells. Western blot of whole-cell extracts showing DNAJC9 depletion and YFP-CENP-A levels in control or DNAJC9-depleted cells treated with 100 μg/ml cycloheximide for the indicated time periods. Alpha-tubulin was used as the loading control. (C) Line graph for results shown in B. Mean with SD from three biological replicates were plotted. (D) Enrichment of CENP-A in chromatin fractions from DNAJC9-depleted HeLa YFP-CENP-A^Low cells. Western blot of soluble and chromatin fractions prepared from control or DNAJC9-depleted cells showing levels of YFP-CENP-A and endogenous CENP-A on chromatin in HeLa YFP-CENP-A^Low. Alpha-tubulin and H2B were used as markers of soluble and chromatin fractions, respectively. (E) Bar graphs showing the levels of YFP-CENP-A or endogenous CENP-A on the chromatin normalized to H2B control in the indicated conditions. Mean with SD were plotted from three biological replicates. *P* values were calculated using unpaired *t* test. Source data are available online for this figure.

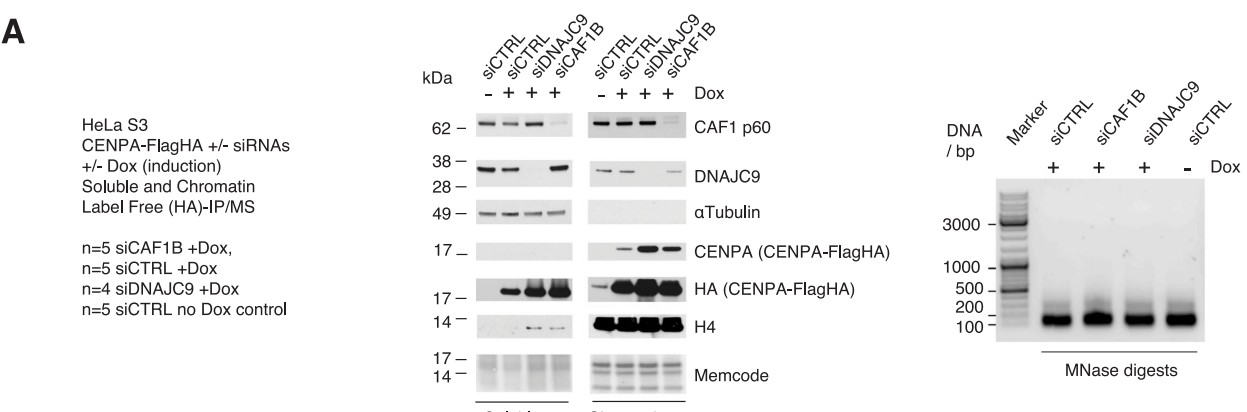

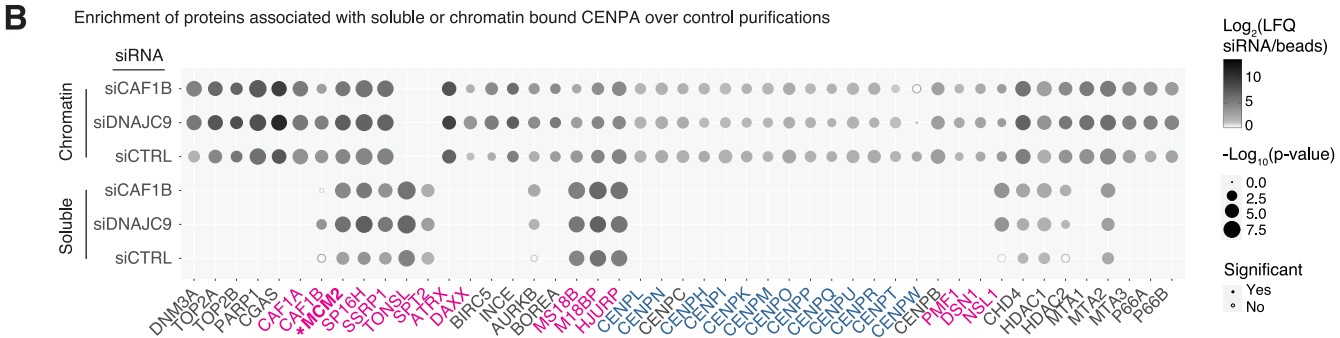

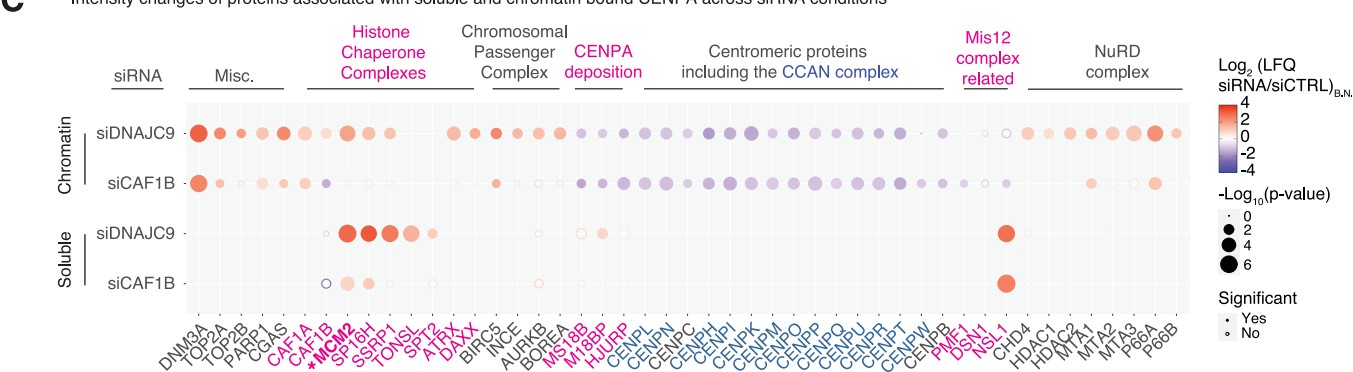

**Figure 5. CENP-A interactomes in soluble and chromatin fractions upon DNAJC9 and CAF1B depletion.**

(A) (Left) Overview of soluble and chromatin label-free CENP-A-Flag-HA IP-MS experiments, siDNAJC9 biological replicate two omitted due to lack of knockdown. Middle: Western blots of soluble and chromatin extracts upon siRNA depletion from cells expressing CENP-A-Flag-HA (DOX + ) and uninduced controls (no DOX). Alpha-Tubulin and H4 were used as loading controls for soluble and chromatin fractions, respectively. Right: MNase digest levels in chromatin extracts. Representative of $n = 4$ biological replicates for siDNAJC9 and $n = 5$ biological replicates for other conditions. (B, C) Bubble plots from soluble and chromatin fraction CENP-A-Flag-HA IP-MS experiments, closed circles represent significant changes, as assessed by two-sided $T$ tests. Statistical parameters in panel B were S0 = 0, FDR = 0.01 and a minimum Log2 fold change of 1.5, and for (C) the parameters were S0 = 0.1 and FDR = 0.05. Proteins referred to by human UniProt protein identification code. See also data analysis steps detailed in Figs. EV2 and EV3 and Dataset EV4, and gene ontology analysis in Fig. EV4 and Dataset EV4. (B) Enrichment of factors associated with soluble and chromatin-bound CENP-A across siRNA-treated conditions compared to control conditions without CENP-A-Flag-HA induction (Dox-). Ratios calculated using raw label-free quantification intensities (LFQ). (C) Enrichment (red) and depletion (blue) of proteins specifically associated with CENP-A-Flag-HA in siDNAJC9 and siCAF1B conditions compared to siCTRL conditions. Ratios calculated from bait normalized label-free quantification intensities (LFQ$_{B.N.}$). Source data are available online for this figure.

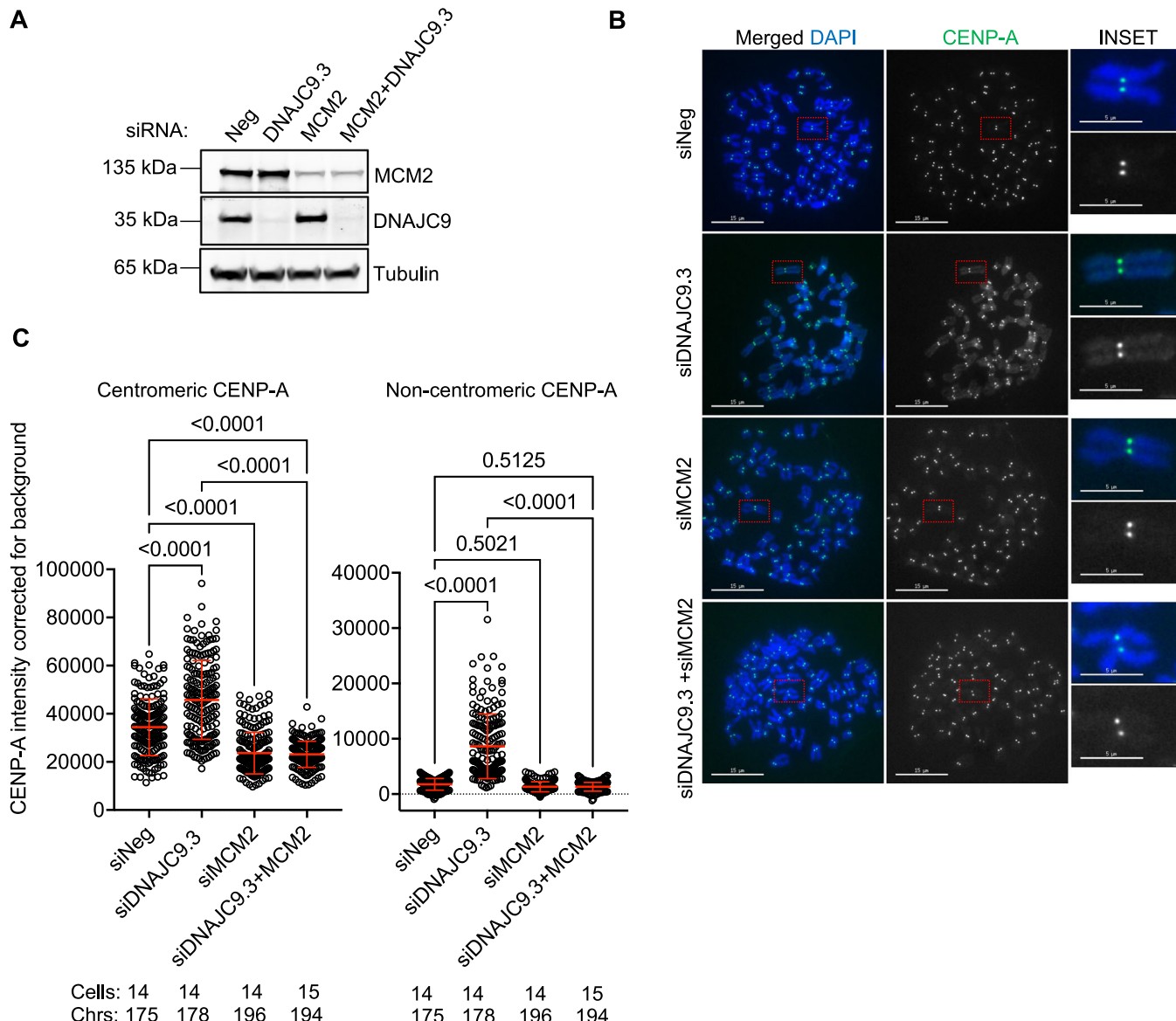

**Figure 6.  MCM2 contributes to CENP-A mislocalization in DNAJC9-depleted cells.**

(**A**) Western blot of whole-cell extracts prepared from control or DNAJC9-depleted cells with or without MCM2 depletion showing the efficiency of DNAJC9 and MCM2 depletions in HeLa YFP-CENP-A$^{Low}$. Alpha-tubulin was used as the loading control. Representative images from three biological replicates are shown. (**B**) MCM2 depletion suppresses CENP-A mislocalization in DNAJC9-depleted HeLa YFP-CENP-A$^{Low}$ cells. Representative images of metaphase chromosome spreads immunostained for CENP-A in control or DNAJC9.3 siRNA-transfected cells with or without MCM2 depletion in HeLa YFP-CENP-A$^{Low}$ from three biological replicates. Scale bar: 15 μm. Scale bar for inset: 5 μm. (**C**) Scatter plots showing CENP-A intensities corrected for background at the centromeric or noncentromeric regions as described in B. Each dot represents value from the individual chromosome. Chrs represent the total number of chromosomes measured per condition from three biological replicates. Red horizontal lines represent mean signal intensity and error bars represent SD across all measurements from three biological replicates. *P* values were derived from one-way ANOVA with Tukey's ad hoc test. Source data are available online for this figure.

prevent mislocalization of CENP-A, we transfected HeLa YFP-CENP-A$^{Low}$ with FLAG-tagged DNAJC9 wild-type (WT) or mutant DNAJC9 constructs that disrupt the catalytic activity of the J domain, the histone-binding domain, or both domains (Hammond et al, 2021) (mutants J, 4A, and 4AJ, respectively; Fig. 7A). Seventy-two hours post-transfection, we observed that cells expressing the J mutant, but not others, exhibited higher nuclear CENP-A signals which are representative of noncentromeric CENP-A localization (Fig. 7B,C). The DNAJC9 J mutant traps histones H3–H4 and

causes them to accumulate in the soluble fraction, due to an inability to properly engage the protein folding activity of HSP70-type enzymes (Hammond et al, 2021).

These results suggest that the mislocalization of CENP-A is linked to a reduced supply of properly folded H3–H4. However, cells expressing the histone-binding mutants (4A or 4AJ) do not exhibit increased nuclear CENP-A levels, probably due to the lack of a dominant negative effect of these mutants over the endogenous DNAJC9 function. This was expected since histone-binding mutant

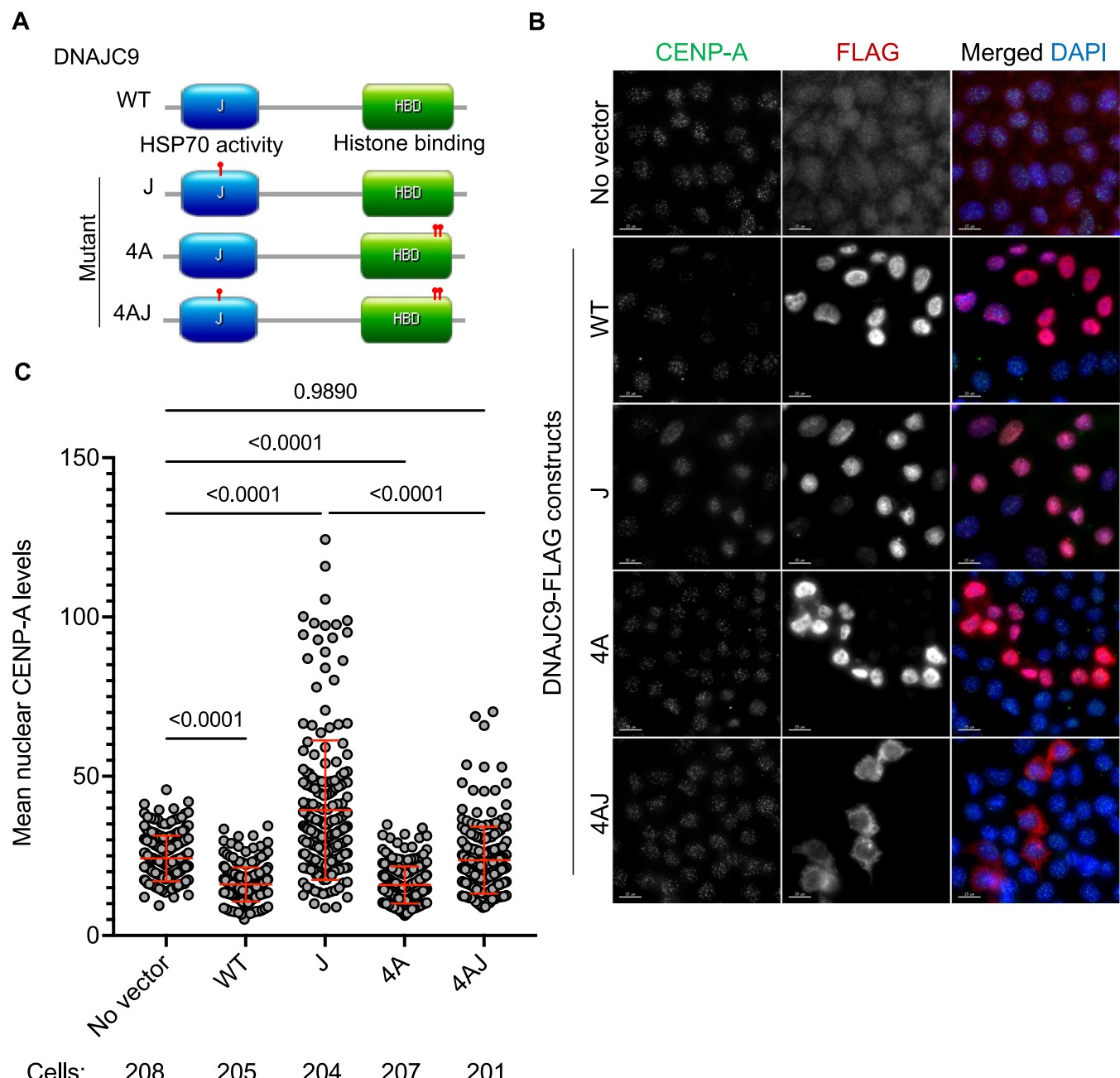

**Figure 7. Catalytically inactive DNAJC9 promotes CENP-A mislocalization in a H3-dependent manner.**

(A) Schematic representing the organization of domains in DNAJC9 and sites of mutations in the J domain and/or histone-binding domain (HBD) marked with red pointers. WT wild-type DNAJC9, J mutant DNAJC9 with mutation in J domain, 4A mutant DNAJC9 with mutations in histone-binding domain, 4AJ DNAJC9 with mutations in histone binding and J domains. (B) Increased nuclear CENP-A levels are observed in cells expressing catalytically inactive J mutant. Immunofluorescence images showing nuclear CENP-A levels in HeLa YFP-CENP-A$^{Low}$ cells from three independent experiments, post transient transfection with the indicated constructs as described in (A), for 72 h. Scale bar: 15 μm. (C) Scatter plot showing nuclear CENP-A intensities for conditions as described in (A). The total number of cells analyzed per condition from three biological replicates are indicated below the graph. Mean with standard deviation (SD) is shown. *P* values were derived from one-way ANOVA with Tukey's ad hoc test. Source data are available online for this figure.

expression alone does not lead to the accumulation of soluble histones (Hammond et al, 2021). Therefore, as an independent approach to phenocopy reduced H3–H4 supply, we depleted histone H3.3 in HeLa YFP-CENP-A$^{Low}$ cells by siRNA-mediated targeting of one of the two H3.3-encoding genes (*H3F3A*). Even

with a partial reduction of total H3.3 levels (Appendix Fig. S5A), we observed significantly higher noncentromeric CENP-A levels (2.66-fold) with nonsignificant changes at the centromere (Appendix Fig. S5B,C). Together, these findings indicate that a blockage in the H3–H4 supply chain provides an opportunity for CENP-A to

highjack components of H3–H4 nucleosome assembly pathways resulting in its mislocalization along chromosome arms.

## Genome-wide mislocalization of CENP-A is observed upon DNAJC9 depletion

We next tested whether CENP-A is mislocalized upon DNAJC9 depletion in the context of immortalized, but non-transformed, hTERT-RPE1 cells which are near diploid cells with functional p53. We constructed an inducible RPE1-Tet-GFP-CENP-A cell line and examined whether DNAJC9-depleted cells exhibit CENP-A mislocalization after DOX induction. Western blotting confirmed the expression of GFP-CENP-A after DOX induction and efficient depletion of DNAJC9 (Fig. EV5A). As observed in HeLa YFP-CENP-A^Low cells (Fig. 4A), transcript levels of *CENP-A* were not significantly altered by DNAJC9 depletion in DOX-treated cells (Fig. EV5B). Analysis of metaphase chromosome spreads from DOX-treated control and siDNAJC9-transfected cells revealed higher levels of CENP-A at both centromeric and noncentromeric regions in DNAJC9-depleted cells (Fig. EV5C). These results show that CENP-A mislocalization upon DNAJC9 depletion can occur both in transformed (HeLa) and non-transformed (hTERT-RPE1) cells.

To identify the sites of CENP-A mislocalization, we performed CUT&RUN (Cleavage Under Targets & Release Using Nuclease) sequencing in control and DNAJC9-depleted GFP-CENP-A expressing RPE1 cells. Depletion of DNAJC9 led to a higher enrichment of noncentromeric CENP-A peaks across all the chromosomes (Fig. 8A), and this was observed in two biological repeats when compared to siNeg control cells (Figs. 8B and EV5D). Our analysis pipeline identified 17,341 significantly enriched CENP-A peaks in DNAJC9-depleted cells common to both biological repeats when compared to siNeg control (Fig. EV5E). These results are consistent with our data showing mislocalization (Fig. 3A,B) and accumulation of CENP-A on chromatin (Fig. 4D) in DNAJC9-depleted cells. This accumulation of noncentromeric CENP-A was observed at genic and intergenic regions (Fig. 8C) associated with active and repressive marks (Fig. 8D), with a bias towards regions that are transcriptionally active (marked by H3K27ac, H3K9ac, and H3K4me3) and more accessible in ATAC-seq datasets (Zhang et al, 2023) (Fig. 8E). In summary, our data shows that loss of DNAJC9 function leads to genome-wide mislocalization of CENP-A in chromosomally stable near diploid RPE1 cells.

## Discussion

Deposition of the histone H3 variant CENP-A at the centromere is essential to maintain the integrity of centromeric chromatin and faithful chromosome segregation. The mislocalization of over-expressed CENP-A to the noncentromeric regions contributes to CIN, as demonstrated in studies from budding yeast, fission yeast, fruit fly, and human cells (Au et al, 2008; Gonzalez et al, 2014; Heun et al, 2006; Shrestha et al, 2017). Despite these observations, proteins that promote or prevent the mislocalization of CENP-A remain largely undefined. The genome-wide RNAi screen presented in this study provides the first comprehensive identification of gene depletions that alter nuclear CENP-A levels and thereby the

localization of CENP-A. Among gene depletions that increase nuclear CENP-A intensity, the RNAi screen identified multiple candidates that function in regulating histone H3 deposition (CHAF1A, CHAF1B, EP400, TRRAP, NASP, DNAJC9) (Cook et al, 2011; Hammond et al, 2021; Piette et al, 2021; Pradhan et al, 2016; Shibahara and Stillman, 1999), suggesting that maintaining H3–H4 homeostasis is critical in preventing CENP-A mislocalization. Here, we defined a novel role for the heat-shock protein and H3–H4 co-chaperone DNAJC9 in preventing the mislocalization of CENP-A and CIN. Analysis of CENP-A interactome in DNAJC9-depleted cells highlighted a role for the histone chaperone MCM2 in promoting CENP-A mislocalization. We provide mechanistic insights into the modes by which CENP-A is mislocalized and show that the catalytic activity of DNAJC9 is required for preventing CENP-A mislocalization. We propose that the integrity of the H3–H4 supply chain regulated by histone chaperones such as DNAJC9 prevents the promiscuous entry of CENP-A into aberrant deposition pathways, and CIN.

In-depth studies were pursued with DNAJC9, one of the lead candidates of the screen, which regulates the folding and supply of H3–H4 (Hammond et al, 2021). We observed increased nuclear CENP-A intensity in DNAJC9-depleted cells at all stages of the cell cycle. In addition, DNAJC9-depleted cells exhibit CENP-C mislocalization to noncentromeric regions, reduced NUF2 levels at the kinetochore, and CIN phenotypes characterized by micronuclei, anaphase bridges, and lagging chromosomes. Together, these results show that defects in native kinetochore structure may contribute to CIN upon DNAJC9 depletion. Previous studies showed that HeLa cells expressing dominant negative DNAJC9 mutants exhibit mitotic defects, including multipolar spindle formation and chromosome missegregation (Piette et al, 2021); however, the mechanistic basis for these phenotypes had not been defined. In summary, these findings show that proper DNAJC9 functionality prevents the mislocalization of CENP-A and CIN.

Using interactome analysis, we provide mechanistic insights into the modes by which CENP-A is mislocalized in DNAJC9-depleted cells. Our data revealed a strong enrichment of chaperones involved in DNA-replication-coupled nucleosome assembly such as TONSL and MCM2 with CENP-A upon DNAJC9 depletion. MCM2 chaperones new soluble histones (Groth et al, 2007; Huang et al, 2015; Jasencakova et al, 2010), and plays a role in recycling parental histones at the replication fork (Gan et al, 2018; Petryk et al, 2018). Meanwhile, TONSL acts as a reader of new histone H4 tails during histone supply (Saredi et al, 2016). MCM2 and TONSL interact in a histone-dependent manner (Hammond et al, 2021; Saredi et al, 2016) and are histone co-chaperone partners with DNAJC9 (Hammond et al, 2021). Interestingly, we observe a stronger enrichment of these factors with soluble CENP-A compared to chromatin-bound CENP-A upon depletion of DNAJC9. MCM2 can also bind CENP-A–H4 in a co-chaperone complex with HJURP (Huang et al, 2015) and part of this function is to promote the reassembly of centromeric CENP-A in the wake of DNA replication (Zasadzinska et al, 2018). Our data showed that co-depletion of MCM2 with DNAJC9 suppressed the mislocalization of CENP-A. Therefore, whilst it is plausible that MCM2 aids in recycling noncentromeric CENP-A, our data supports that loss in the fidelity of new histone H3–H4 supply pathways is a major contributing factor to CENP-A mislocalization upon DNAJC9 depletion. This concept is strengthened by our findings of increased CENP-A

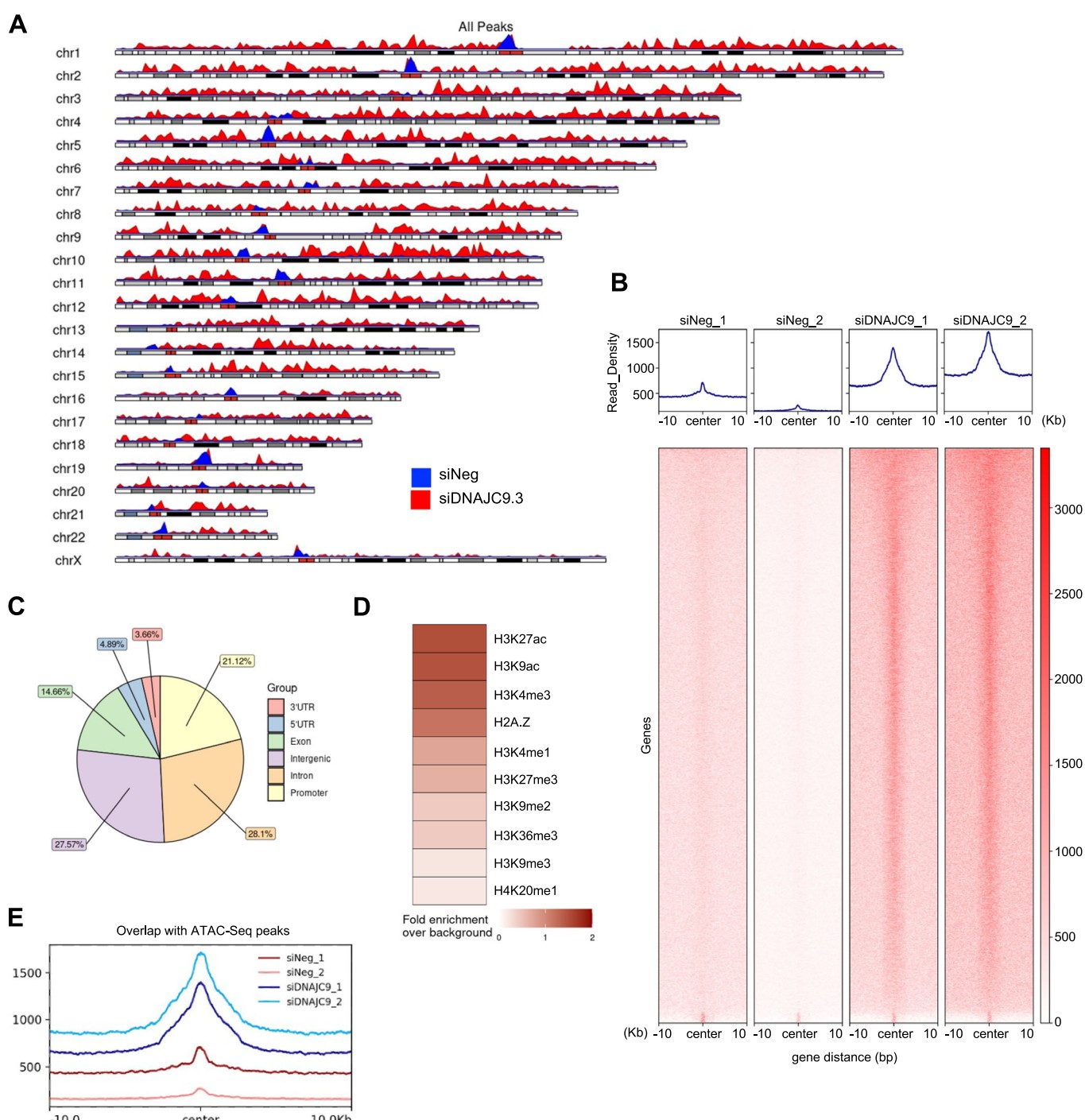

**Figure 8. Genome-wide CUT&RUN analysis shows mislocalization of CENP-A to noncentromeric regions of DNAJC9-depleted RPE1 cells.**

(**A**) Karyoplot of CENP-A CUT&RUN peak density, representing the distribution of the CENP-A identified in control siNeg (blue track) and siDNAJC9 (red track) treated RPE1-Tet-GFP-CENP-A cells. Plot represents the average from two biological replicates for each condition. (**B**) Heatmap of genes (N = 9253) associated with peaks identified by CUT&RUN sequencing of CENP-A in control (siNeg) and DNAJC9-depleted cells in DOX-treated RPE1-Tet-GFP-CENP-A cells in two biological replicates. Read density (z-score normalized) for each condition is shown above the heatmap and centered on the midpoint of the CENP-A peaks identified in DNAJC9-depleted cells. (**C**) Pie chart representing the distribution of the 17,341 significant common peaks identified in DNAJC9-depleted CENP-A CUT&RUN datasets across the indicated genomic features. (**D**) Heatmap showing the fold change of significantly enriched CENP-A CUT&RUN peaks in DNAJC9-depleted cells compared to control cells (siNeg), calculated for genomic loci associated with various epigenetic marks. (**E**) Read density plot showing CENP-A CUT&RUN peaks identified in indicated conditions that overlap with peaks from ATAC-Seq experiments published previously (GSM6383021, (Zhang et al, 2023)). Plots are centered on the midpoint of the peaks identified. Source data are available online for this figure.

mislocalization in cells expressing the catalytically inactive DNAJC9 J mutant. The J mutant can bind H3–H4 but cannot recruit and stimulate HSP70 to catalyze proper folding, resulting in the accumulation of H3 and H4 as non-productive histone supply intermediates in the soluble fraction (Hammond et al, 2021). Thus, the expression of DNAJC9 J mutant likely favors the deposition of CENP-A to the noncentromeric chromatin by reducing the pool of properly folded H3–H4 and thereby increasing H3–H4 histone chaperone availability. Furthermore, CENP-A mislocalization observed in the context of a partial depletion of H3.3 phenocopied loss of DNAJC9 functionality. Together this demonstrates that H3–H4 supply defects promote CENP-A mislocalization and thus loss of fidelity of histone supply chains drives CIN, which is potentially linked to tumorigenesis.

Similar to the ability of CENP-A to enter DNA-replication-coupled nucleosome assembly pathways upon DNAJC9 loss, we also observed enhanced association of CENP-A with pathways linked to heterochromatin formation and maintenance (e.g., DAXX-ATRX and the NuRD complex), transcription (e.g., SPT2 and the FACT complex), DNA damage signaling/repair (TONSL and PARP1) and, DNA topology (TOP2A and TOP2B). Of these factors, DAXX has been previously implicated in promoting CENP-A mislocalization (Lacoste et al, 2014; Shrestha et al, 2017). Thus, it appears that several routes can facilitate the mislocalization of CENP-A, suggesting that CENP-A deposition is somewhat opportunistic. As we have demonstrated here, CENP-A mislocalization can be influenced by a variety of histone supply chain defects that include reduced supply of H3–H4 and abrogation of DNAJC9-directed protein folding activities.

Mislocalization of overexpressed CENP-A was also observed in RPE1 cells upon DNAJC9 depletion, suggesting that this phenotype is not limited to transformed cells. Loss of p53 function has been reported to upregulate CENP-A levels (Filipescu et al, 2017) and impact the transition of CENP-A overexpressing cells from epithelial to mesenchymal states (Jeffery et al, 2021). Our data demonstrates that DNAJC9 loss is not linked to changes in CENP-A mRNA levels but rather increases the stability of CENP-A on a protein level. This could explain why mislocalization of CENP-A upon DNAJC9 depletion was observed in both p53 defective (HeLa) and p53 active (RPE1) cells. Genome-wide analysis revealed that the CENP-A sites enriched in DNAJC9-depleted RPE1 cells overlap with transcriptionally active and open chromatin regions, as reported in previous studies with HeLa and SW480 cell lines (Athwal et al, 2015; Lacoste et al, 2014; Nechemia-Arbely et al, 2019). These results emphasize that there are multiple pathways for CENP-A mislocalization, and DNAJC9 which binds H3.1/2/3–H4 has a strong influence on histone supply chain equilibria linked to DNA replication and transcription, as established previously (Hammond et al, 2021).

In summary, we have defined novel regulators of CENP-A mislocalization and characterized a role for the histone H3–H4 chaperone functionality of DNAJC9 in preventing CENP-A mislocalization and CIN phenotypes. We propose a model (Fig. 9) showing that CENP-A localization is restricted to the centromeric region under unperturbed H3–H4 supply conditions. Loss of DNAJC9 function induced by siRNA-mediated depletion or expression of the catalytically inactive J mutant leads to mislocalization of CENP-A to noncentromeric regions. The DNA-replication-associated histone chaperone MCM2 promotes ectopic CENP-A deposition in the absence of DNAJC9. The genome-wide mislocalization of CENP-A drives CIN phenotypes marked by micronuclei and mitotic chromosome segregation defects. Our data supports previous studies showing that histone variant supply chains are inter-connected by a network of histone chaperones (Carraro et al, 2023; Hammond et al, 2021; Hammond et al, 2017), and here we highlight DNAJC9 as a key factor that promotes the fidelity of histone specificity in this network. In providing this function, we demonstrate that DNAJC9 prevents the promiscuous entry of CENP-A into H3–H4 deposition pathways to prevent CIN, which is a known hallmark of many cancers (Hanahan, 2022; Hanahan and Weinberg, 2011). Together, our studies highlight how perturbing the equilibria between histone supply chains and histone chaperone networks can have major downstream consequences for chromatin organization.

# Methods

## Cell culture

All cell lines except hTERT-RPE1 mentioned in this study were cultured in DMEM (12491023, ThermoFisher Scientific) supplemented with 10% FCS (Sigma-F6178), penicillin/streptomycin (P/S) (15140122, ThermoFisher Scientific), fungizone (15290018, ThermoFisher Scientific), and L-glutamine (25030081, ThermoFisher Scientific) at 37 °C with 5% $CO_2$ in a humidified incubator. hTERT-RPE1 cells were grown in DMEM/F-12 (11320033, ThermoFisher Scientific) supplemented with 10% FCS (Sigma-F6178), penicillin/streptomycin (15140122, ThermoFisher Scientific), and fungizone (15290018, ThermoFisher Scientific), at 37 °C with 5% $CO_2$. Frozen stocks of cell lines were prepared in culture media containing 50% FCS and 5% DMSO and stored at −80 °C. Cell lines that tested negative for Mycoplasma contamination using a detection kit (30-1012 K, ATCC) were used in this study.

## Transfections and cell line generation

siRNA transfections were performed using Lipofectamine RNAi Max reagent (13778075, ThermoFisher Scientific) following the manufacturer's instructions. All siRNA transfections other than those performed in primary and secondary RNAi screens were conducted for 96 h. For transient transfection of plasmid constructs expressing wild type or mutant DNAJC9, HeLa YFP-CENP-A$^{Low}$ cells were transfected with 800 ng plasmid and cells were analyzed after 72 h of transfection. Lipofectamine 2000 was used for transfection following manufacturer protocols. The plasmid constructs expressing DNAJC9 wild-type, J mutant (H43Q-D45N), 4A (Q224A-R227A-M238A-Y242A) or 4AJ (H43Q-D45N-Q224A-R227A-M238A-Y242A) mutants are described previously (Hammond et al, 2021). To generate the plasmid encoding DOX-inducible siRNA-resistant DNAJC9 (pCE1000), synthesized cDNA was obtained from Twist Biosciences and subsequently cloned into pCW57-MCS1-P2A-MCS2 (Hygro; Addgene 80922) (Barger et al, 2019) using isothermal assembly (Gibson et al, 2009). pCE1000 has been deposited in Addgene (ID: 202548). pCW57-MCS1-P2A-MCS2 (Hygro) was a gift from Adam Karpf (Addgene plasmid # 80922; http://n2t.net/addgene:80922; RRID: Addgene_80922). Stable cell line expressing DOX-inducible siRNA-resistant DNAJC9

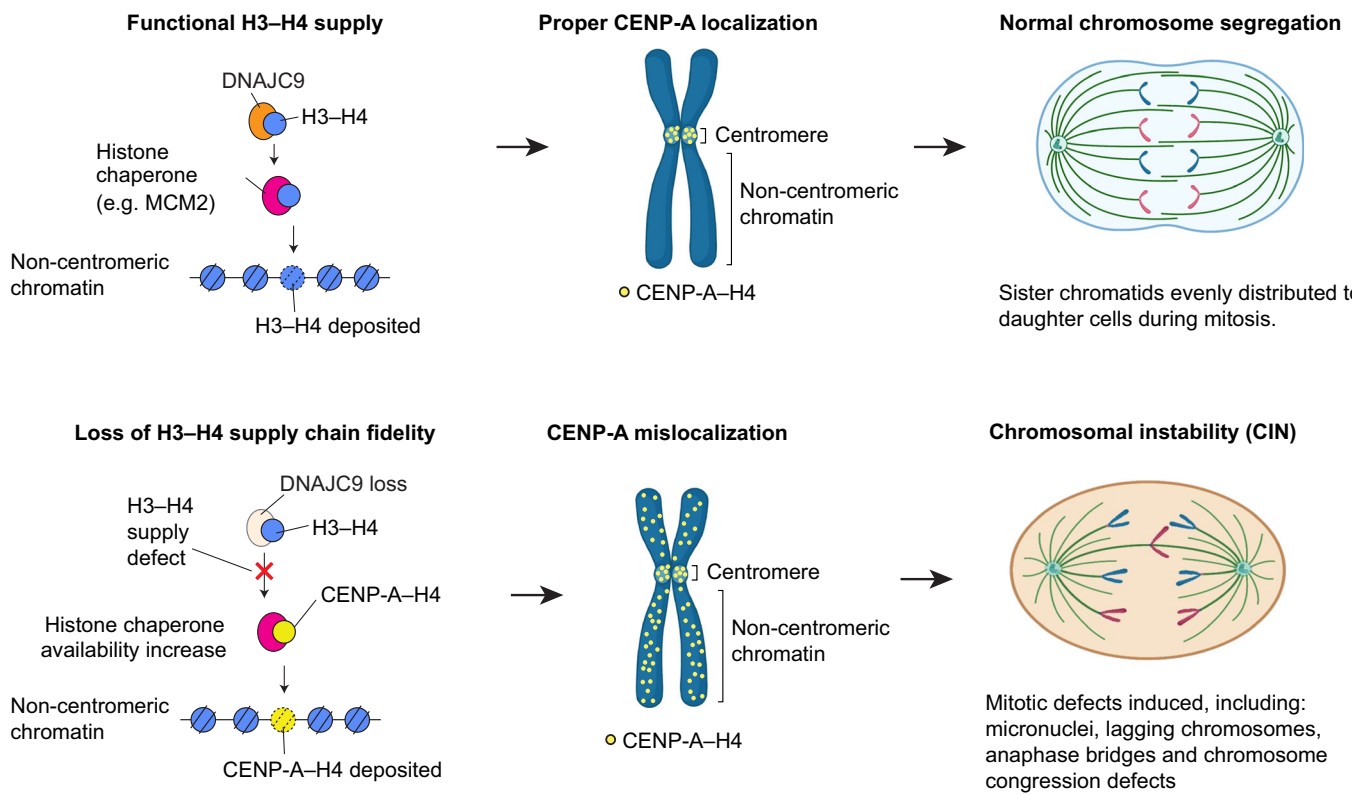

**Figure 9.  Model for role of DNAJC9 in preventing CENP-A mislocalization and chromosomal instability by maintaining the fidelity of H3–H4 supply chains.**

In the context of wild-type DNAJC9 with unperturbed H3–H4 supply, CENP-A localizes to the centromeric region, maintaining normal chromosome segregation status. We propose that loss of DNAJC9 function reduces the supply of H3–H4 (Hammond et al, 2021) and promotes the mislocalization of CENP-A to noncentromeric regions in an MCM2-mediated pathway. Ectopic localization of CENP-A leads to higher incidence of CIN phenotypes. Illustrations created with BioRender.com.

cDNA was constructed by lentiviral transduction of HeLa YFP-CENP-A$^{Low}$. Lentiviral particles were produced in HEK293T cells transfected with pCE1000 using BioT (B01-01, Bioland Scientific) following standard procedures. After 48 h of transfection, viral supernatant was harvested and filtered prior to transduction in HeLa YFP-CENP-A$^{Low}$ cells using polybrene (10 μg/ml). The transduced cells were selected with 200 μg/ml hygromycin after 48 h. Antibiotic-resistant clones were pooled and expanded to screen for expression of siRNA-resistant *DNAJC9* cDNA by immunofluorescence. To induce the expression of the siRNA-resistant *DNAJC9* cDNA, cells were treated with 1 μg/ml DOX (D3072, Sigma Life Science). Stable RPE1-Tet-GFP-CENP-A cell line was constructed by lentiviral transduction with plasmid encoding Tet inducible GFP-CENP-A (pMG0051). pMG0051 was constructed using an isothermal assembly (Gibson et al, 2009). Briefly, Addgene plasmid 83481 was modified by digesting it with NheI and BamHI to remove Cas9 and replace it with a multiple cloning site. This new plasmid was then digested and assembled with synthesized *CENP-A* cDNA (Twist Biosciences) and GFP (amplified from another plasmid). pCW-Cas9-Blast was a gift from Mohan Babu (Addgene plasmid # 83481; http://n2t.net/addgene:83481; RRID: Addgene_83481). Lentiviral packaging was done in HEK293T cells and viral pool post 48 h transfection was collected and filtered. Parental RPE1-hTERT cell line from ATCC was transduced with the viral particles and selected with 10 μg/ml

blasticidin for 10 days. Polyclonal populations resistant to blasticidin were sorted for single cells using Flow cytometry into a 96-well plate. A single-cell clone expressing GFP-CENP-A with minimal noncentromeric signal upon induction with 1 μg/ml DOX for 48 h was selected.

## Primary siRNA screen and secondary validation

The primary RNAi screen was conducted using the Ambion Silencer® Select Human Genome siRNA Library, which consists of three unique, non-overlapping, non-pooled siRNAs per gene target. siRNA reagents (2 μl of a 400 nM stock) were stamped into 384-well microplates (black, clear bottom PhenoPlate™ 384-well microplates, PerkinElmer, USA) using a Velocity11 vPrep liquid handling system (Agilent, USA) integrated into a BioCel robotic platform (Agilent) in columns 1–22, leaving columns 23–24 empty for biological positive control (Silencer® Select CHAF1B siRNA, ThermoFisher Scientific), transfection positive control (AllStars Hs Cell Death Control siRNA, Qiagen), and transfection negative control (Silencer® Select negative control siRNA, ThermoFisher Scientific). RNAiMAX (0.2 μL; Invitrogen) was added in 20 μL screening media to wells using a Matrix WellMate and Microplate Stacker (ThermoFisher Scientific). Plates were incubated for 45 min at room temperature to allow for the formation of siRNA-lipid complexes. Cells were seeded at a density of 1500 cells/well in 20 μL

screening media and cultured for 3 days at 37 °C, 5% $CO_2$, and 95% humidity. Then, the cells were fixed with 4% paraformaldehyde and imaged with Opera Phenix Plus High-Content Screening System (PerkinElmer, USA). Quality control using Z' was applied to exclude plates with poor performance. All plates were then subject to visual inspection to further remove technical errors. YFP median nucleus intensity per well was normalized to negative control siRNA. The gross off-targeting of a siRNA was computed as the weighted sum of all off-target effects that the given siRNA could potentially be associated with. Then this gross off-target contribution was subtracted from the experimental Z-score to form the on-target Z-score. The gene-level Z-scores were calculated using the median corrected on-target Z-score. We considered those with gene-level Z-score < -2 or > 2 as the potential candidates. For the follow-up validation, 199 gene candidates from the primary screen were chosen to test against three more independent siRNAs (Dharmacon ON-TARGETplus). The activity of all 6 siRNAs of each candidate from both primary and follow-up screens were analyzed with the above method of primary RNAi screen.

## Flow cytometric analysis of cell cycle

Cells were harvested and washed with PBS before fixing in ice-cold 70% ethanol overnight at 4 °C. Following ice-cold PBS washes, cells were treated with RNase (0.1 mg/ml) for 20 min, stained with propidium iodide (10 μg/ml) in dark for 15 min, and analyzed on a BD FACSymphony A5. Raw files (.fcs) were processed using FlowJo (10.8.1) for gating cells and analyzing cell cycle profiles.

## Immunostaining and immunoblotting

Immunostaining and immunoblotting protocols were followed as previously described (Shrestha et al, 2023). Briefly, cells grown on coverslips were washed with PBS and fixed with ice-cold methanol for 1 min followed by three PBS washes. Fixed cells were blocked in 1% BSA prepared in PBS + 0.1% Tween-20 for 1 h at room temperature. Primary and secondary antibody incubations were performed for 1 h at room temperature and stained with DAPI. Coverslips were washed thrice with PBS + 0.1% Tween-20, followed by one wash with water and mounted on glass slides using mounting solution (ProLong Gold Antifade, Invitrogen). The following primary antibodies were used in 1:500 dilutions: mouse anti-CENP-A (ADI-KAM-CC006-E, Enzo Life Sciences), guinea pig anti-CENP-C (PD030, MBL Life Science), rabbit anti-GFP (ab290, Abcam), rabbit anti-NUF2 (ab122962, Abcam), rabbit anti-FLAG (14793 S, Cell Signaling Technology), rabbit anti-DNAJC9 (ab150394, Abcam). The secondary antibodies used: goat anti-mouse DyLight 488, goat anti-rabbit DyLight 594, goat anti-rabbit DyLight 488 and goat anti-guinea pig DyLight 594 (ThermoFisher Scientific). For immunoblotting, whole-cell extracts were prepared in Laemmli buffer and run on 4–12% Bis-tris gels (Invitrogen). The blots were probed with the following primary antibodies: mouse anti-CENP-A at 1:500 (ADI-KAM-CC006-E, ENZO), rabbit anti-H2B (07-371, Millipore Sigma) at 1:2000, rabbit anti-DNAJC9 at 1:1000 (ab150394, Abcam), rabbit anti-MCM2 at 1:1000 (4007 S, Cell Signaling Technology), rabbit anti-H3.3 at 1:1000 (ab176840, Abcam), rabbit anti-CAF1B/p60 at 1:1000 (Quivy et al, 2004), mouse anti-αTubulin at 1:5000 (T9026, Sigma), rabbit anti-HA at 1:2000 (C29F4, Cell signaling technology), rabbit anti-H4 at 1:2000

(Millipore, 05-858), and rabbit anti-αTubulin at 1:2000 (ab176560, Abcam). The secondary antibodies used: sheep anti-mouse HRP conjugated and donkey anti-rabbit HRP conjugated (GE Healthcare), goat anti-mouse HRP (115-035-068, Jackson ImmunoResearch Europe Ltd) and donkey anti-rabbit HRP (711-035-152, Jackson ImmunoResearch Europe Ltd.), goat anti-rabbit IgG StarBright Blue 700 and Goat anti-Mouse IgG StarBright Blue 520 (Bio-Rad) at 1:5000–20,000 dilution. SuperSignal West Pico PLUS or Pierce™ ECL Chemiluminescent substrate (34578 and 32106, ThermoFisher Scientific) was used to image blots, and the signal intensities of bands from western blots were quantified using ImageJ software (Schneider et al, 2012).

## Microscopy and image analysis

Images were acquired on DeltaVision Elite imaging system (GE Healthcare, USA) with a CoolSNAP charge-coupled device camera mounted on an Olympus IX-70 inverted microscope. Exposure times set for different filters were: 0.005 s for DAPI and 0.1 s for mCherry and FITC. For nuclear intensity measurements of CENP-A, maximum intensity projection of each image was generated using *softWoRx* and exported as 24-bit RGB TIF file for analysis using *Fiji*. Interphase nuclei were defined based on DAPI staining; mitotic cells were removed and cells on the edge were filtered out. Mean nuclear intensities of CENP-A were measured for the regions of interest marked by DAPI signal. Quantitative analysis of CENP-A, CENP-C, and NUF2 levels at the centromeric or noncentromeric regions were performed using the data inspector tool of *Softworx* as described previously (Shrestha et al, 2021).

## Immunofluorescence microscopy to analyze CENP-A levels in different cell cycle stages

Cells grown on coverslips were treated with 10 μM EdU (Click-&-Go™ EdU 594 Cell Proliferation Assay Kit, Click Chemistry Tools) for 20 min. Cells were washed twice with PBS and fixed with ice-cold methanol for 1 min. Labeling was performed according to the standard manufacturer's protocol. After EdU staining, cells were blocked for 30 min in 1% BSA/PBST followed by staining for CENP-A and DAPI as described above. Cells were imaged with 60X NA 1.42 oil immersion objective on DeltaVision Core system (Applied Precision/GE Healthcare, Issaquah, WA) and Z stack images of 15 sections of step-size 0.2 μm were acquired. Exposure times set for different channels were: 0.005 s for DAPI and 0.1 s for mCherry and FITC. Nuclear masking was performed as described above. Mean nuclear intensities of CENP-A and EdU signals, and the integrated density of DAPI signal were measured for the assigned region of interest based on DAPI mask. G1, S, and G2 populations were identified according to the relative intensities of DAPI and EdU signals as shown in Appendix Fig. S3. Mean CENP-A intensities for each of these populations were extracted for siNeg and siDNAJC9.3-transfected cells and plotted in GraphPad Prism.

## Metaphase chromosome spread and immunofluorescence

Chromosome spreads were done as previously described (Shrestha et al, 2023). Briefly, cells were treated with 0.1 μg/ml Colcemide (15212012, Gibco Life Technologies) and harvested after 4 h. Cell

pellets were resuspended in hypotonic solution (75 mM potassium chloride) and incubated at 37 °C for 15 min. Around 25,000 cells were cytospun onto glass slides and air-dried before storing at 4 °C overnight. Cells were rehydrated with KCM buffer (10 mM Tris-HCl pH 8.0, 120 mM KCl, 20 mM NaCl, 0.5 mM EDTA) containing 0.1% Triton X-100 for 2 min followed by permeabilization using KCM buffer containing 0.5% Triton X-100. For chromosome spreads performed under high salt concentrations, cells were treated with permeabilization buffer (KCM buffer with 0.5% Triton X-100) containing 500 mM NaCl for 20 min. After blocking the cells with 1% BSA in KCM (0.1% Triton X-100) for 30 min at room temperature, immunostaining was performed with primary antibodies (1:500 dilution) followed by secondary antibody incubations, each for 30 min. Spreads were treated with 3.7% formaldehyde solution prepared in KCM for 15 min, washed with PBS followed by DAPI staining and mounting (ProLong Gold Antifade, Invitrogen).

## RT-qPCR

Total RNA was isolated from cell pellets using TRIzol reagent and purified with RNeasy Mini kit (Qiagen). Quality and quantity of the purified RNA was assessed using NanoDrop spectrophotometer and around 0.5–1 μg total RNA per sample was used for cDNA synthesis (Superscript III Reverse transcriptase, Invitrogen). For quantitative PCR (qPCR), equal amount of cDNA was analyzed with Fast SYBR Green Master Mix (Thermo Scientific). Each sample was run in technical duplicates. Transcript levels of *CENP-A*, *MCM2* and *DNAJC9* were assessed by normalizing against *GAPDH* using $2^{-\Delta Ct}$ for each biological replicate. The sequences of the qPCR primers are listed in Dataset EV3.

## Cycloheximide assay

Cells transfected with siNegative or siDNAJC9.3 oligos were treated with 100 μg/ml cycloheximide (C1988, Sigma-Aldrich) for 0, 3, 6, 9, and 12 h. Whole-cell extracts were prepared after the designated time points in Laemmli buffer and analyzed by immunoblotting.

## Histone extraction

Cells were processed as described previously (Shrestha et al, 2023). In brief, freshly harvested cells were pelleted and treated with Triton X-100 extraction buffer (0.5% Triton X-100 (v/v), 2 mM phenylmethylsulfonyl fluoride (PMSF), 0.02% (w/v) NaN₃) for 10 min on ice to extract soluble fraction. The cell pellets were washed in the extraction buffer and treated with 0.2 N HCl overnight at 4 °C. The supernatant from the acid-treated samples was saved as chromatin fraction and neutralized with 2 M NaOH. Protein concentration was determined using DC (detergent compatible) protein assay (Bio-Rad).

## CUT&RUN sequencing analysis

CUT&RUN assay was performed using CUTANA™ ChIC/CUT&RUN Kit v3 (SKU:14-1048, EpiCypher) following the manufacturer's protocol (Manual v3.3). Briefly, 500,000 siRNA-transfected RPE1-Tet-GFP-CENP-A cells were harvested and

incubated with activated Concanavalin A beads. The bead-cell slurry was incubated with mouse anti-CENP-A antibody (MA1-20832, Invitrogen) or mouse IgG control (sc-2025, Santa Cruz Biotechnology) at 1:50 dilution in antibody buffer overnight at 4 °C. The beads were washed twice with ice-cold cell permeabilization buffer followed by pAG-MNase digestion at room temperature for 10 min. Activation of pAG-MNase was done by incubating the beads with calcium chloride-containing cell permeabilization buffer for 2 h at 4 °C. The reaction was halted by the addition of stop buffer containing 50 pg/μl *E. coli* Spike-in DNA and the chromatin fragments were released by incubation at 37 °C for 10 min. The DNA was purified using the DNA cleanup columns provided with the kit. Libraries were prepared using the Accel-NGS® 2 S Plus DNA Library Kit (21024/21096) and size selection was performed to exclude fragments greater than 500 bp. Paired-end sequencing of libraries was done on Illumina NextSeq 2000 with total reads per sample ranging from 25 to 30 million.

A custom pipeline was utilized for data analysis: CARLISLE (version 2.3.0; https://doi.org/10.5281/zenodo.10483877). Briefly, raw fastq files were trimmed with Cutadapt (version 4.0) (Martin, 2011) and aligned using bowtie2 (version 2-2.4.5) (Langmead and Salzberg, 2012) against T2T-CHM13v2.0 (https://www.ncbi.nlm.nih.gov/datasets/genome/GCF_009914755.1/). For spike-in controls, the trimmed FASTQ files were aligned against the *Escherichia coli* MG1655 genome. Duplicated reads were removed using Picard tools (version 2.27.3) (http://broadinstitute.github.io/picard/) and custom Python (version 3.9) scripts, and normalization factors were derived based on the uniquely mapped fragments in the corresponding spike-in control data. The enriched regions were identified using GoPeaks (Yashar et al, 2022) spike-in normalization, with default "--broad" parameters applied. Differential analysis was performed using R (version 4.2.2) package DESeq2 (Love et al, 2014) to obtain a final list of regions for downstream analysis (*P* value 0.05, Log2 FC 1.1). For annotation of the genomic features described in the pie chart (Fig. 8C), each peak/enriched region in CUT&RUN assays was annotated with the nearby gene displaying the shortest distance between TSS and the center of each peak using R (version 4.2.2) package ChipSeeker (Wang et al, 2022; Yu et al, 2015). Genome browser track and genomic coordinate heatmaps were obtained using deepTools (version 3.5.1) (Ramirez et al, 2016). Enrichment analysis (Fig. 8D) was performed using ChIP-Atlas (https://chip-atlas.org/), sub-setting for ChIP experiments in "neural" cell types and hTERT-RPE1 cells. Significantly differentiated peaks were used as the input and results were further sub-set for the following publicly available ChIP-Seq datasets from RPE1 cells: H3K27me3 (GSM5576209), H3K36me3 (GSM5576210), H3K4me3 (GSM5576211), H3K9ac (GSM5576212), H3K9me3 (GSM5576213), H4K20me1 (GSM5576214) (Rang et al, 2022), H2A.Z (GSM6429697), H3K27ac (GSM6429695), H3K4me1 (GSM6429704) (Friskes et al, 2022), and H3K9me2 (GSM5975181).

## Global interactome analysis of soluble and chromatin-bound CENP-A

Cells expressing CENP-A-Flag-HA were created by lentiviral transduction of HeLa S3 suspension cells as described previously (Hammond et al, 2021). Cells were grown in a humidified incubator at 37 °C with 5% CO₂, in DMEM + GlutaMax with 10% FBS and 1% P/S prior to transfection. Cells resuspended in DMEM + GlutaMax with 5% FBS

(15 × 10⁶ cells in 18.8 ml) were transfected with Lipofectamine™ RNAiMAX (0.1 ml) and Silencer® Select siRNAs (0.1 ml, 50 μM) each diluted in Opti-MEM™ (3 ml) resulting in a total volume of 25 ml. After 24 h, transfected cell suspensions were expanded to 40 ml with DMEM + GlutaMax with 10% FBS and 1% P/S. With the same media, cell suspensions were expanded to 70 ml after 48 h, and CENP-A-Flag-HA expression from the integrated pTetOne puro plasmid promoter was induced with Doxycycline (+DOX, 250 ng/ml), unless otherwise stated. Similarly, cell suspensions were expanded to 160 ml after 72 h, and DOX was replenished (250 ng/ml) for induced conditions. Cells were harvested after 96 h by centrifugation, washed in PBS (50 ml, 37 °C), resuspended in PBS (1 ml, 37 °C), isolated by centrifugation, aspirated, and stored at −80 °C. This process was repeated for five biological replicates in the following conditions: (1) "siDNAJC9_-CENP-A" + DOX; (2) "siCAF1B_CENP-A" + DOX; (3) "siCTRL_-CENP-A" + DOX; (4) "siCTRL_X" without DOX. The sequence and catalog numbers for the siRNA oligos used are listed in Dataset EV3. For DNAJC9 depletion, siDNAJC9.3 was used. Cell pellets were extracted on ice for 25 min with chromatin wash buffer (ChWB: 300 mM NaCl, 0.5% Nonidet P40, 50 mM HEPES pH 7.9, 0.4 mM EDTA, 5% glycerol, 1/100 NaF and β-GP, 1/1000 PMSF, leupeptin, pepstatin, 1/10,000 TSA, 1/500 Na₃VO₄) and soluble extracts were isolated from chromatin pellets by centrifugation. Chromatin digestion buffer (DB) was prepared by supplementing ChWB (49 ml) with 10 mM CaCl₂ (1 ml, 0.5 M), and supplementing 30 ml of the resultant dilution buffer with MNase (150 μl, 763 U/μl). Chromatin pellets were digested in DB with mixing for 1.5 h (1250 rpm, 37 °C). After centrifugation (5 mins, 13,100 rpm, 4 °C), soluble and chromatin extracts were filtered (0.45 μm). Protein concentrations were equalized with their equivalent final extraction buffer, and extracts were incubated overnight at 4 °C with anti-HA beads (88836, Thermo Scientific). Beads were extensively washed with ice-cold buffers, three times with ChWB, three times with minimal wash buffer (MWB: 300 mM NaCl, 50 mM HEPES pH 7.9) and one time with 50 mM NH₄HCO₃. Samples were digested using sequencing-grade modified trypsin on-beads in 50 mM NH₄HCO₃ and spin-filtered (0.45 μm). The resulting peptide-containing samples were desalted and purified with homemade StageTips at high-pH as described previously (Hammond et al, 2021).

MS samples were analyzed on an EASY-nLC 1200 system (Thermo) coupled to an Orbitrap Exploris™ 480 mass spectrometer (Thermo). Separation of peptides was performed using 20-cm columns (75 μm internal diameter) packed in-house with ReproSil-Pur 120 C18-AQ 1.9 μm beads (Dr. Maisch). Elution of peptides from the column was achieved using a gradient ranging from buffer A (0.1% formic acid) to buffer B (80% acetonitrile in 0.1% formic acid), at a flow of 250 nl/min. The gradient length was 80 min per sample, including ramp-up and wash-out, with an analytical gradient of 60 min ranging from 5% B to 32% B. Analytical columns were heated to 40 °C using a column oven, and ionization was achieved using a NanoSpray Flex™ NG ion source. Spray voltage was set to 2 kV, ion transfer tube temperature to 275 °C, and RF funnel level to 40%. Full scan range was set to 300–1300 *m/z*, MS1 resolution to 120,000, MS1 AGC target to "200" (2,000,000 charges), and MS1 maximum injection time to "Auto". Precursors with charges 2-6 were selected for fragmentation using an isolation width of 1.3 *m/z*, and fragmented using higher-energy collision disassociation (HCD) with normalized collision energy of 25. Precursors were excluded from re-sequencing by setting a dynamic exclusion of 80 s. MS2 resolution was set to 45,000, MS2 AGC target to "200" (200,000 charges), intensity threshold to 230,000 charges per second, MS2 maximum injection time to "Auto", and number of dependent scans (TopN) to 9.

## MS data analysis

All RAW files were analyzed using MaxQuant software (version 1.6.3.4). The human FASTA database used in this study was downloaded from UniProt on April 29, 2022. Default MaxQuant settings were used, with exceptions specified below. Label-free quantification (LFQ) was enabled, with "Skip normalization" checked. Matching between runs (MBR) and iBAQ quantification were enabled. All data were filtered by posterior error probability to achieve a false discovery rate of <1% (default), at both the peptide-spectrum match and the protein assignment levels. Statistical handling of data is detailed in relevant figures and legends. MaxQuant text output (proteinGroups.txt) was further analyzed using the freely available Perseus software (Tyanova et al, 2016), version 1.6.2.3. As the knockdown of DNAJC9 in biological replicate two failed, the datasets "siDNAJC9_CENP-A_Sol_02" and "siDNAJC9_CENP-A_Chr_02" were filtered out during the analysis. In addition, the proteomics data was filtered to exclude contaminant hits, reverse-database hits, and proteins only identified by site, Log2 transformed, and filtered for detection in 100% of biological replicates for at least one condition, excluding the uninduced/beads-only control condition. Where stated, missing values were imputed with a downshift of 1.8 and a width of 0.3. Student's two-sample *t* testing was performed with permutation-based FDR control, with s0 values stated for each experiment, to derive *P* values corrected for multiple-hypothesis testing (i.e., *q*-values). GO-analysis was performed using the String-db "Proteins with Values/Ranks–- Functional Enrichment Analysis". Graphs were visualized in GraphPad Prism v9.0.0. GO-term clustering, bubble plots and Venn diagrams were visualized with R version 4.2.2 using the libraries rColorBrewer v1.1–3, ggplot2 version v3.4.2, scales v1.2.1, plyr v1.8.8, dplyr v1.1.2, tidyverse v2.0.0, and eulerr v7.0.0.

## Data availability

The mass spectrometry proteomics data have been deposited to the ProteomeXchange Consortium via the PRIDE (Perez-Riverol et al, 2022) partner repository with the dataset identifier PXD043961 and processed data is presented in Dataset EV4. The sequencing data for CENP-A CUT&RUN is deposited in NCBI Gene Expression Omnibus (GEO) with accession ID GSE253387. Source data for the Western blot and immunofluorescence images included in Expanded View and Appendix Figures is available on Figshare (https://doi.org/10.6084/m9.figshare.25436131).

## Peer review information

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

## Acknowledgements

The authors thank Bao Tran and Jyoti Shetty (NCI CCR Sequencing Facility, Frederick National Laboratory for Cancer Research) for CUT&RUN Sequencing, the NCI CCR Genomics Core for Tape station analysis of CUT&RUN samples, Vassiliki Saloura and Sohyoung Kim (CCR/NCI/NIH) for expert advice with CUT&RUN experiments. We also thank Tom Misteli for critical reading and comments on the manuscript, Gianluca Pegoraro and Laurent Ozbun of the High-Throughout Imaging Facility (CCR/NCI/NIH), for optimization of the high-throughput imaging assays and pilot screens, and members of the Basrai laboratory for discussions and comments on the manuscript. This study utilized the high-performance computational capabilities of the Biowulf Linux cluster at the National Institutes of Health, Bethesda, MD (https://hpc.nih.gov). This work was supported by the Intramural Research Program of the National Cancer Institute (NCI), Center for Cancer Research (CCR): project number ZIA BC 010822 (MAB), project number ZIA BC 011704 (NJC), and 75N91019D00024 (VK). The work in Ken Chih-Chien Cheng's laboratory was supported by the NIH Intramural Research Program (NIH IRP) and the Intramural Research Program of the National Center for Advancing Translational Sciences (NCATS). RC was supported by Federal funds from the National Cancer Institute, National Institutes of Health, under Contract No. HHSN261201500003I. MLN is supported by the Independent Research Fund Denmark (0135-00096B and 8020-00220B), the European Union's Horizon 2020 research and innovation program under grant agreement EPIC-XS-823839, and the Danish Cancer Society (R146-A9159-16-S2). AG's laboratory was supported by the European Research Council (ERC CoG 724436), the Lundbeck Foundation (R313-2019-448) and the Novo Nordisk Foundation (NNF21OC0067425). Research at CPR is supported by the Novo Nordisk Foundation (NNF14CC0001). The content of this publication does not necessarily reflect the views or policies of the U.S. Department of Health and Human Services, nor does mention of trade names, commercial products, or organizations imply endorsement by the U.S. Government.

## Author contributions

**Vinutha Balachandra**: Conceptualization; Formal analysis; Validation; Investigation; Methodology; Writing—original draft; Writing—review and editing. **Roshan L Shrestha**: Conceptualization; Investigation; Methodology; Writing—review and editing. **Colin M Hammond**: Conceptualization; Resources; Formal analysis; Investigation; Methodology; Writing—original draft; Writing—review and editing. **Shinjen Lin**: Formal analysis; Investigation; Methodology. **Ivo A Hendriks**: Formal analysis; Investigation; Writing—review and editing. **Subhash Chandra Sethi**: Investigation; Writing—review and editing. **Lu Chen**: Formal analysis. **Samantha Sevilla**: Formal analysis; Investigation; Writing—review and editing. **Natasha J Caplen**: Conceptualization; Funding acquisition; Writing—review and editing. **Raj Chari**: Resources; Funding acquisition; Methodology; Writing—review and editing. **Tatiana S Karpova**: Resources; Writing—review and editing. **Katherine McKinnon**: Resources; Writing—review and editing. **Matthew AM Todd**: Investigation. **Vishal Koparde**: Formal analysis; Investigation; Writing—review and editing. **Ken Chih-Chien Cheng**: Resources; Methodology; Writing—review and editing. **Michael L Nielsen**: Conceptualization; Resources; Supervision; Funding acquisition; Writing—review and editing. **Anja Groth**: Conceptualization; Resources; Supervision; Funding acquisition; Writing—review and editing. **Munira A Basrai**: Conceptualization; Supervision; Funding acquisition; Validation; Writing—original draft; Project administration; Writing—review and editing.

## Funding

## Disclosure and competing interests statement

CMH and AG are inventors on a filed patent application covering the therapeutic targeting of TONSL for cancer therapy. AG is a co-founder and chief scientific officer (CSO) of Ankrin Therapeutics. The remaining authors declare no competing interests.

# Expanded View Figures

**Figure EV1.   DNAJC9 depletion contributes to mislocalization of CENP-A in HeLa parental and HeLa YFP-CENP-A^High cells.**                                                                                               ▶

(**A**) Exogenous YFP-CENP-A is stably localized to noncentromeric regions in DNAJC9-depleted HeLa YFP-CENP-A^Low cells. Representative images of metaphase chromosome spreads prepared from cells extracted with 0.5 M NaCl prior to fixation. Immunostaining was done using anti-GFP and anti-CENP-C antibodies. Scale bar: 10 μm. Scale bar for insets: 1 μm. (**B**) Scatter plots showing YFP-CENP-A intensities corrected for background at the centromeric or noncentromeric regions as described in A, from three biological replicates. Median values are shown above graph. Each circle represents the value from an individual chromosome. Chrs refers to the total number of chromosomes measured per condition from three biological replicates. Mean with standard deviation is shown across all measurements from three biological replicates. *P* values were calculated using Mann–Whitney *U* test. (**C**) CENP-A is mislocalized in DNAJC9-depleted parental HeLa cells. Representative images of metaphase chromosome spreads immunostained for CENP-A in siNeg or siDNAJC9.3-transfected parental HeLa cells. Scale bar: 5 μm. Scale bar for insets: 1 μm. (**D**) Scatter plots showing CENP-A intensities corrected for background at the centromeric or noncentromeric regions as described in (**C**), from three biological replicates. Median values are shown above graph. Each circle represents the value from an individual chromosome. Chrs refers to the total number of chromosomes measured per condition from three biological replicates. Mean with standard deviation is shown across all measurements from three biological replicates. *P* values were calculated using Mann–Whitney *U* test. (**E**) Incidence of CIN phenotypes are not altered in parental HeLa cells with DNAJC9 depletion. Bar graphs showing incidence of micronuclei and defective chromosome segregation in HeLa parental cells transfected with control (siNeg) or siDNAJC9.3. Mean with SD was plotted from three biological replicates and *P* value was calculated using unpaired *t* test. *N* denotes the total number of cells analyzed per condition. (**F**) CENP-A is mislocalized in DNAJC9-depleted HeLa YFP-CENP-A^High cells. Representative images of metaphase chromosome spreads immunostained for CENP-A in siNeg or siDNAJC9.3-transfected HeLa YFP-CENP-A^High cells. Scale bar: 15 μm. Scale bar for insets: 5 μm. (**G**) Scatter plots showing CENP-A intensities corrected for background at the centromeric or noncentromeric regions as described in (**F**), from three biological replicates. Median intensity is shown above graph. Each circle represents the value from an individual chromosome. Chrs refers to the total number of chromosomes measured per condition from three biological replicates. Mean with standard deviation is shown across all measurements from three biological replicates. *P* values were calculated using Mann–Whitney *U* test. (**H**) Increased incidence of CIN phenotypes in HeLa YFP-CENP-A^High cells. Representative IF images showing defects in chromosome segregation in DNAJC9-depleted HeLa YFP-CENP-A^High cells immunostained for DNAJC9 and CENP-A. Scale bar: 5 μm. Bar graph shows percent cells with defective chromosome segregation in cells transfected with control (siNeg) or siDNAJC9.3. Mean with SD was plotted from three biological replicates and *P* value was calculated using unpaired *t* test. *N* denotes the total number of cells analyzed per condition.

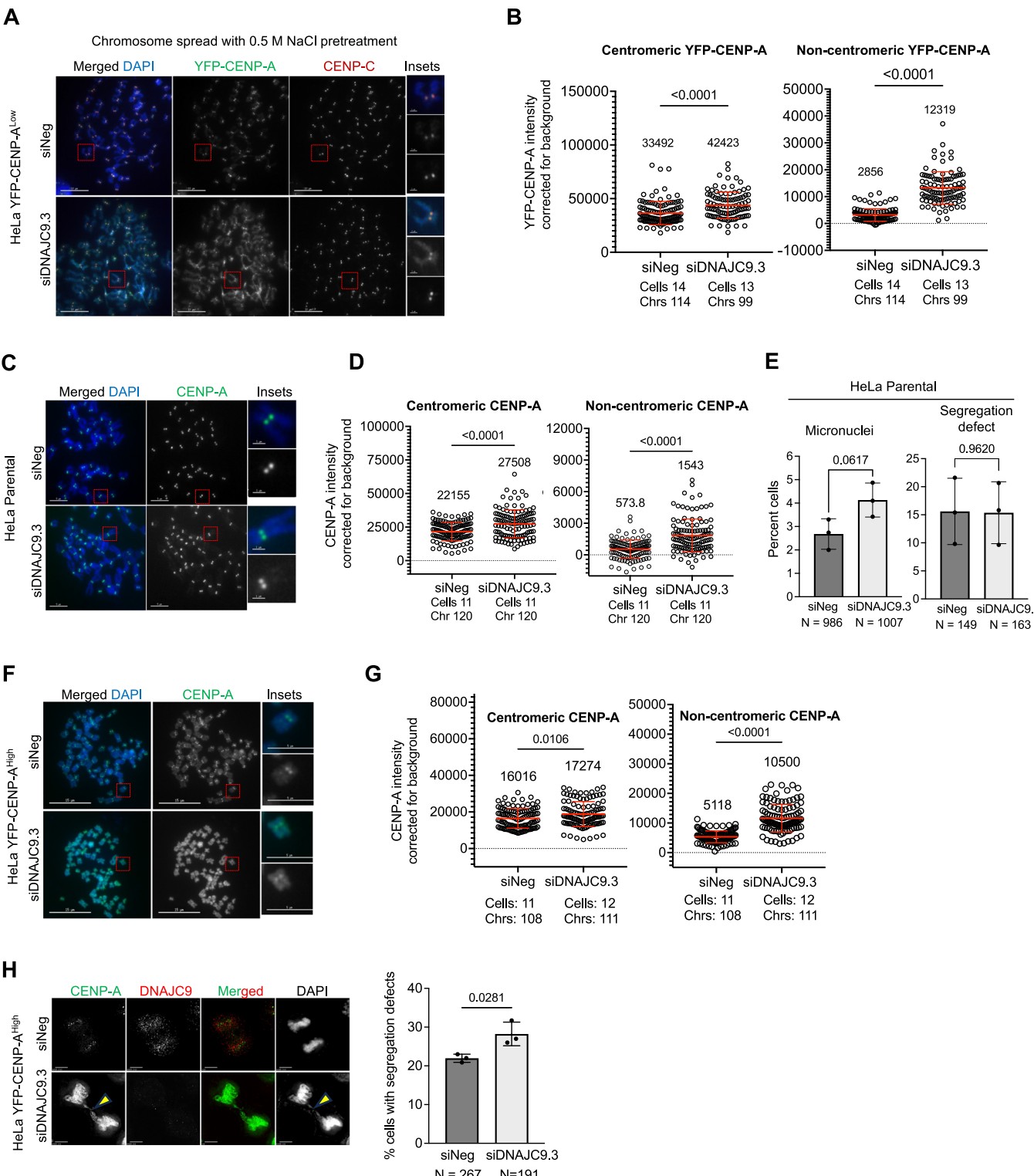

**A**

MS Analysis Workflow

(1) Raw data (3871 proteins - soluble and chromatin fractions)

(2) Filtering reverse hits, only identified by site and contaminants (3713 proteins), the second replicate for the siDNAJC9 condition was omitted due to lack of knockdown.

(3) Filtering for observations in 100 % of experiments siCTRL or siDNAJC9 or siCAF1B in <u>soluble</u> CENPA IP-MS replicates (soluble: 1813 proteins)

(4) Filtering for enrichment over beads in any condition (955 proteins, including 261 imputed ratios identified in ≤1 replicate in the bead only condition)  - panel B

(5) Bait normalisation - panel C

(6) siCTRL vs siDNAJC9 and siCTRL vs siCAF1B volcano plots - no imputed ratios - panels D & E

**B**   Soluble CENPA IP-MS

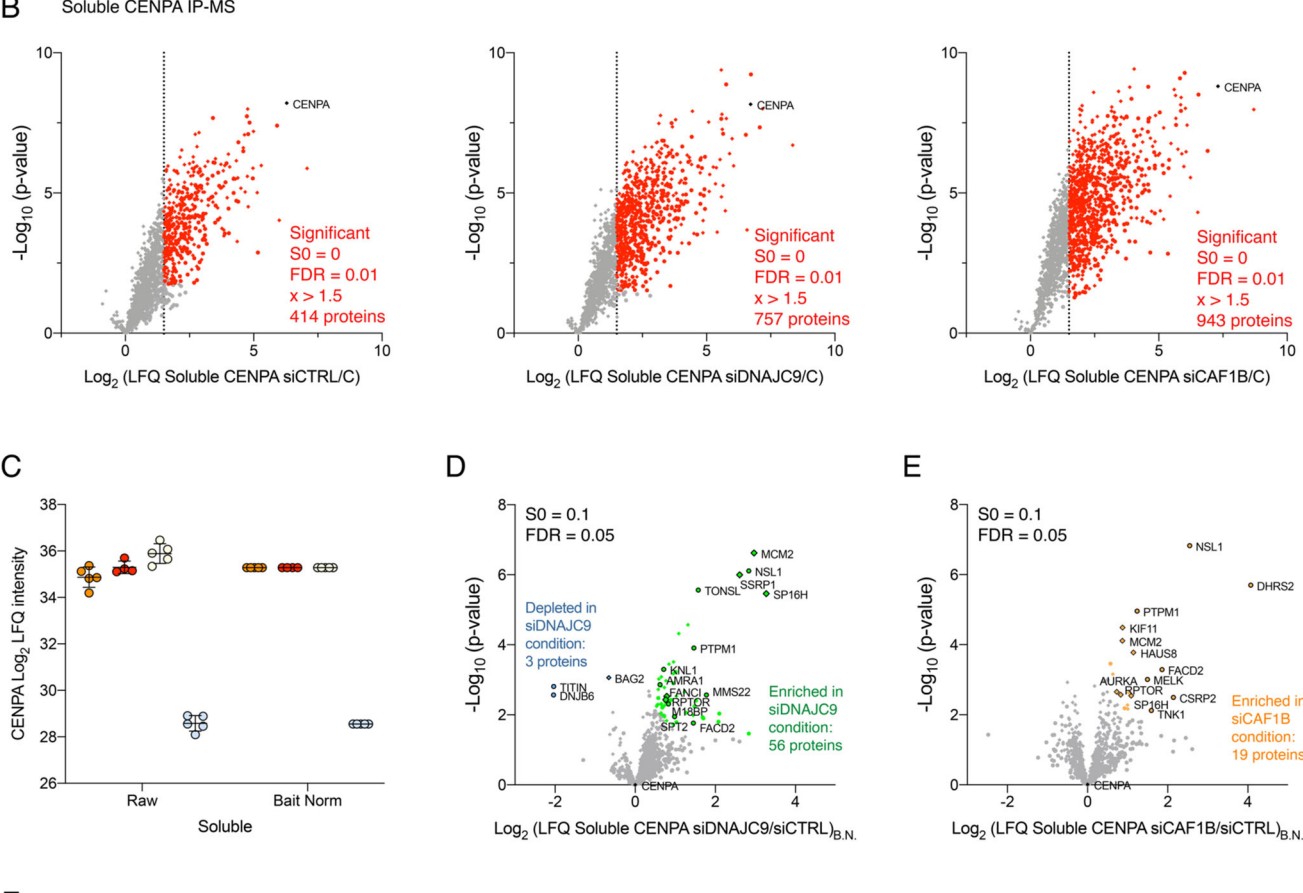

**Figure EV2.   Soluble CENP-A-Flag-HA IP-MS data analysis.**

(A) Overview of data analysis workflow. Data from $n = 5$ biological replicates and $n = 4$ biological replicates for siDNAJC9 conditions. See also Dataset EV4. (B) Volcano plots showing enrichment of proteins associated with soluble CENP-A-Flag-HA in siCTRL, siDNAJC9 and siCAF1B conditions compared to "C" uninduced control conditions. Factors highlighted in red are significant based on two-sided $T$ tests with $S0 = 0$, $FDR = 0.01$ and have a minimum Log2 fold change of 1.5 in their ratio of LFQ intensity (siRNA/C). Circular data points represent factors where ratios were imputed due to a lack of sufficient observations (>1) in uninduced control condition biological replicates. Proteins referred to by human UniProt protein identification code. (C) CENP-A LFQ intensity in purifications before and after bait normalization. (D, E) Volcano plots showing changes in bait normalized LFQ intensities ($LFQ_{B.N.}$) for factors specifically enriched with soluble CENP-A-Flag-HA comparing intensity changes in (D) siDNAJC9 and (E) siCAF1B conditions compared to DOX+ siCTRL conditions. Colors were used to highlight CENP-A and factors that show significant changes as assessed by two-sided $T$ tests with $S0 = 0.1$ and $FDR = 0.05$. Proteins referred to by human UniProt protein identification code. (F) Venn diagram color-coded as in (D, E), showing the overlap of factors classified as statistically changing in siDNAJC9 and siCAF1B conditions calculated using InteractiVenn (Heberle et al, 2015).

A

MS Analysis Workflow

(1) Raw data (3871 proteins - soluble and chromatin fractions)

(2) Filtering reverse hits, only identified by site and contaminants (3713 proteins), the second replicate for the siDNAJC9 condition was omitted due to lack of knockdown.

(3) Filtering for observations in 100 % of experiments siCTRL or siDNAJC9 or siCAF1B in <u>chromatin</u> CENPA IP-MS replicates (1395 proteins)

(4) Filtering for enrichment over beads in any condition (1045 proteins, including 342 imputed ratios identified in ≤1 replicate in the bead only condition) - panel B

(5) Bait normalisation - panel C

(6) siCTRL vs siDNAJC9 and siCTRL vs siCAF1B volcano plots - no imputed ratios - panels D & E

B    Chromatin CENPA IP-MS

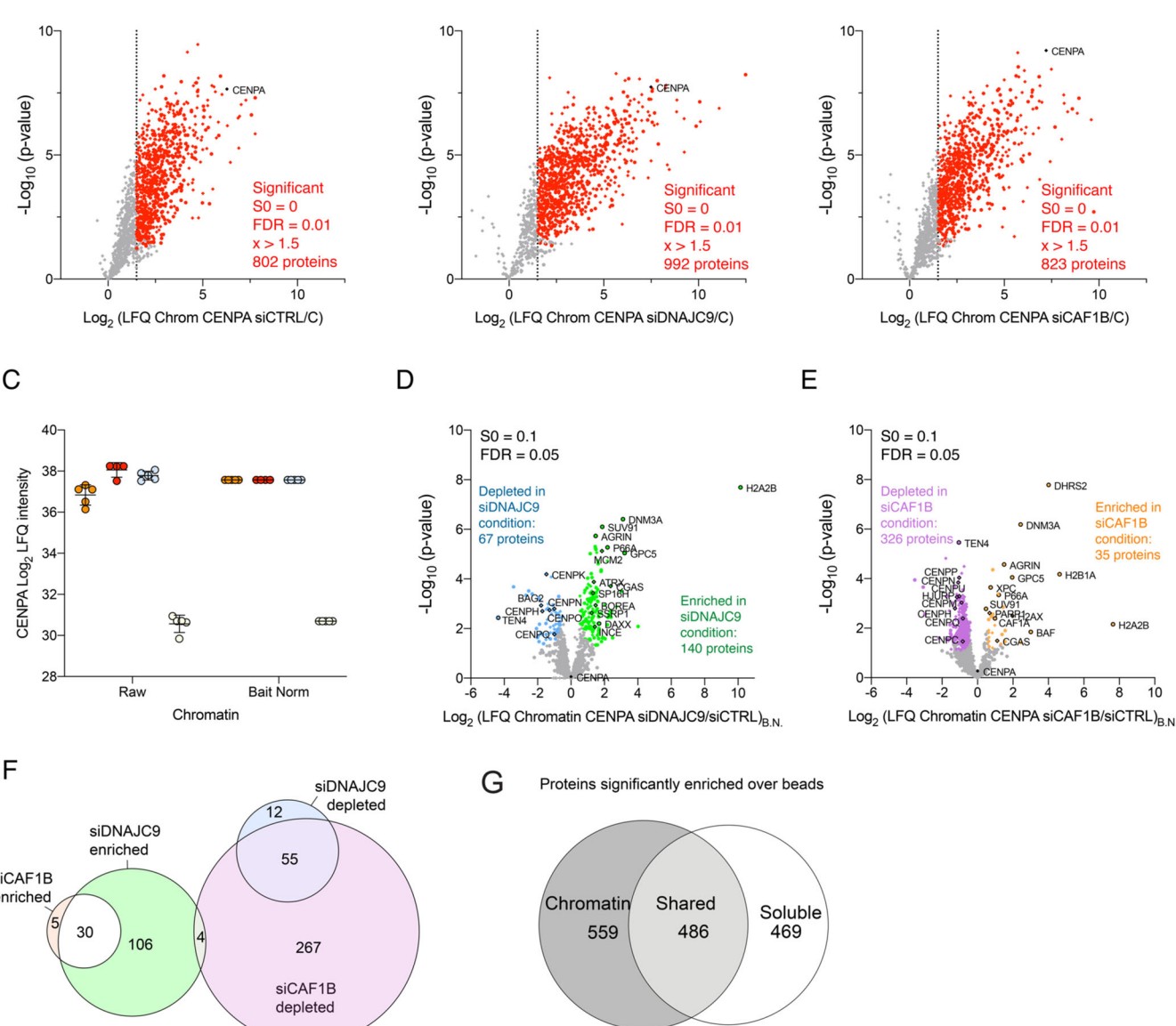

**Figure EV3. Chromatin CENP-A-Flag-HA IP-MS data analysis.**

(A) Overview of data analysis workflow. Data from $n = 5$ biological replicates and $n = 4$ biological replicates for siDNAJC9 conditions. See also Dataset EV4. (B) Volcano plots showing enrichment of proteins associated with chromatin-bound CENP-A-Flag-HA in siCTRL, siDNAJC9 and siCAF1B conditions compared to "C" uninduced control conditions. Factors highlighted in red are significant based on two-sided $T$ tests with $S0 = 0$, $FDR = 0.01$ and have a minimum Log2 fold change of 1.5 in their ratio of LFQ intensity (siRNA/C). Circular data points represent factors where ratios were imputed due to a lack of sufficient observations (>1) in uninduced control condition replicates. Proteins referred to by human UniProt protein identification code. (C) CENP-A LFQ intensity in purifications before and after bait normalization. (D, E) Volcano plots showing changes in bait normalized LFQ intensities ($LFQ_{B.N.}$) for factors specifically enriched with chromatin-bound CENP-A-Flag-HA comparing intensity changes in (D) siDNAJC9 and (E) siCAF1B conditions compared to DOX+ siCTRL conditions. Colors were used to highlight CENP-A and factors that show significant changes as assessed by two-sided $T$ tests with $S0 = 0.1$ and $FDR = 0.05$. Proteins referred to by human UniProt protein identification code. (F) Venn diagram, color-coded as in (D, E), showing the overlap of factors classified as statistically changing in siDNAJC9 and siCAF1B conditions calculated using InteractiVenn (Heberle et al, 2015). (G) Venn diagram, showing the overlap of factors specifically enriched with soluble and chromatin-bound CENP-A-Flag-HA calculated using InteractiVenn (Heberle et al, 2015).

A    Categorisation of GO-term clusters (Biological Process) enriched
in CENPA soluble IP-MS (Log2 siCTRL/beads ranked)

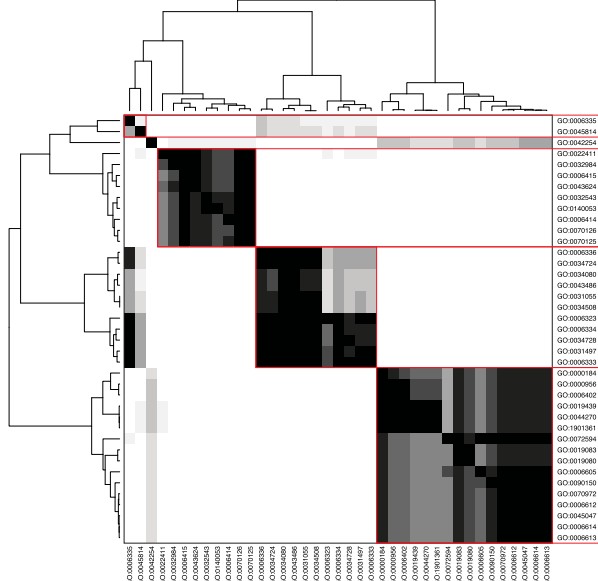

Rank) GO term catagory (mean enrichment score)

2) DNA replication-dependent nucleosome assembly &
Negative regulation of gene expression, epigenetic (3.20)

3) Mitochondrial translation (1.15)

1) CENP-A containing nucleosome assembly &
DNA replication-independent nucleosome assembly  (3.82)

4) Ribosomal proteins, *including ribosome biogenesis (0.20)

B    Categorisation of GO-term clusters (Biological Process) enriched
in CENPA chromatin IP-MS (Log2 siCTRL/beads ranked)

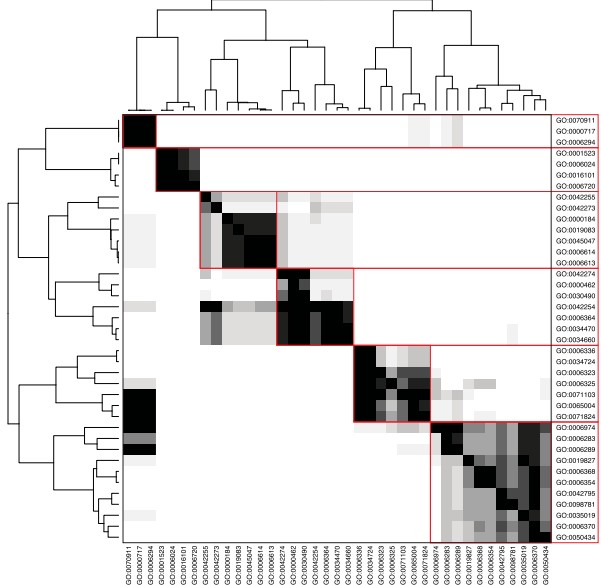

Rank) GO term catagories (mean enrichment score)

2) Nucleotide excision repair related (3.41)

1) Isoprenoid related metabolism (3.91)

6) Ribosome biogenesis (0.86)

5) rRNA processing & ribosome biogenesis (1.30)

4) Chromatin assembly & organisation,
inc. histone chaperones
& centromeric proteins (1.71)

3) Transcription related (2.60)

C    **GO terms representing highest enriched clusters:**

**Soluble CENPA IP-MS enriched (count, enrichment, FDR)**
GO:0034080, CENP-A containing nucleosome assembly (13, 5.28, 7.34E-06)
GO:0006335, DNA replication-dependent nucleosome assembly (6, 4.36, 0.0007)
**Chromatin CENPA IP-MS enriched (count, enrichment, FDR)**
GO:0006720, Isoprenoid metabolic process, (8, 3.64, 0.0054)
GO:0070911, Global genome nucleotide-excision repair (9, 3.41, 0.0054)
GO:0006354, DNA-templated transcription, elongation (24, 2.51, 0.00082)
GO:0006336, DNA replication-independent nucleosome assembly (26, 2.32, 0.0075)

◀ **Figure EV4.  Biological processes enriched with soluble and chromatin-bound CENP-A.**

(A, B) GO-analysis of proteins ranked by enrichment over beads in (A) soluble and (B) chromatin CENP-A IP-MS experiments were input into the String-db "Proteins with Values/Ranks - Functional Enrichment Analysis". To assess the redundancy of Biological Process GO-terms enriched, terms were clustered by the percentage identity of their associated proteins. GO-term clusters of terms were ranked by the mean fold enrichment of associated GO-terms in the String-db analysis. See also Dataset EV4. (C) GO-terms representing most highly enriched GO-term clusters defined in A and B.

   

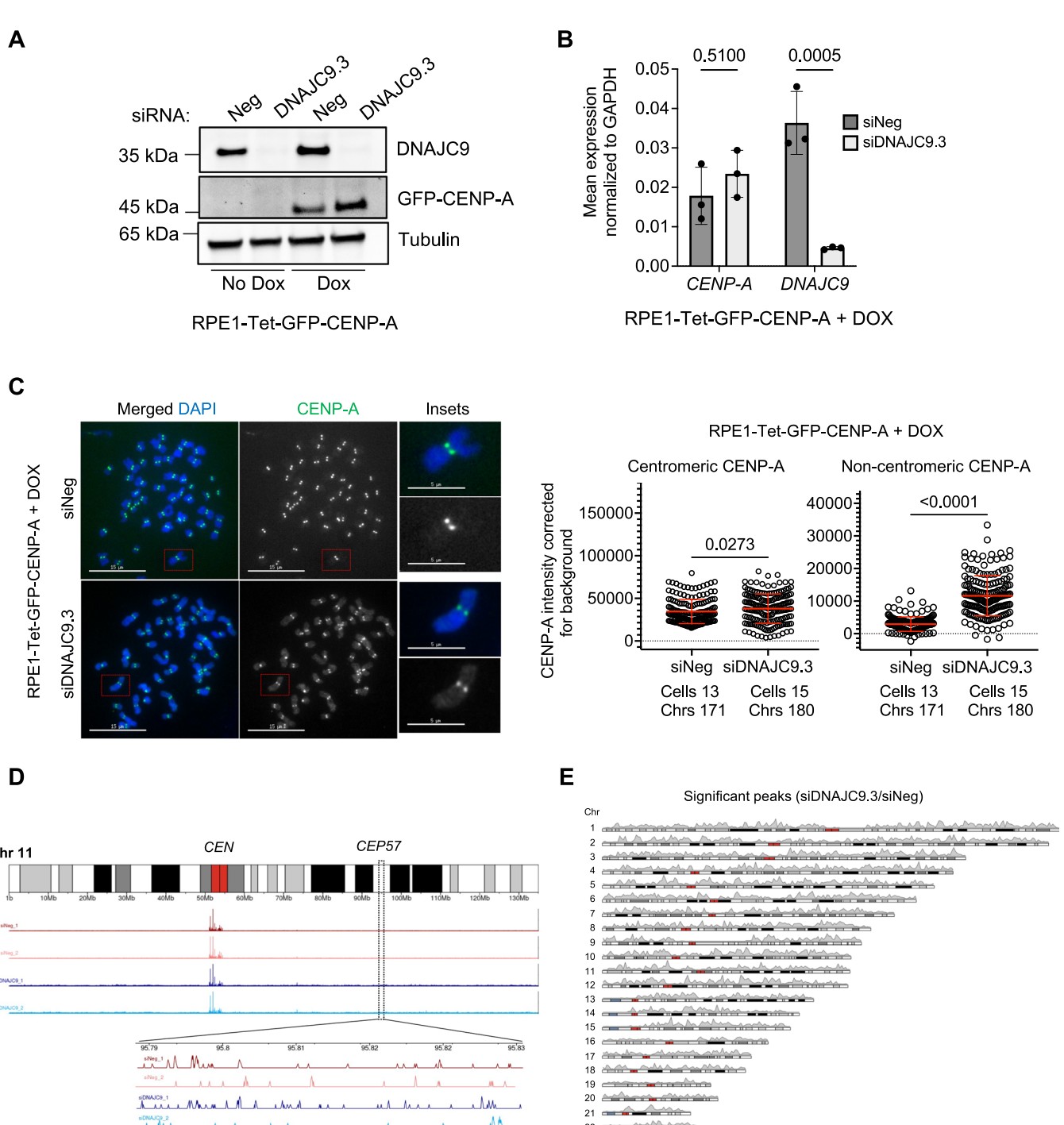

◀ **Figure EV5. DNAJC9 depletion promotes CENP-A mislocalization in RPE1-Tet-GFP-CENP-A cells without affecting CENP-A RNA levels.**

(A) Western blot of whole-cell extracts showing DNAJC9 depletion in siNeg or siDNAJC9.3-transfected RPE1-Tet-GFP-CENP-A cells with or without 1 μg/ml DOX induction for 48 h post-siRNA transfection. Alpha-tubulin was used as the loading control. (B) Transcript levels of *CENP-A* are not altered in DNAJC9-depleted RPE1-Tet-GFP-CENP-A cells treated with 1 μg/ml DOX for 48 h. Bar graphs showing mean RNA levels of *CENP-A* and *DNAJC9* normalized to *GAPDH* in control and DNAJC9-depleted cells using RT-qPCR. Mean values with standard deviation from three biological replicates were plotted and p values were calculated from two-way ANOVA with Sidak's multiple correction test. (C). CENP-A is mislocalized in DOX-treated RPE1-Tet-GFP-CENP-A cells with DNAJC9 depletion. (Left panel) Representative images of metaphase chromosome spreads immunostained for CENP-A in siNeg or siDNAJC9.3-transfected RPE1-Tet-GFP-CENP-A cells treated with 1 μg/ml DOX for 48 h post-siRNA transfection. Scale bar: 15 μm. Scale bar for insets: 5 μm. (Right) Scatter plots showing CENP-A intensities corrected for background at the centromeric or noncentromeric regions as described above, from three biological replicates. Each circle represents the value from an individual chromosome. Chrs refers to the total number of chromosomes measured per condition from three biological replicates. Mean with standard deviation is shown across all measurements from three biological replicates. *P* values were calculated using Mann–Whitney *U* test. (D) Example of binding profile of CENP-A on chromosome 11 in the CENP-A CUT&RUN sequencing datasets with siRNA treatments indicated. Peaks spanning the centromeric region (*CEN*) and an adjacent representative noncentromeric region (*CEP57*) are highlighted. Magnified view of noncentromeric localization of CENP-A at *CEP57* is also shown. (E) Karyoplot showing the distribution of the 17,341 significantly enriched (adjusted *P* value less than or equal to 0.05) CENP-A CUT&RUN peaks identified in DNAJC9-depleted GFP-CENP-A expressing RPE1 cells compared to siNeg-treated control. Plot represents the average from two biological replicates.

    