## [Peer Review File · The EMBO Journal]

DNAJC9 prevents CENP-A mislocalization and chromosomal instability by maintaining the fidelity of histone supply chains

Vinutha Balachandra, Roshan L. Shrestha, Colin M. Hammond, Shinjen Lin, Ivo A. Hendriks, Subhash Chandra Sethi, Lu Chen, Samantha Sevilla, Natasha J. Caplen, Raj Chari, Tatiana S. Karpova, Katherine McKinnon, Matthew A. M. Todd, Vishal Koparde, Ken Chih-Chien Cheng, Michael L. Nielsen, Anja Groth, and Munira A. Basrai

Corresponding author(s): Munira Basrai (basrain@nih.gov) , Colin Hammond (colin.hammond@liverpool.ac.uk)

Review Timeline:

Submission Date:	28th Sep 23
Editorial Decision:	27th Nov 23
Revision Received:	9th Feb 24
Editorial Decision:	1st Mar 24
Revision Received:	20th Mar 24
Accepted:	25th Mar 24

Editor: Hartmut Vodermaier

Transaction Report:

EMBOJ: Referee reports for EMBOJ-2023-115742

Dear Munira,

Thank you again for submitting your manuscript on factors preventing CENP-A mislocalization to The EMBO Journal. We have now received the reports of three expert referees, which I copy below for your information. As you will see, the referees express interest in the work, but also raise a considerable number of conceptual and technical concerns. Since it is not clear if and how these issues could be adequately addressed during a regular round of major revision, I would like to give you a chance to consider the reports and to provide a tentative point-by-point response, prior to taking a final decision in this case. I would therefore appreciate if you carefully looked into the attached reviews together with your coworkers, and prepared a revision plan detailing how each of the referees' points might be addressed or clarified. I would be particularly interested in your responses to the important main concern of referee 2; as well as to hear whether you might be able to obtain better insights into the mechanistic basis of DNAJC9 depletion phenotypes.

Based on these preliminary responses (which I may also discuss with some of the referees) and possible follow-up discussions via email or video call, we would then decide whether a major revision for The EMBO Journal would seem realistic and justified in this case. It would be great if you could get back to me with such a response by early next week.

Looking forward to hearing from you,

Best regards,

Hartmut

Referee #1 (Report for Author)

General summary and opinion:

In the present study, Balachandra et al. investigate factors that may be critical regulators of the localization of the centromeric variant of histone H3, CENP-A. For this, they performed a genome-wide RNAi screen followed by an image-based analysis of nuclear CENP-A levels, as a

readout for mislocalized CENP-A which becomes enriched in non-centromeric chromatin. Amongst top candidates whose depletion leads to increased nuclear CENP-A intensity, many function in regulating deposition of other H3 variants, which supports the idea that H3-H4 homeostasis is key for preventing CENP-A mislocalization. They notably identify CHAF1A/B, consistent with their previous findings (Shrestha et al., 2019), and DNAJC9 (recently discovered as a co-chaperone for H3.1/2/3-H4). Then, they further investigate the importance of DNAJC9 in regulating the localization of CENP-A by analyzing metaphase chromosome spreads. Depleting DNAJC9 or targeting its J-domain (required for proper folding of H3-H4) confirmed that DNAJC9 is critical to regulate CENP-A localization in HeLa and RPE-1 cell lines. The targeting of DNAJC9 and subsequent CENP-A mislocalization lead to increased CIN. In addition, while previous work showed that DNAJC9 depletion leads to reduced load of MCM2 associated with histones H4 and H3.1 (Hammond et al., 2021), the interactome analysis of Balachandra et al. shows an enhanced interaction of CENP-A with MCM2 in DNAJC9-depleted cells. Furthermore, co-depletion of MCM2 and DNAJC9 could suppress CENP-A mislocalization. Therefore, they concluded that DNAJC9 prevents the mislocalization of CENP-A and CIN, and that MCM2 could contribute to recycling non-centromeric CENP-A.

Interestingly, overexpression of CENP-A is a common feature of many cancers (reviewed in Renaud-Pageot et al., 2022). Although we know that p53 contributes to the regulation of CENP-A expression (Filipescu et al., 2017), TCGA PanCancer Atlas studies (accessible on CBioPortal) show that both p53-defective and p53 wild type (p53-WT) cancers can display high levels of CENP-A, indicating additional levels of control that should be considered. Moreover, studies in human cell lines showed that overexpression of CENP-A leads to its association with other H3 chaperones (in addition to HJURP) and mislocalization at non-centromeric regions (Lacoste et al., 2014), which causes chromosomal instability (CIN) (Shrestha et al., 2017, 2021). Given that CIN is a hallmark of cancer that is strongly linked to tumor evolution, understanding factors that can be involved in CENP-A mislocalization is an important question that could help in the prognosis and treatment of CENP-A overexpressing cancers. This includes factors regulating CENP-A expression (in terms of levels and cell-cycle regulation), along with factors that could promote and prevent CENP-A mislocalization per se. Thus, in this context, the data presented here is relevant for the field.

In summary, this manuscript represents an interesting study that would deserve publication. However, to make a strong contribution, this would require addressing a number of points that we have outlined below, including suggestions for how, in particular, authors' conclusions could be better substantiated.

Comments:

- Authors propose that DNAJC9 "prevents CENP-A mislocalization and CIN". They also mention that high levels of CENP-A alone contribute to its mislocalization in a dose-dependent manner, and this is why they chose the "YFP-CENP-ALow" cell line (with 3-fold higher exogenous CENP-A

compared to endogenous levels in the parental cell line) and not the "YFP-CENP-A^{High}" (17-fold higher).

Of note, CENP-A levels in patient tumors can go over 1000-fold relative to healthy tissue (Jeffery et al., 2021). Thus, to strengthen their data, additional experiments could:

- Test whether, following DNAJC9 depletion in the context of CENP-A^{High}, they can detect increased ectopic CENP-A levels, and/or worsened CIN phenotype.
- Assess whether DNAJC9 co-overexpression with CENP-A would be sufficient to counteract CENP-A mislocalization (e.g., in the CENP-A^{High} cell line).
- CENP-A mislocalization induced by depletion of DNAJC9 in HeLa and RPE-1 cells could be due to an overexpression of CENP-A. The RT-qPCR performed for HeLa YFP-CENP-A^{Low} is a nice confirmation that CENP-A is unaffected, at the transcription level, by DNAJC9 and/or MCM2.
- Ideally, the same analysis should be performed in the RPE-1 cell line too to reinforce the point
- Figure EV8 could be titled "CENP-A mRNA levels are unaffected by depletion of DNAJC9, MCM2 or both in HeLa YFP Low."
- Authors assessed total CENP-A levels in the different HeLa cell lines by Western Blot (Fig. EV1). Ideally, the siDNAJC9 condition should be added for all, to compare CENP-A protein levels in all conditions as a control throughout their experiments.
- In the abstract page 2, first sentence: "... CENP-A is overexpressed and mislocalized in many cancers." Although CENP-A is frequently overexpressed in cancer, its mislocalization has actually only been documented in cell lines. Since this has not been formally shown in patient tumors, a little caution is required here. The sentence should be rephrased and could highlight this important link between CENP-A overexpression and mislocalization in cell lines, along with the work of Verrelle et al. (2021), which gives hints in terms of changes in CENP-A subnuclear distribution in normal and tumoral tissues.
- In the Introduction page 3, line 6: "The evolutionary conserved histone 3 variant CENP-A". While the essential functions of CENP-A are largely conserved, analysis of CENP-A homologs from several species shows a divergent ("fast-evolving") N-terminal tail. The sentence should be rephrased and refer to the function of CENP-A.
- Page 4, starting from line 8: "Preventing the mislocalization of overexpressed CENP-A by depleting the H3.3 chaperone DAXX (Lacoste et al., 2014) rescued the invasiveness and CIN phenotypes of CENP-A overexpressing cells..." The sentence should be rephrased: the reference is correct regarding the use of DAXX depletion to reestablish centromeric localization of CENP-A, but Lacoste et al. did not assess cell invasiveness. Of note, depletion of the HIRA complex has also been proposed to prevent CENP-A mislocalization (Nye et al., 2018) and could be cited too.
- Fig 1D: Images of DNAJC9 staining should be added.
- Fig 2C/D: To confirm that non-centromeric CENP-A signal can be detected at all cell cycle stages, authors relied on an image-based analysis of DAPI and EdU signals to detect G1/S/G2 cell populations.
- A representative image for each population should be added.
- In addition, figure legend indicates N as the number of cells analyzed per condition. This number does not appear on fig. 2D and should be added.
- Fig 3E: Fig should also show endogenous CENP-A on membrane at 17kDa.

- Fig6A (Western Blot of soluble and chromatin fractions): In addition to Memcode staining, Western should show appropriate loading controls for each cell fraction.
- Fig6B/C: The factors that will further be tested in the study could be highlighted.
- Page 16, line 20: "Controls included untransfected cells and cells expressing full-length DNAJC9 (WT)" The sentence may need to be rephrased as "Controls included untransfected cells and cells transfected with a full-length DNAJC9 (WT) vector."
- Page 17, authors argue that a blockade in the H3-H4 supply chain promotes the mislocalization of CENP-A (based on results showed in figure 8 and the impact of depleting DNAJC9 in general). This is indeed an interesting observation. However, one could expect the H3-H4 assembly pathway to be affected by a mutation in the HBD (or both domains) of DNAJC9. Based on results in figure 8, this is not the case (probably thanks to the endogenous DNAJC9 context), and it would be useful to comment on this result.
- Fig 9: Third section of the scheme shows J mutant of DNAJC9 bound to misfolded H3, associated with the chromosome. For clarity purposes, when describing their model, the authors could reiterate that J mutant of DNAJC9 becomes trapped on chromatin through spurious interactions of misfolded histones with DNA (Hammond et al., 2021).
- p53 is involved in the regulation of CENP-A expression (Filipescu et al., 2017) and a key determinant of how CENP-A impacts cell state and therapeutic response. The manuscript could thus mention the p53 status of the cells used for the analysis (HPV18 inhibited for HeLa and WT for RPE-1) and comment how DNAJC9 seems to prevent the mislocalization in both p53 backgrounds.
- Previous work showed that CENP-A nucleosome particles, homotypic at the centromere, involve a heterotypic tetramer containing CENP-A and H3.3 at ectopic positions (Lacoste et al., 2014). Based on their results, Balachandra et al. argue that a blockade in the H3-H4 supply chain could promote aberrant deposition of CENP-A. Ideally, authors could comment further on the type of particle that could be expected at ectopic position in the context of a shortage of H3-H4.
- Fig EV1: For clarity purposes, the figure could show the full name of cell lines (HeLa, HeLa S3 or RPE-1).
- Table S3: siRNA sequences are not aligned and part of them cut in the pdf (but the excel format is fine).

Referee #2 (Report for Author)

In the study titled "The histone H3-H4 co-chaperone DNAJC9 prevents CENP-A mislocalization and chromosomal instability" by Balachandra et al, the authors utilized a genome-wide RNAi screen in cells overexpressing CENP-A. Their primary objective was to identify proteins critical for preventing CENP-A mislocalization, using image-based analysis as their initial tool. Among the proteins screened, DNAJC9 emerged as a noteworthy factor influencing the localization of exogenous CENP-A. Cells subjected to DNAJC9 depletion exhibited an enrichment of CENP-A on chromatin, CENP-A mislocalization, and a mild chromosomal instability (CIN) phenotypes. An interactome analysis the showed that an CENP-A interacts more with the H3-H4 chaperone

MCM2 in DNAJC9-depleted cells and that co-depletion of MCM2 and DNAJC9 suppressed CENP-A mislocalization.

The manuscript is based on an interesting genome-wide RNA screen with many interesting findings that the authors may choose to work on in the future. In this study they decided to analyze DNAJC9. There is however, one fundamental aspect concerns the screen and that is partially addressed in Figure EV1A. In this figure, the authors presented whole-cell extract Western blots. However, it is essential to confirm whether CENP-A is genuinely incorporated into chromatin. They could do it for instance using some biochemical techniques such as chromatin salt extraction. The chromosome spreads featured in the later figures and the cell fractionation (chromatin/soluble) are insufficient to determine a genuine incorporation of exogenous CENP-A.

If CENP-A does not get incorporated the entire screen and the function of DNAJC9 in CENP-A localization needs to be questions. In this context, it is also important to address whether the tagged version of CENP-A rescues endogenous CENP-A depletion phenotypes.

I assume that CENP-A is also in the genome wide screen, is then the tagged version designed in a way that it does not get degraded by the siRNA used for CENP-A?

Figure 1D does not seem to be the same magnifications and there is no scale bar shown on the figure. Furthermore, the authors should explicitly state whether all three siRNAs targeting candidate genes yielded similar phenotypes. Table 1 lacks clarity and it remains unclear which parts represent different aspects of the study. Especially, the authors should highlight, which candidates only exhibited CENP-A localization effects with only one or two siRNAs. Generally, the extended data may be presented in a more user-friendly manner.

Figure 2: How the depletion of DNAJC9 impact the cell cycle is important and needs more carefully analysis for reasons the authors mention in the manuscript (Nechemia-Arbely et al, 2019). In EV3 b/c quantifications are missing and should be included especially because it looks like there is a shift towards G2/M.

Figure 3F: Previous publications on CENP-A overexpression reported an ectopic localization with clear foci forming at certain chromatin domains. Here, the ectopic signal is very uniform as a haze over all of the chromatin (and the surrounding). How do the authors explain this difference?

In Figure 4, the increase in mitotic defects seems minimal, it also seems very low in Hela cells in general, 'wt' Hela usually have more than the reported percentage of missegregation.

Figure 5 D: The authors have taken H2B as loading control for chromatin, which is fine. However, according to their model, H3-H4 is misfolded in DNAJC9 depleted cells and presumably then reduced to make space for more CENPA (as visible on their blot). This

hypothesis should be tested by for instance western blotting. The authors show in 5D that endogenous levels of CENP-A are also stabilized in DNAJC9 depleted cells, which is a very interesting finding because it suggests that no misexpression/mutation of CENP-A is required for more CENP-A. Would this suffice to see mitotic errors? In other words, are there mitotic errors in DNAJC9 depleted cells without exogenous CENP-A and could this be rescued by a reduction of CENP-A (for instance a partial depletion)? This would help to come up with nice hypothesis for cancer cells with excess CENP-A.

In Figure 7, I cannot follow their conclusion ('enhanced interaction of MCM2 with CENP-A contributes to CENP-A mislocalization in DNAJC9-depleted cells'). They show that '... increased CENP-A levels in siDNAJC9.3-transfected cells at both centromeric and non-centromeric regions when compared to siNeg-transfected cells (Fig 7B and C). MCM2 depletion alone resulted in reduced centromeric CENP-A levels relative to control cells, consistent with previous observation for a role of MCM2 in centromeric association of CENP-A (Zasadzinska et al., 2018). Cells co-depleted for both MCM2 and DNAJC9 showed reduced levels of CENP-A at both centromeric and non-centromeric regions (Fig 7B and C).' My first conclusion would be that MCM2 is upstream of DNAJC9's effects on CENP-A. From the results shown, I am not convinced that MCM2 contributes to CENP-A mislocalization in DNAJC9-depleted cells.

Figure 8: the conclusions and the subtitle cannot be drawn from the data shown since the authors have not examined H3 at all and only conclude this from published data that were performed with a different experimental set up (for instance the CENP-A overexpression).

There are many questions that remain unanswered by the study, such as the function of DNAJC9 depletion on endogenous CENP-A and its implications in cancer, particularly when CENP-A is misexpressed. Without this information, our understanding of the molecular mechanisms of CIN in cancer cells and the role of replication in controlling CENP-A levels does not significantly increase from what we already know (Zasadzinska et al 2018; Nechemia-Arbely et al, 2019), especially if we don't know whether ectopically expressed CENP-A is chromatin incorporated or chromatin associated and whether the used construct is functional/rescues CENP-A depletion.

Their model is interesting but again experiments are missing to prove this hypothesis. The authors claim that they 'demonstrate that DNAJC9 prevents the promiscuous entry of CENP-A into H3-H4 deposition pathways to prevent CIN, which is a known hallmark of many cancers'. However, CENP-A can get ectopically incorporated without any deleterious effects on mitosis as long as it is taken out again in a timely fashion. Wouldn't be the model than that DNAJC9 is involved in CENP-A's accurate removal?

It would also have been interesting to model CENP-A to the structure of DNAJC9, MCM2 and histones that has been already solved by some of the coauthors (Hammond et al 2021).

Importantly, the authors may want to consider ways to address the physiological role of the interaction of CENP-A with DNAJC9 and MCM2. Is it similar to what has been proposed for H3-H4?

The authors discuss CIN throughout the manuscript but do not address this extensively experimentally. There are data bases of different cancer entities that allow the correlation of DNAJC9 and MCM2 expression with CENP-A misexpression and perhaps there is even a cancer model where the authors could test their hypothesis directly.

In summary, the authors report on an interesting finding, however, their conclusions are not backed up entirely by their experiments which seem in part premature.

Referee #3 (Report for Author)

This study by Balachandra et al., by employing a genome-wide RNAi screen, identified several regulators of CENP-A localisation, including DNAJC9, a member of the J-domain containing heat shock protein HSP40 family discovered recently as a co-chaperone for H3/H4. DNAJC9 depletion was found to mislocalise CENP-A and increase nuclear CENP-A throughout the cell cycle. In addition, DNAJC9-depleted cells exhibited chromosome instability (CIN) phenotypes, such as increased micronuclei and defective chromosome segregation. Interactome analysis revealed that DNAJC9 depletion disrupts the balance of histone chaperones, particularly affecting the association of CENP-A with MCM2, a protein involved in DNA replication and nucleosome assembly. Co-depletion of MCM2 and DNAJC9 reduces CENP-A mislocalization, suggesting MCM2's involvement in this process. Based on these key observations, the authors conclude that DNAJC9 is crucial for preventing CENP-A mislocalisation and maintaining chromosome stability, and it does so by regulating the supply chain of histone H3-H4.

Overall, it is a solid work that convincingly highlights how disrupting the function of a DNAJC9, which is H3/H4 chaperone, can lead to CENP-A mislocalisation and chromosome instability. However, the way DNAJC9's contribution is phrased in this study (including the title) appears to suggest an active role for DNAJC9 in preventing CENP-A mislocalisation. In contrast, the data presented here indicate that DNAJC9's contribution to correct CENP-A localisation is rather indirect.

Concerns:

- While DNAJC9 depletion increases both centromeric and non-centromeric CENP-A levels, only the non-centromeric CENP-C levels are affected. It would be helpful if authors could explicitly comment on this.
- The interactome analysis shows that DNAJC9 depletion reduces the association of CENP-A deposition factors in the chromatin fraction: while the text mentions the reduction of Mis18a, figure 6C does not show Mis18a, only Mis18b, Mis18BP1 and HJURP are shown. It is important

to clarify if Mis18a was seen and if not, the authors need to comment on this observation.

- Experiments with DNAJC9 show that only the DNAJC9 J mutant (lacking enzymatic activity) increases CENP-A levels. Either the H3/H4 binding deficient mutant or the J mutant in combination with the H3/H4 binding defective mutant does not affect CENP -A levels. Possible explanations for these observations (why H3/H4 binding defective or the catalytically inactive J mutant, while in combination with H3/H4 binding defective mutant does not affect the CENP-A levels) are not adequately provided and need to be included in the revised manuscript. Details of the mutations (residue numbers) are not provided anywhere in the main text. This is crucial information and needs to be included in the revised manuscript.

EMBOJ-2023-115742:**The histone H3–H4 co-chaperone DNAJC9 prevents CENP-A mislocalization and chromosomal instability (Balachandra et al.)****November 22nd, 2023****Summary of revisions**

We appreciate the comments from the reviewers and have addressed them in a point-by-point manner by proposing to perform additional experiments, incorporating editorial changes and providing an explanation to their queries. In addition to editorial changes, we highlight below a list of experiments and changes that will be made to the figures.

List of experimental revisions:

1. Deplete DNAJC9 in HeLa YFP-CENP-A High cells and examine ectopic localization of CENP-A in mitotic chromosome spreads. Quantification for centromeric and non-centromeric CENP-A signals will be added.
2. Examine CIN phenotypes (micronuclei and mitotic errors) in HeLa YFP-CENP-A High cells depleted for DNAJC9
3. RT-qPCR to analyse CENP-A transcript levels in RPE1-Tet-GFP-CENP-A cells with and without depletion of DNAJC9.
4. Western blotting to show endogenous CENP-A in Fig 3E.
5. Genome wide analysis of CENP-A CUT&RUN Seq in RPE1-Tet-GFP-CENP-A in siNeg and siDNAJC9 transfected RPE1-Tet-GFP-CENP-A cells with two biological repeats. The data includes: 1. Genomic tracks of CENP-A peaks in siNeg and siDNAJC9 conditions with zoomed-in view of centromeric and non-centromeric regions; 2. Pie charts showing percent distribution of peaks enriched in siDNAJC9 condition over various genomic features; 3. Karyoplot showing whole genome distribution of enriched CENP-A peaks upon depletion of siDNAJC9; 4. Overlap of CENP-A CUT and RUN profile against previously identified ATAC-Seq sites (GSM6383021) for siNeg and siDNAJC9.

List of figure changes:

6. Fig 2C will be changed to include representative image of G1, S and G2 nuclei. Fig 2D will be revised to show the number of cells analysed.
7. Fig 6A will be revised to include appropriate loading controls for each cell fraction. Fig 6C will be revised to highlight the candidate chosen for in-depth studies i.e. MCM2.

8. Fig 9: Based on reviewer's comment and our data, we will revise the model to show that the J mutant is also trapped with misfolded histones in the soluble fraction causing an increase in misfolded H3–H4.
9. Fig EV1 will be revised to show the full name of cell lines (HeLa, HeLa S3).
10. Figure 1D will be revised to include panel with scale bar.
11. Table 1 will be revised to include additional details requested by the reviewer.

EMBOJ-2023-115742:**The histone H3–H4 co-chaperone DNAJC9 prevents CENP-A mislocalization and chromosomal instability (Balachandra et al.)****Response to Reviewers comments indicated in blue font.****November 22nd 2023****Referee #1 (Report for Author)**

General summary and opinion:

In the present study, Balachandra *et al.* investigate factors that may be critical regulators of the localization of the centromeric variant of histone H3, CENP-A. For this, they performed a genome-wide RNAi screen followed by an image-based analysis of nuclear CENP-A levels, as a readout for mislocalized CENP-A which becomes enriched in non-centromeric chromatin. Amongst top candidates whose depletion leads to increased nuclear CENP-A intensity, many function in regulating deposition of other H3 variants, which supports the idea that H3-H4 homeostasis is key for preventing CENP-A mislocalization. They notably identify CHAF1A/B, consistent with their previous findings (Shrestha et al., 2019), and DNAJC9 (recently discovered as a co-chaperone for H3.1/2/3-H4). Then, they further investigate the importance of DNAJC9 in regulating the localization of CENP-A by analyzing metaphase chromosome spreads. Depleting DNAJC9 or targeting its J-domain (required for proper folding of H3-H4) confirmed that DNAJC9 is critical to regulate CENP-A localization in HeLa and RPE-1 cell lines. The targeting of DNAJC9 and subsequent CENP-A mislocalization lead to increased CIN. In addition, while previous work showed that DNAJC9 depletion leads to reduced load of MCM2 associated with histones H4 and H3.1 (Hammond et al., 2021), the interactome analysis of Balachandra et al. shows an enhanced interaction of CENP-A with MCM2 in DNAJC9-depleted cells. Furthermore, co-depletion of MCM2 and DNAJC9 could suppress CENP-A mislocalization. Therefore, they concluded that DNAJC9 prevents the mislocalization of CENP-A and CIN, and that MCM2 could contribute to recycling non-centromeric CENP-A.

Interestingly, overexpression of CENP-A is a common feature of many cancers (reviewed in Renaud-Pageot et al., 2022). Although we know that p53 contributes to the regulation of CENP-A expression (Filipescu et al., 2017), TCGA PanCancer Atlas studies (accessible on CBioPortal) show that both p53-defective and p53 wild type (p53-WT) cancers can display high levels of CENP-A, indicating additional levels of control that should be considered. Moreover, studies in human cell lines showed that overexpression of CENP-A leads to its association with other H3 chaperones (in addition to HJURP) and mislocalization at non-centromeric regions (Lacoste et al., 2014), which causes chromosomal instability (CIN) (Shrestha et al., 2017, 2021). Given that CIN is a hallmark of cancer that is strongly linked to tumor evolution, understanding factors that can be involved in CENP-A mislocalization is an important question that could help in the prognosis and treatment of CENP-A overexpressing cancers. This includes factors regulating CENP-A expression (in terms of levels and

cell-cycle regulation), along with factors that could promote and prevent CENP-A mislocalization per se. Thus, in this context, the data presented here is relevant for the field.

In summary, this manuscript represents an interesting study that would deserve publication. However, to make a strong contribution, this would require addressing a number of points that we have outlined below, including suggestions for how, in particular, authors' conclusions could be better substantiated. We thank the reviewer for their insightful summary of our work and for highlighting the relevance to the field. We appreciate the nature of the constructive critique which is aimed at improving our manuscript and we will address the points as outlined below.

Comments:

1. Authors propose that DNAJC9 "prevents CENP-A mislocalization and CIN". They also mention that high levels of CENP-A alone contribute to its mislocalization in a dose-dependent manner, and this is why they chose the "YFP-CENP-ALow" cell line (with 3-fold higher exogenous CENP-A compared to endogenous levels in the parental cell line) and not the "YFP-CENP-AHigh" (17-fold higher). Of note, CENP-A levels in patient tumors can go over 1000-fold relative to healthy tissue (Jeffery et al., 2021). Thus, to strengthen their data, additional experiments could:

- Test whether, following DNAJC9 depletion in the context of CENP-AHigh, they can detect increased ectopic CENP-A levels, and/or worsened CIN phenotype.

We agree with the reviewer. We used the HeLa YFP-CENP-A High cell line for the siRNA screen where we observed increased nuclear YFP-CENP-A intensity upon DNAJC9 depletion (Fig 1D), which correlates with increased mislocalization of CENP-A. However, as suggested by the reviewer we will examine the ectopic localization of CENP-A in metaphase chromosomes, and CIN phenotypes in HeLa YFP-CENP-A High cells upon DNAJC9 depletion.

2. Assess whether DNAJC9 co-overexpression with CENP-A would be sufficient to counteract CENP-A mislocalization (e.g., in the CENP-A High cell line).

The proposed model for the role of DNAJC9 in preventing CENP-A mislocalization and CIN is through maintenance of H3-H4 supply (Fig 9) and hence perturbation of H3-H4 supply in the context of DNAJC9 depletion is proposed to favour CENP-A mislocalization. It is possible that DNAJC9 overexpression may further increase H3-H4 supply and thereby competitively prevent mislocalization of CENP-A in the HeLa YFP-CENP-A High cell line. The Tet-inducible siRNA resistant DNAJC9 construct that we used to examine the functional specificity of DNAJC9 in the current manuscript (Fig 2A,B) can be expressed in HeLa YFP-CENP-A High cells to examine this possibility. However, in our opinion over-expression of DNAJC9 alone is highly unlikely to rescue the phenotype as it would not

be sufficient to increase H3-H4 biosynthesis and supply alone. We can test this possibility but note that a negative result is also compatible with the proposed model.

3. CENP-A mislocalization induced by depletion of DNAJC9 in HeLa and RPE-1 cells could be due to an overexpression of CENP-A. The RT-qPCR performed for HeLa YFP-CENP-A^{Low} is a nice confirmation that CENP-A is unaffected, at the transcription level, by DNAJC9 and/or MCM2.

- Ideally, the same analysis should be performed in the RPE-1 cell line too to reinforce the point. We agree with the reviewer and will perform qPCRs for CENP-A mRNA in RPE-1 cells with and without depletion of DNAJC9 to complement the chromosome spread data for CENP-A mislocalization in Figure 3F.

4. Figure EV8 could be titled "CENP-A mRNA levels are unaffected by depletion of DNAJC9, MCM2 or both in HeLa YFP Low."

We agree with the reviewer in terms of clarifying the title for Figure EV8 to state that CENP-A mRNA levels are unaffected by depletion or co-depletion of DNAJC9 and MCM2 in HeLa YFP Low cells.

5. Authors assessed total CENP-A levels in the different HeLa cell lines by Western Blot (Fig. EV1). Ideally, the siDNAJC9 condition should be added for all, to compare CENP-A protein levels in all conditions as a control throughout their experiments.

The Western blot shown in Fig EV1 aims to compare the steady state levels of CENP-A in different cell lines, and based on this we decided to use HeLa YFP low cell line for in-depth studies. We have shown the effects of DNAJC9 depletion on CENP-A levels in the HeLa CENP-A-YFP-Low cell line (Fig 5D), HeLa S3 CENP-A-Flag-HA (Fig 6A) and RPE1-Tet-GFP-CENP-A (Fig 3E). In addition, we will include western blot for CENP-A-YFP-High cell line.

6. In the abstract page 2, first sentence: "... CENP-A is overexpressed and mislocalized in many cancers." Although CENP-A is frequently overexpressed in cancer, its mislocalization has actually only been documented in cell lines. Since this has not been formally shown in patient tumors, a little caution is required here. The sentence should be rephrased and could highlight this important link between CENP-A overexpression and mislocalization in cell lines, along with the work of Verrelle et al. (2021), which gives hints in terms of changes in CENP-A subnuclear distribution in normal and tumoral tissues.

We agree with the reviewer and will make the suggested changes to the references.

7. In the Introduction page 3, line 6: "The evolutionary conserved histone 3 variant CENP-A". While the essential functions of CENP-A are largely conserved, analysis of CENP-A homologs from several

species shows a divergent ("fast-evolving") N-terminal tail. The sentence should be rephrased and refer to the function of CENP-A.

We agree with the reviewer and will make this amendment.

8. Page 4, starting from line 8: "Preventing the mislocalization of overexpressed CENP-A by depleting the H3.3 chaperone DAXX (Lacoste et al., 2014) rescued the invasiveness and CIN phenotypes of CENP-A overexpressing cells..." The sentence should be rephrased: the reference is correct regarding the use of DAXX depletion to reestablish centromeric localization of CENP-A, but Lacoste et al. did not assess cell invasiveness. Of note, depletion of the HIRA complex has also been proposed to prevent CENP-A mislocalization (Nye et al., 2018) and could be cited too.

We will correct this and include the reference for the HIRA complex.

9. Fig 1D: Images of DNAJC9 staining should be added.

Fig 1D is a representative image from the primary siRNA screen, therefore DNAJC9 staining was not performed, and hence cannot be included.

10. Fig 2C/D: To confirm that non-centromeric CENP-A signal can be detected at all cell cycle stages, authors relied on an image-based analysis of DAPI and EdU signals to detect G1/S/G2 cell populations. - A representative image for each population should be added.

We will include insets in Fig 2C to separately show the G1, S and G2 nuclei.

11. In addition, figure legend indicates N as the number of cells analyzed per condition. This number does not appear on fig. 2D and should be added.

We thank the reviewer for pointing out the omission of the number of cells in the figure. We will add this information to the revised Fig 2D.

12. Fig 3E: Fig should also show endogenous CENP-A on membrane at 17kDa.

We agree with the reviewer and in fact, had probed the western blots to examine endogenous CENP-A. However, we find that the levels of endogenous CENP-A are barely detectable on the western blot and long exposure shows high background, and hence was not included in the manuscript. We will repeat the western blotting with higher sample amount to allow better detection of the endogenous CENP-A signal with less background and include this panel in the revised manuscript.

13. Fig6A (Western Blot of soluble and chromatin fractions): In addition to Memcode staining, Western should show appropriate loading controls for each cell fraction.

We will provide western blot data as requested by the reviewer.

14. Fig6B/C: The factors that will further be tested in the study could be highlighted.

We agree with the reviewer and will highlight MCM2 in the figure.

15. Page 16, line 20: "Controls included untransfected cells and cells expressing full-length DNAJC9 (WT)" The sentence may need to be rephrased as "Controls included untransfected cells and cells transfected with a full-length DNAJC9 (WT) vector."

We will change this in the revised submission.

16. Page 17, authors argue that a blockade in the H3-H4 supply chain promotes the mislocalization of CENP-A (based on results showed in figure 8 and the impact of depleting DNAJC9 in general). This is indeed an interesting observation. However, one could expect the H3-H4 assembly pathway to be affected by a mutation in the HBD (or both domains) of DNAJC9. Based on results in figure 8, this is not the case (probably thanks to the endogenous DNAJC9 context), and it would be useful to comment on this result.

We appreciate the insight from the reviewer and will include a comment on this in the revised manuscript.

17. Fig 9: Third section of the scheme shows J mutant of DNAJC9 bound to misfolded H3, associated with the chromosome. For clarity purposes, when describing their model, the authors could reiterate that J mutant of DNAJC9 becomes trapped on chromatin through spurious interactions of misfolded histones with DNA (Hammond et al., 2021).

Based on the reviewer's comment and Hammond *et al.* 2021, we will revise the model to show that the J mutant is trapped with misfolded histones both in the soluble and the chromatin fraction contributing to an increase in misfolded H3-H4.

18. p53 is involved in the regulation of CENP-A expression (Filipescu et al., 2017) and a key determinant of how CENP-A impacts cell state and therapeutic response. The manuscript could thus mention the p53 status of the cells used for the analysis (HPV18 inhibited for HeLa and WT for RPE-1) and comment how DNAJC9 seems to prevent the mislocalization in both p53 backgrounds.

We agree with the reviewer about including observations from previous studies with p53 by Filipescu *et al.* 2017 and add a sentence in the discussion that mislocalization of CENP-A is observed in both WT and mutant p53 backgrounds.

19. Previous work showed that CENP-A nucleosome particles, homotypic at the centromere, involve a heterotypic tetramer containing CENP-A and H3.3 at ectopic positions (Lacoste *et al.*, 2014). Based on their results, Balachandra et al. argue that a blockade in the H3-H4 supply chain could promote

aberrant deposition of CENP-A. Ideally, authors could comment further on the type of particle that could be expected at ectopic position in the context of a shortage of H3-H4.

The reviewer brings up an interesting point about the composition of the nucleosome. Based on our interactome data, we see H3.3 enriched with soluble CENP-A upon DNAJC9 depletion but not on chromatin. In the context of DNAJC9 depletion, the exact composition of the ectopic CENP-A nucleosomes is not clear, however, as H3.3 is misfolded in the absence of DNAJC9, it is conceivable that homotypic CENP-A nucleosomes are formed at ectopic sites. We will include this comment in the discussion of the revised manuscript.

20. Fig EV1: For clarity purposes, the figure could show the full name of cell lines (HeLa, HeLa S3 or RPE-1).

We agree with the reviewer and will revise the figure.

21. Table S3: siRNA sequences are not aligned and part of them cut in the pdf (but the excel format is fine).

We will make the required change.

Referee #2 (Report for Author)

In the study titled "The histone H3-H4 co-chaperone DNAJC9 prevents CENP-A mislocalization and chromosomal instability" by Balachandra et al, the authors utilized a genome-wide RNAi screen in cells overexpressing CENP-A. Their primary objective was to identify proteins critical for preventing CENP-A mislocalization, using image-based analysis as their initial tool. Among the proteins screened, DNAJC9 emerged as a noteworthy factor influencing the localization of exogenous CENP-A. Cells subjected to DNAJC9 depletion exhibited an enrichment of CENP-A on chromatin, CENP-A mislocalization, and a mild chromosomal instability (CIN) phenotypes. An interactome analysis showed that CENP-A interacts more with the H3-H4 chaperone MCM2 in DNAJC9-depleted cells and that co-depletion of MCM2 and DNAJC9 suppressed CENP-A mislocalization.

1. The manuscript is based on an interesting genome-wide RNA screen with many interesting findings that the authors may choose to work on in the future. In this study they decided to analyze DNAJC9. There is however, one fundamental aspect concerns the screen and that is partially addressed in Figure EV1A. In this figure, the authors presented whole-cell extract Western blots. However, it is essential to confirm whether CENP-A is genuinely incorporated into chromatin. They could do it for instance using some biochemical techniques such as chromatin salt extraction. The chromosome spreads featured in the later figures and the cell fractionation (chromatin/soluble) are insufficient to determine a genuine incorporation of exogenous CENP-A.

We agree with the reviewer's comment for additional data to support the genuine incorporation of exogenous CENP-A for our studies. The western blot in Fig EV1 compares the steady state levels of CENP-A in different cell lines to justify the choice of HeLa YFP-CENP-A Low for in-depth studies. To address the reviewer's comment about incorporation of mislocalized CENP-A into chromatin, we have recently completed CENP-A CUT&RUN Seq to examine genome-wide binding profile of CENP-A in siDNAJC9 and siNeg cells. These experiments were done using stable diploid model RPE1-Tet-GFP-CENP-A induced with Doxycycline which show mislocalization of CENP-A on chromosome spreads upon DNAJC9 depletion (Fig 3F,G). The rationale for using RPE1 cells is to support the cell biology and biochemical data from HeLa cells in an independent model which is not aneuploid.

Our CUT&RUN Seq data with two independent repeats for siNeg and siDNAJC9 identified 16,613 non-centromeric CENP-A peaks enriched in siDNAJC9 condition, majority of which map to promoter, intronic and exon regions (Fig 1A below). The karyoplots show whole genome distribution of significantly enriched CENP-A peaks upon depletion of siDNAJC9 compared to siNeg control along all the chromosomes (two replicates of each; Fig 1B below). We also observed that the enriched peaks identified in two repeats of siDNAJC9 condition overlap with open chromatin regions based on a published ATAC-Seq dataset of RPE1 (Fig 1C below; GSM6383021). We will also include

representative genomic tracks of zoomed-in peak profile for CENP-A at centromeric and non-centromeric regions in siNeg and siDNAJC9 conditions. The new figure will strengthen our conclusion that mislocalized CENP-A is incorporated into chromatin in DNAJC9-depleted cells.

Fig 1. A. Genome wide analysis of CENP-A CUT&RUN Seq in RPE1-Tet-GFP-CENP-A with DNAJC9 depletion. A. Pie charts showing percent distribution of peaks enriched in siDNAJC9 condition over various genomic features; B. Karyoplot showing whole genome distribution of enriched CENP-A peaks upon depletion of siDNAJC9 C. Overlay of CENP-A CUT and RUN profile against previously identified ATAC-Seq sites (GSM6383021) for siNeg and siDNAJC9. The data presented in all the subpanels represents two independent biological repeats of siNeg and siDNAJC9.

2. If CENP-A does not get incorporated the entire screen and the function of DNAJC9 in CENP-A localization needs to be questions. In this context, it is also important to address whether the tagged version of CENP-A rescues endogenous CENP-A depletion phenotypes.

Please refer to our response to comment #1 above for a new Figure reporting CENP-A CUT&RUN Seq data to address the incorporation of CENP-A into chromatin in cells depleted for DNAJC9.

For the functionality of the tagged version of CENP-A, there are several published studies (Fachinetti *et al.*, NCB 2013; Nechemia-Arbely *et al.*, JCB 2017; Black *et al.*, Nature 2004; Foltz *et al.*, Cell, 2009; Kops *et al.*, PNAS, 2004) that have used the same construct (YFP-CENP-A) and importantly, the tagged version of CENP-A used in our studies has been shown to localize to the centromere and support long term cell viability (Nechemia-Arbely *et al.*, JCB 2017).

3. I assume that CENP-A is also in the genome wide screen, is then the tagged version designed in a way that it does not get degraded by the siRNA used for CENP-A?

The tagged version (YFP-CENP-A) is not siRNA resistant and can be targeted by CENP-A siRNA which is included in the genome-wide screen.

4. Figure 1D does not seem to be the same magnifications and there is no scale bar shown on the figure. Furthermore, the authors should explicitly state whether all three siRNAs targeting candidate genes yielded similar phenotypes. Table 1 lacks clarity and it remains unclear which parts represent different aspects of the study. Especially, the authors should highlight, which candidates only exhibited CENP-A localization effects with only one or two siRNAs. Generally, the extended data may be presented in a more user-friendly manner.

We apologize for the omission of the details in Table 1. We will revise Figure 1D showing the scale bar. The revised Table 1 will include additional details requested by the reviewer.

5. Figure 2: How the depletion of DNAJC9 impact the cell cycle is important and needs more carefully analysis for reasons the authors mention in the manuscript (Nechemia-Arbely et al, 2019). In EV3 b/c quantifications are missing and should be included especially because it looks like there is a shift towards G2/M.

The reviewer raises an important point about the effect of DNAJC9 depletion on the cell cycle given the cell cycle stage-dependent CENP-A mislocalization reported in a previous study (Nechemia-Arbely *et al*, 2019). To address this aspect, we performed FACS analysis of asynchronous cells with and without DNAJC9 depletion which revealed that the cell cycle profile is unchanged (Fig EV4). As requested by the reviewer we will include quantifications for Fig. EV3 B,C. Furthermore, results from Figure 2D show that cells depleted for DNAJC9 exhibit increased nuclear CENP-A levels at G1, S and G2 stages.

6. Figure 3F: Previous publications on CENP-A overexpression reported an ectopic localization with clear foci forming at certain chromatin domains. Here, the ectopic signal is very uniform as a haze over all of the chromatin (and the surrounding). How do the authors explain this difference?

The nature of ectopic CENP-A signal depends on the context that promotes mislocalization. Upon depletion of an H3-H4 chaperone such as DNAJC9, a genome-wide broad distribution of CENP-A is expected, which appears as a uniform signal on mitotic chromosome spreads as shown in Fig 3. We do not observe signal outside the DAPI stained region and our quantifications include background correction to exclude signal outside the chromatin region. In addition, our CUT&RUN data showing broad distribution of the non-centromeric peaks enriched in siDNAJC9 condition complements the uniform pattern of ectopic CENP-A signal observed in our chromosome spreads. We note that

previous studies report foci-like signal of ectopic CENP-A under unperturbed conditions where specific gene depletions were not performed (Lacoste *et al.*, 2014; Athwal *et al.*, 2015).

7. In Figure 4, the increase in mitotic defects seems minimal, it also seems very low in HeLa cells in general, 'wt' HeLa usually have more than the reported percentage of missegregation.

Our parental HeLa cells exhibit a relatively low range of CIN phenotypes (micronuclei, segregation defect) (Figure 4G) as we previously reported (Shrestha *et al.*, 2017, 2023). This is consistent with other studies for low levels of CIN in HeLa parental cells (Piette *et al.*, 2021). In Figure 4E and F, we presented data showing significant elevation of CIN phenotypes in HeLa YFP-CENP-A Low cells upon depletion of DNAJC9. Similar to results with CHAF1B depletion (Shrestha *et al.*, 2023), the low levels of CENP-A mislocalization upon DNAJC9 depletion in the parental HeLa cell line (2.7-fold; Fig 3C, D) are insufficient to induce CIN phenotypes (Fig 4G). We hope this clarifies the reviewer's concern.

8. Figure 5 D: The authors have taken H2B as loading control for chromatin, which is fine. However, according to their model, H3-H4 is misfolded in DNAJC9 depleted cells and presumably then reduced to make space for more CENP-A (as visible on their blot). This hypothesis should be tested by for instance western blotting.

The reviewer raises a good point. The concern with western blot analysis of chromatin bound H3 is that misfolded H3-H4 can spuriously interact with DNA (Hammond *et al.*, 2021), and western blots would not be able to distinguish between productive and non-productive association of H3-H4 with the chromatin. Alternatively, this hypothesis could be tested by examining the localization of CENP-A using mitotic chromosome spreads from control and H3-depleted cells. However, robust H3 depletion would require siRNA-mediated targeting of all forms of H3 (H3.1/2/3).

9. The authors show in 5D that endogenous levels of CENP-A are also stabilized in DNAJC9 depleted cells, which is a very interesting finding because it suggests that no misexpression/mutation of CENP-A is required for more CENP-A. Would this suffice to see mitotic errors? In other words, are there mitotic errors in DNAJC9 depleted cells without exogenous CENP-A and could this be rescued by a reduction of CENP-A (for instance a partial depletion)? This would help to come up with nice hypothesis for cancer cells with excess CENP-A.

The reviewer's suggestion to examine if depletion of DNAJC9 leads to increased mitotic errors in HeLa parental cells is described in Figure 4G and our results do not show a significant increase in mitotic errors in these cells as explained in response to comment #7 above.

10. In Figure 7, I cannot follow their conclusion that enhanced interaction of MCM2 with CENP-A contributes to CENP-A mislocalization in DNAJC9-depleted cells. They show that increased CENP-A

levels in siDNAJC9.3-transfected cells at both centromeric and non-centromeric regions when compared to siNeg-transfected cells (Fig 7B and C). MCM2 depletion alone resulted in reduced centromeric CENP-A levels relative to control cells, consistent with previous observation for a role of MCM2 in centromeric association of CENP-A (Zasadzinska et al., 2018). Cells co-depleted for both MCM2 and DNAJC9 showed reduced levels of CENP-A at both centromeric and non-centromeric regions (Fig 7B 'and C)'' My first conclusion would be that MCM2 is upstream of DNAJC9's effects on CENP-A. From the results shown, I am not convinced that MCM2 contributes to CENP-A mislocalization in DNAJC9-depleted cells.

Our conclusion that MCM2 is responsible for CENP-A mislocalization upon DNAJC9 loss is supported by following complementary lines of strong evidence:

- MCM2 is highly enriched with soluble CENP-A upon DNAJC9 depletion - Figure 6C:

- CENP-A mislocalization induced by DNAJC9 depletion is rescued by the co-depletion of MCM2. Figure 8C:

In addition, we note that it is not possible for MCM2 to be upstream of DNAJC9 in the CENP-A deposition pathway as DNAJC9 does not bind CENP-A (Hammond *et al* 2021), also supported by our IP-MS datasets where we do not specifically enrich DNAJC9 with CENP-A (Fig EV5D, 6D). Our data demonstrate that MCM2 aberrantly binds soluble CENP-A upon DNAJC9 loss and that this is

required for CENP-A mislocalization. In our opinion, the most parsimonious conclusion to these findings is that MCM2 contributes to CENP-A mislocalization upon DNAJC9 loss.

See data in Hammond *et al.* 2021. DNAJC9 does not bind CENP-A:

11. Figure 8: the conclusions and the subtitle cannot be drawn from the data shown since the authors have not examined H3 at all and only conclude this from published data that were performed with a different experimental set up (for instance the CENP-A overexpression).

We appreciate the reviewer's point for additional data to support our conclusion that a blockage in H3-H4 supply chain provides an opportunity for CENP-A mislocalization. We could address this comment by examining the localization of CENP-A using mitotic chromosome spreads from control and H3-depleted cells. However, as noted above in comment #8, robust H3 depletion would require siRNA-mediated targeting of all forms of H3 (H3.1/2/3).

12. There are many questions that remain unanswered by the study, such as the function of DNAJC9 depletion on endogenous CENP-A and its implications in cancer, particularly when CENP-A is misexpressed. Without this information, our understanding of the molecular mechanisms of CIN in cancer cells and the role of replication in controlling CENP-A levels does not significantly increase from what we already know (Zasadzinska *et al.* 2018; Nechemia-Arbely *et al.*, 2019), especially if we don't know whether ectopically expressed CENP-A is chromatin incorporated or chromatin associated and whether the used construct is functional/rescues CENP-A depletion.

We agree with the reviewer that our work opens exciting new research avenues, and the translation of our findings to specific cancer drivers and types is one of them. However, we politely disagree that our study, that describes the functional and mechanistic basis of DNAJC9's role in protecting cells

from CENP-A mislocalization and CIN, has not gone beyond current state of the art. Neither Zasadzinska *et al.* 2018 and Nechemia-Arbely *et al.* 2019 describe the role of DNAJC9 in restricting CENP-A mislocalization or that this mislocalization is a consequence of loss of fidelity of H3–H4 supply pathways that drives more CENP-A-H4 into the chromosomal arms via the MCM2 pathway. We have addressed the functionality of the tagged version and incorporation into chromatin in responses to comments 1 and 2 above.

13. Their model is interesting but again experiments are missing to prove this hypothesis. The authors claim that they 'demonstrate that DNAJC9 prevents the promiscuous entry of CENP-A into H3-H4 deposition pathways to prevent CIN, which is a known hallmark of many cancers'. However, CENP-A can get ectopically incorporated without any deleterious effects on mitosis as long as it is taken out again in a timely fashion. Wouldn't be the model that DNAJC9 is involved in CENP-A's accurate removal?

We agree that the CENP-A mislocalization phenotype could be explained by either increased incorporation of CENP-A at ectopic sites or defects in the removal of mislocalized CENP-A. However, our high depth interactome analysis shows that DNAJC9 does not physically associate with CENP-A, consistent with previous findings (Hammond *et al.* 2021), arguing against a model where DNAJC9 plays a direct role in evicting mislocalized CENP-A.

14. It would also have been interesting to model CENP-A to the structure of DNAJC9, MCM2 and histones that has been already solved by some of the coauthors (Hammond et al 2021).

We have shown previously in Hammond *et al.* 2021 that DNAJC9 does not bind CENP-A.

15. Importantly, the authors may want to consider ways to address the physiological role of the interaction of CENP-A with DNAJC9 and MCM2. Is it similar to what has been proposed for H3-H4? As noted above, DNAJC9 does not bind CENP-A-H4. It has been shown previously that CENP-A–H4 is able to bind MCM2 (Huang *et al.* 2015).

16. The authors discuss CIN throughout the manuscript but do not address this extensively experimentally.

We emphasize that we have characterized the known readouts of CIN, namely micronuclei and mitotic segregation defects reported in our previous studies with CENP-A overexpressing cells (Shrestha *et al.* 2017, 2021). These CIN phenotypes have been assessed in the context of both endogenous (Fig 4G) and overexpressed levels of CENP-A (Fig 4E,F) upon DNAJC9 depletion.

17. There are data bases of different cancer entities that allow the correlation of DNAJC9 and MCM2 expression with CENP-A misexpression and perhaps there is even a cancer model where the authors

could test their hypothesis directly.

We thank the reviewer for this point. Since we propose that the loss of DNAJC9 function promotes CENP-A mislocalization, our hypothesis is ideally best tested in a cancer model overexpressing CENP-A with DNAJC9 downregulation. Translating our findings to a cancer model would require detailed analysis of histone chaperone expression levels and basal level of chromosomal instability relative to matched non-cancerous control groups. We believe that such analysis although relevant is currently beyond the scope of the study presented in this manuscript.

18. In summary, the authors report on an interesting finding, however, their conclusions are not backed up entirely by their experiments which seem in part premature.

We have elucidated using multiple orthogonal approaches that DNAJC9 restricts CENP-A-H4 mislocalization, and have identified a mechanism contributing to CENP-A-H4 mislocalization in the absence of DNAJC9 that implicates MCM2. Nevertheless, in response to the various concerns raised by the reviewer, we note that the revised manuscript will include experimental evidence key to reinforce the major finding presented in the study, i.e. genome-wide binding profile of CENP-A in siDNAJC9 and control cells using CENP-A CUT&RUN Sequencing. We recognize the reviewers request of extending our study to cancer models, but we feel that this goes beyond the scope of the current work and deserves in-depth exploration in a separate study.

Referee #3 (Report for Author)

This study by Balachandra et al., by employing a genome-wide RNAi screen, identified several regulators of CENP-A localisation, including DNAJC9, a member of the J-domain containing heat shock protein HSP40 family discovered recently as a co-chaperone for H3/H4. DNAJC9 depletion was found to mislocalise CENP-A and increase nuclear CENP-A throughout the cell cycle. In addition, DNAJC9-depleted cells exhibited chromosome instability (CIN) phenotypes, such as increased micronuclei and defective chromosome segregation. Interactome analysis revealed that DNAJC9 depletion disrupts the balance of histone chaperones, particularly affecting the association of CENP-A with MCM2, a protein involved in DNA replication and nucleosome assembly. Co-depletion of MCM2 and DNAJC9 reduces CENP-A mislocalization, suggesting MCM2's involvement in this process. Based on these key observations, the authors conclude that DNAJC9 is crucial for preventing CENP-A mislocalization and maintaining chromosome stability, and it does so by regulating the supply chain of histone H3-H4.

Overall, it is a solid work that convincingly highlights how disrupting the function of a DNAJC9, which is H3/H4 chaperone, can lead to CENP-A mislocalization and chromosome instability. However, the way DNAJC9's contribution is phrased in this study (including the title) appears to suggest an active role for DNAJC9 in preventing CENP-A mislocalization. In contrast, the data presented here indicate that DNAJC9's contribution to correct CENP-A localisation is rather indirect. We thank the reviewer for their positive appraisal of our work, and we will adjust the phrasing of the title to “DNAJC9 prevents CENP-A mislocalization and chromosomal instability by maintaining the fidelity of H3-H4 supply chains”.

Concerns:

1. While DNAJC9 depletion increases both centromeric and non-centromeric CENP-A levels, only the non-centromeric CENP-C levels are affected. It would be helpful if authors could explicitly comment on this.

We thank the reviewer for this point. We note that HeLa YFP-CENP-A Low cells expresses around 3-fold higher CENP-A levels (Fig EV1) in the context of endogenous levels of CENP-C. The fold increase in CENP-A levels at the non-centromeric regions (5.6-fold) is higher than at the centromeric region (1.3-fold; Fig 3A,B). Considering the stoichiometric nature of CENP-A-CENP-C interaction, the high CENP-A levels at the non-centromeric regions may promote more CENP-C localization at these sites (2-fold; Fig 4B) than at the centromeric region (0.96-fold; Fig 4B).

2. The interactome analysis shows that DNAJC9 depletion reduces the association of CENP-A deposition factors in the chromatin fraction: while the text mentions the reduction of Mis18a, figure 6C does not show Mis18a, only Mis18b, Mis18BP1 and HJURP are shown. It is important to clarify if Mis18a was seen and if not, the authors need to comment on this observation.

We thank the reviewer for pointing this out. We did observe a significant loss of MIS18B with chromatin bound CENP-A, however, in this cellular fraction we have insufficient observations to calculate any fold change data for MIS18A. We have now corrected MIS18A to MIS18B in the text and apologize for this mistake.

3. Experiments with DNAJC9 show that only the DNAJC9 J mutant (lacking enzymatic activity) increases CENP-A levels. Either the H3/H4 binding deficient mutant or the J mutant in combination with the H3/H4 binding defective mutant does not affect CENP -A levels. Possible explanations for these observations (why H3/H4 binding defective or the catalytically inactive J mutant, while in combination with H3/H4 binding defective mutant does not affect the CENP-A levels) are not adequately provided and need to be included in the revised manuscript. Details of the mutations (residue numbers) are not provided anywhere in the main text. This is crucial information and needs to be included in the revised manuscript.

We agree with the reviewer that it is important to explain our observation with each of the DNAJC9 mutants. Importantly, these data demonstrate that the DNAJC9 catalytic J mutant only has a dominant negative effect when DNAJC9 binds and traps the histone cargo in an unproductive intermediate. The experiment is conducted in the presence of endogenous DNAJC9; thus the histone binding mutants are not expected to have an effect. We will clarify this and add the details of the mutated residues in the revised manuscript.

Dr. Munira A. Basrai
NCI, NIH
Genetics Branch
41 Medlars Drive
Room B624
Bethesda, MD 20892

27th Nov 2023

Re: EMBOJ-2023-115742

The histone H3-H4 co-chaperone DNAJC9 prevents CENP-A mislocalization and chromosomal instability

Dear Munira,

Thank you for sending me a tentative point-by-point response and revision plan for your manuscript on the roles DNAJC9 and intact H3-H4 supply for preventing CENP-A mislocalization. I have now had a chance to consider your detailed answers, and I appreciate in particular your clarifications regarding the indirect function of DNAJC9, unrelated to CENP-A binding/chaperoning. I also realize that you appear to be in a good position to answer many other concerns of the referees, and especially the incorporation of the genome-wide CUT & RUN data should be able to address a key issue of referee 2. With regard to other points, I agree that a DNAJC9 overexpression experiment suggested by referee 1 may not by itself further help the conclusion, and also that additional extension into cancer models and CIN mechanisms would fall beyond the scope of the present study. The one experiment I would however still consider important would be attempting the H3.1-3 depletion for analyzing CENP-A on chromosome spreads, as discussed in your response #8 and #11 to referee 2.

I am therefore inviting you to prepare and resubmit a manuscript revised and extended along these lines - happily within a revision period extended beyond the default three months, during which competing manuscript published elsewhere would, as per our policies, have no effect on our final decision on your study. In any case, please do keep me updated about the progress of your revision work or in case of unexpected problems with some of the experiments, to discuss further proceedings.

I should remind you that our policy to allow only a single round of (major) revision makes it important to carefully revise and answer all points raised to the referees' satisfaction at this point. Finally, please note the detailed information and guidelines on how to prepare a revision below (and in our online Guide to Authors) - closely adhering to them shall greatly facilitate the editorial process at the time of resubmission.

Thank you again for the opportunity to consider this work, and I look forward to receiving your revision in due time.

With kind regards,

Hartmut

- size of the scale bars that are mandatory for all micrograph panels
- the statistical test used to generate error bars and P-values
- the type error bars (e.g., S.E.M., S.D.)
- the number (n) and nature (biological or technical replicate) of independent experiments underlying each data point
- Figures may not include error bars for experiments with $n < 3$; scatter plots showing individual data points should be used

instead.

9) Digital image enhancement is acceptable practice, as long as it accurately represents the original data and conforms to community standards. If a figure has been subjected to significant electronic manipulation, this must be clearly noted in the figure legend and/or the 'Materials and Methods' section. The editors reserve the right to request original versions of figures and the original images that were used to assemble the figure. Finally, we generally encourage uploading of numerical as well as gel/blot image source data; for details see: embopress.org/page/journal/14602075/authorguide#sourcedata

At EMBO Press, we ask authors to provide source data for the main manuscript figures. Our source data coordinator will contact you to discuss which figure panels we would need source data for and will also provide you with helpful tips on how to upload and organize the files.

In the interest of ensuring the conceptual advance provided by the work, we recommend submitting a revision within 3 months (25th Feb 2024). Please discuss the revision progress ahead of this time with the editor if you require more time to complete the revisions. Use the link below to submit your revision:

Link Not Available

EMBOJ-2023-115742:

DNAJC9 prevents CENP-A mislocalization and chromosomal instability by maintaining the fidelity of histone supply chains (Balachandra et al.)

Summary of revisions

We appreciate the comments from the reviewers and have addressed them in a point-by-point manner by performing additional experiments, incorporating editorial changes, and providing an explanation to their queries. In addition to editorial changes, we enlist below a list of new figures added to the manuscript and changes made to the figures.

List of new figures added in response to experimental revisions:

1. Fig 8 - Genome-wide analysis of CENP-A CUT&RUN Sequencing in RPE1-Tet-GFP-CENP-A cells with DNAJC9 depletion showing: A. Heat map of read density in two repeats; B. Karyoplot representation of all peaks identified in CENP-A CUT&RUN analysis averaged from two biological repeats; C. Pie chart of genomic features linked to the enriched peaks D. Heat map of enrichment analysis across known histone features; E. Overlap of read density with previously published ATAC-Seq dataset.
2. Fig EV4A, B – Metaphase chromosome spreads with high salt extraction (0.5 M NaCl) showing YFP-CENP-A intensities at centromeric and non-centromeric regions in HeLa YFP-CENP-A Low cells with DNAJC9 depletion.
3. Fig EV4 F,G - Metaphase chromosome spreads showing CENP-A intensities at centromeric and non-centromeric regions in HeLa YFP-CENP-A High cells with DNAJC9 depletion.
4. Fig EV4H – Chromosome segregation errors in DNAJC9-depleted HeLa YFP-CENP-A High cells
5. Fig EV9 – A. Western blot analysis of H3.3 depletion in HeLa YFP-CENP-A Low cells. B, C. Metaphase chromosome spreads showing CENP-A intensities at centromeric and non-centromeric regions in HeLa YFP-CENP-A Low cells with DNAJC9 depletion.
6. Fig EV10B – RT-qPCR analysis of CENP-A transcript levels in RPE1-Tet-GFP-CENP-A cells with and without depletion of DNAJC9.
7. Fig EV10 – D. Example of CENP-A profile on Chromosome 11; E. Karyoplot showing significantly enriched peaks in DNAJC9-depleted cells.

List of Figure/Table changes:

1. Fig 1D from primary siRNA screen replaced with new figure to show scale bar
2. Fig 2C changed to include representative image of G1, S and G2 nuclei. Fig 2D revised to show the number of cells analyzed per condition.
3. Fig 3 in original submission showed mitotic chromosome spreads in HeLa YFP-CENP-A Low, HeLa parental and RPE1-Tet-GFP-CENP-A. The chromosome spread data for HeLa parental and RPE1-Tet-GFP-CENP-A is now shown as Fig EV4 C, D and Fig EV10C, respectively. The new Fig 3 shows chromosome spread data for CENP-C, IF data for NUF2 and CIN phenotypes in HeLa YFP-CENP-A Low (Fig 3C-G), which was originally shown in Fig 4. Also, the CIN phenotypes in HeLa parental cells are now shown in Fig EV4E.
4. Fig 5A revised to include loading controls for each cellular fraction. Fig 5B,C revised to highlight the candidate chosen for in-depth studies i.e. MCM2.
5. Fig EV1 revised to show the full name of cell lines (HeLa, HeLa S3).
6. Fig EV3 revised to show representative figure for DAPI and EdU intensities in A and B, and bar chart showing the percent distribution of G1, S and G2 cells based on DAPI and EdU cell cycle binning. FACS profile of cell cycle shown as Fig EV4 in the original manuscript is now moved to Fig EV3D, E.
7. Table S1 has been revised to include two additional columns for each gene from the primary siRNA screen – 1. Number of siRNA tested per gene; 2. Number of siRNA treatments that led to YFP-CENP-A signal intensity greater than 2 fold.
8. Table S4 has been revised to update figure references and manuscript title

EMBOJ-2023-115742:

DNAJC9 prevents CENP-A mislocalization and chromosomal instability by maintaining the fidelity of histone supply chains (Balachandra et al.)

Response to Reviewers' comments indicated in blue font.

Referee #1 (Report for Author)

General summary and opinion:

In the present study, Balachandra *et al.* investigate factors that may be critical regulators of the localization of the centromeric variant of histone H3, CENP-A. For this, they performed a genome-wide RNAi screen followed by an image-based analysis of nuclear CENP-A levels, as a readout for mislocalized CENP-A which becomes enriched in non-centromeric chromatin. Amongst top candidates whose depletion leads to increased nuclear CENP-A intensity, many function in regulating deposition of other H3 variants, which supports the idea that H3-H4 homeostasis is key for preventing CENP-A mislocalization. They notably identify CHAF1A/B, consistent with their previous findings (Shrestha et al., 2019), and DNAJC9 (recently discovered as a co-chaperone for H3.1/2/3-H4). Then, they further investigate the importance of DNAJC9 in regulating the localization of CENP-A by analyzing metaphase chromosome spreads. Depleting DNAJC9 or targeting its J-domain (required for proper folding of H3-H4) confirmed that DNAJC9 is critical to regulate CENP-A localization in HeLa and RPE-1 cell lines. The targeting of DNAJC9 and subsequent CENP-A mislocalization lead to increased CIN. In addition, while previous work showed that DNAJC9 depletion leads to reduced load of MCM2 associated with histones H4 and H3.1 (Hammond et al., 2021), the interactome analysis of Balachandra et al. shows an enhanced interaction of CENP-A with MCM2 in DNAJC9-depleted cells. Furthermore, co-depletion of MCM2 and DNAJC9 could suppress CENP-A mislocalization. Therefore, they concluded that DNAJC9 prevents the mislocalization of CENP-A and CIN, and that MCM2 could contribute to recycling non-centromeric CENP-A.

Interestingly, overexpression of CENP-A is a common feature of many cancers (reviewed in Renaud-Pageot et al., 2022). Although we know that p53 contributes to the regulation of CENP-A expression (Filipescu et al., 2017), TCGA PanCancer Atlas studies (accessible on CBioPortal) show that both p53-defective and p53 wild type (p53-WT) cancers can display high levels of CENP-A, indicating additional levels of control that should be considered. Moreover, studies in human cell lines showed that overexpression of CENP-A leads to its association with other H3 chaperones (in addition to HJURP) and mislocalization at non-centromeric regions (Lacoste et al., 2014), which causes chromosomal instability (CIN) (Shrestha et al., 2017, 2021). Given that CIN is a hallmark of cancer that is strongly linked to tumor evolution, understanding factors that can be involved in CENP-A mislocalization is an important question that could help in the prognosis and treatment of CENP-A overexpressing cancers. This includes factors regulating CENP-A expression (in terms of levels and cell-cycle regulation), along with factors that could promote and prevent CENP-A mislocalization per se. Thus, in this context, the data presented here is relevant for the field.

In summary, this manuscript represents an interesting study that would deserve publication. However, to make a strong contribution, this would require addressing a number of points that we have outlined below, including suggestions for how, in particular, authors' conclusions could be better substantiated.

We thank the reviewer for their insightful summary of our work and for highlighting the relevance to the field. We appreciate the nature of the constructive critique which is aimed at improving our manuscript and we will address the points as outlined below.

Comments:

Authors propose that DNAJC9 "prevents CENP-A mislocalization and CIN". They also mention that high levels of CENP-A alone contribute to its mislocalization in a dose-dependent manner, and this is why they chose the "YFP-CENP-ALow" cell line (with 3-fold higher exogenous CENP-A compared to endogenous levels in the parental cell line) and not the "YFP-CENP-AHigh" (17-fold higher). Of note, CENP-A levels in patient tumors can go over 1000-fold relative to healthy tissue (Jeffery et al., 2021). Thus, to strengthen their data, additional experiments could:

1. Test whether, following DNAJC9 depletion in the context of CENP-A High, they can detect increased ectopic CENP-A levels, and/or worsened CIN phenotype.

As suggested by the reviewer, we examined the localization of CENP-A in metaphase chromosomes, and observed that the non-centromeric CENP-A levels were significantly higher (2-fold) in DNAJC9-depleted HeLa YFP-CENP-A High cells (Fig EV4 F,G; revised manuscript). We also scored for chromosome segregation errors in HeLa YFP-CENP-A High cells following DNAJC9 depletion and were able to detect a moderate but significant increase in chromosome segregation errors upon DNAJC9 depletion (Fig EV4 H). These results have been included in page #11 of the revised manuscript.

2. Assess whether DNAJC9 co-overexpression with CENP-A would be sufficient to counteract CENP-A mislocalization (e.g., in the CENP-A High cell line).

The reviewer suggests that CENP-A mislocalization could be suppressed in DNAJC9 overexpressing cells. This might hold true if overexpression of DNAJC9 alone would be sufficient to increase the supply of H3-H4. Although the effect of DNAJC9 overexpression on CENP-A mislocalization could be tested, the interpretation of this data would rely on determining whether H3-H4 supply is increased in this case. Since a negative result is also compatible with the proposed model that DNAJC9 maintains the supply of H3-H4 and this prevents mislocalization of CENP-A (Fig. 9), we have not performed this experiment.

3. CENP-A mislocalization induced by depletion of DNAJC9 in HeLa and RPE-1 cells could be due to an overexpression of CENP-A. The RT-qPCR performed for HeLa YFP-CENP-A Low is a nice confirmation that CENP-A is unaffected, at the transcription level, by DNAJC9 and/or MCM2.

- Ideally, the same analysis should be performed in the RPE-1 cell line too to reinforce the point

We agree with the reviewer and have now included RT-qPCR data showing that *CENP-A* transcript levels are not significantly altered in DNAJC9-depleted RPE1-Tet-GFP-CENP-A cells with DOX treatment (Fig EV10B and page #18 of revised manuscript).

4. Figure EV8 could be titled "CENP-A mRNA levels are unaffected by depletion of DNAJC9, MCM2 or both in HeLa YFP Low."

We agree with the reviewer and have changed the title to state that "CENP-A mRNA levels are unaffected by depletion or co-depletion of DNAJC9 and MCM2 in HeLa YFP-CENP-A Low cells".

5. Authors assessed total CENP-A levels in the different HeLa cell lines by Western Blot (Fig. EV1). Ideally, the siDNAJC9 condition should be added for all, to compare CENP-A protein levels in all conditions as a control throughout their experiments.

The Western blot shown in Fig EV1 aims to compare the steady state levels of CENP-A in different cell lines, based on which HeLa YFP-CENP-A low cell line was chosen for in-depth studies. We have shown the effects of DNAJC9 depletion on CENP-A levels in different cell lines in the following figures: the HeLa CENP-A-YFP-Low cell line (Fig 4D), HeLa S3 CENP-A-Flag-HA (Fig 5A) and RPE1-Tet-GFP-CENP-A (Fig EV10A).

6. In the abstract page 2, first sentence: "... CENP-A is overexpressed and mislocalized in many cancers." Although CENP-A is frequently overexpressed in cancer, its mislocalization has actually only been documented in cell lines. Since this has not been formally shown in patient tumors, a little caution is required here. The sentence should be rephrased and could highlight this important link between CENP-A overexpression and mislocalization in cell lines, along with the work of Verrelle *et al.* (2021), which gives hints in terms of changes in CENP-A subnuclear distribution in normal and tumoral tissues.

We thank the reviewer for this suggestion. The first sentence of the abstract page 2 has been modified to "The centromeric histone H3 variant CENP-A is overexpressed in many cancers." The work of Verrelle *et al.* (2021) highlighting changes in the subnuclear distribution of CENP-A in cancer tissues has been added in the second paragraph of the introduction (Page #5) instead of the abstract due to word limit.

7. In the Introduction page 3, line 6: "The evolutionary conserved histone 3 variant CENP-A". While the essential functions of CENP-A are largely conserved, analysis of CENP-A homologs from several species shows a divergent ("fast-evolving") N-terminal tail. The sentence should be rephrased and refer to the function of CENP-A.

We thank the reviewer for this comment. To account for the rapidly evolving N terminal tail of CENP-A, line 6 of the Introduction on page #3 has been modified to remove the general evolutionary conservation of CENP-A.

8. Page 4, starting from line 8: "Preventing the mislocalization of overexpressed CENP-A by depleting the H3.3 chaperone DAXX (Lacoste *et al.*, 2014) rescued the invasiveness and CIN phenotypes of CENP-A overexpressing cells..." The sentence should be rephrased: the reference is correct regarding the use of DAXX depletion to reestablish centromeric localization of CENP-A, but Lacoste *et al.* did not assess cell invasiveness. Of note, depletion of the HIRA complex has also been proposed to prevent CENP-A mislocalization (Nye *et al.*, 2018) and could be cited too.

The reviewer is correct that Lacoste *et al.* (2014) did not show rescue of invasiveness upon depletion of DAXX. This is reflected in our statement that separates the citations for suppression of mislocalization and suppression of invasiveness by DAXX as noted on page #4, "The mislocalization of overexpressed CENP-A can be prevented by depleting the H3.3-H4 chaperone DAXX (Lacoste *et al.*, 2014) and this reduced the invasiveness of cancer cells and rescued CIN phenotypes (Shrestha *et al.*, 2017; Shrestha *et al.*, 2021)."

We wish to note that the Nye *et al.* (2018) study reports that the H3.3 chaperone HIRA promotes ectopic localization of CENP-A in the context of HJURP or DAXX depletion in SW480 cells. However, depletion of HIRA alone enhances the ectopic localization of CENP-A (Fig 1 B and C of (Nye *et al.*, 2018)), as reported with HIRA depletion in HeLa cells (Lacoste *et al.*, 2014; Shrestha *et al.*, 2023) and

earlier observation in the budding yeast (Ciftci-Yilmaz *et al*, 2018). We have cited these papers describing the increased CENP-A mislocalization upon HIRA depletion alone in the revised manuscript.

9. Fig 1D: Images of DNAJC9 staining should be added.

Fig 1D is a representative image from the primary siRNA screen in which DNAJC9 staining was not performed, and hence cannot be included. To demonstrate the efficiency of DNAJC9 depletion in HeLa CENP-A-YFP-High cells (cell line used in the screen) using immunofluorescence (IF), we have now included representative IF images of control or DNAJC9 depleted cells showing DNAJC9 staining (Fig EV 4H) in the revised manuscript.

10. Fig 2C/D: To confirm that non-centromeric CENP-A signal can be detected at all cell cycle stages, authors relied on an image-based analysis of DAPI and EdU signals to detect G1/S/G2 cell populations. - A representative image for each population should be added.

Representative image for each population has been added in revised Fig 2C.

11. In addition, figure legend indicates N as the number of cells analyzed per condition. This number does not appear on fig. 2D and should be added.

We thank the reviewer for pointing out the omission of the number of cells in the figure. The revised Fig 2D includes the number of cells analyzed per condition.

12. Fig 3E: Fig should also show endogenous CENP-A on membrane at 17kDa.

We appreciate the reviewer's suggestion to show endogenous CENP-A for Fig 3E (Revised Fig EV10A). However, we have not included this panel as we find that the levels of endogenous CENP-A are barely detectable on the western blot upon DOX-induced overexpression of exogenous GFP-CENP-A and long exposure shows high background. This is consistent with previous observations that report downregulation of endogenous CENP-A in the presence of ectopically expressed CENP-A in RPE1 (Swartz *et al*, 2019) and HeLa (Lacoste *et al.*, 2014) cells.

13. Fig6A (Western Blot of soluble and chromatin fractions): In addition to Memcode staining, Western should show appropriate loading controls for each cell fraction.

We agree with the reviewer about the loading controls and the revised figure has loading control for both soluble (alpha-tubulin) and chromatin (histone H4) fractions (see Figure 5A).

14. Fig6B/C: The factors that will further be tested in the study could be highlighted.

We agree with the reviewer. The factor pursued for in-depth studies (MCM2) has been highlighted in the revised Fig 5 B/C.

15. Page 16, line 20: "Controls included untransfected cells and cells expressing full-length DNAJC9 (WT)" The sentence may need to be rephrased as "Controls included untransfected cells and cells transfected with a full-length DNAJC9 (WT) vector.".

We thank the reviewer for pointing this out, we have revised this section in the manuscript.

16. Page 17, authors argue that a blockade in the H3-H4 supply chain promotes the mislocalization of CENP-A (based on results showed in figure 8 and the impact of depleting DNAJC9 in general). This is

indeed an interesting observation. However, one could expect the H3-H4 assembly pathway to be affected by a mutation in the HBD (or both domains) of DNAJC9. Based on results in figure 8, this is not the case (probably thanks to the endogenous DNAJC9 context), and it would be useful to comment on this result. We agree that the phenotypes of the histone-binding mutants should be discussed in the results section. As pointed out by the reviewer, the histone-binding mutants likely do not exhibit a dominant negative effect in the context of endogenous DNAJC9 function. The revised manuscript includes this explanation on page #17.

17. Fig 9: Third section of the scheme shows J mutant of DNAJC9 bound to misfolded H3, associated with the chromosome. For clarity purposes, when describing their model, the authors could reiterate that J mutant of DNAJC9 becomes trapped on chromatin through spurious interactions of misfolded histones with DNA (Hammond et al., 2021).

The model in Figure 9 has been changed to highlight the major findings of this study in the revised manuscript and does not include the section on the J mutant.

18. p53 is involved in the regulation of CENP-A expression (Filipescu et al., 2017) and a key determinant of how CENP-A impacts cell state and therapeutic response. The manuscript could thus mention the p53 status of the cells used for the analysis (HPV18 inhibited for HeLa and WT for RPE-1) and comment how DNAJC9 seems to prevent the mislocalization in both p53 backgrounds.

We have now included observations from previous studies regarding the role of p53 in CENP-A regulation (Filipescu *et al*, 2017) and commented on the p53 status in the cell lines used to study CENP-A mislocalization upon DNAJC9 depletion. Please refer to the discussion section of the revised manuscript on page #22.

19. Previous work showed that CENP-A nucleosome particles, homotypic at the centromere, involve a heterotypic tetramer containing CENP-A and H3.3 at ectopic positions (Lacoste *et al.*, 2014). Based on their results, Balachandra et al. argue that a blockade in the H3-H4 supply chain could promote aberrant deposition of CENP-A. Ideally, authors could comment further on the type of particle that could be expected at ectopic position in the context of a shortage of H3-H4.

The reviewer brings up an interesting point about the composition of the nucleosome. Our interactome data shows H3.3 enriched with soluble CENP-A upon DNAJC9 depletion but not on chromatin. In-depth studies in the future would be required to decipher the exact composition of the ectopic CENP-A nucleosomes in the context of DNAJC9 depletion, hence, we prefer not to comment on this based on our current data.

20. Fig EV1: For clarity purposes, the figure could show the full name of cell lines (HeLa, HeLa S3 or RPE-1).

Fig EV1 has been revised to show the full name of cell lines.

21. Table S3: siRNA sequences are not aligned and part of them cut in the pdf (but the excel format is fine).

The excel format for Table S3 is included as supplementary data.

Referee #2 (Report for Author)

In the study titled "The histone H3-H4 co-chaperone DNAJC9 prevents CENP-A mislocalization and chromosomal instability" by Balachandra et al, the authors utilized a genome-wide RNAi screen in cells overexpressing CENP-A. Their primary objective was to identify proteins critical for preventing CENP-A mislocalization, using image-based analysis as their initial tool. Among the proteins screened, DNAJC9 emerged as a noteworthy factor influencing the localization of exogenous CENP-A. Cells subjected to DNAJC9 depletion exhibited an enrichment of CENP-A on chromatin, CENP-A mislocalization, and a mild chromosomal instability (CIN) phenotypes. An interactome analysis showed that CENP-A interacts more with the H3-H4 chaperone MCM2 in DNAJC9-depleted cells and that co-depletion of MCM2 and DNAJC9 suppressed CENP-A mislocalization.

1. The manuscript is based on an interesting genome-wide RNA screen with many interesting findings that the authors may choose to work on in the future. In this study they decided to analyze DNAJC9. There is however, one fundamental aspect concerns the screen and that is partially addressed in Figure EV1A. In this figure, the authors presented whole-cell extract Western blots. However, it is essential to confirm whether CENP-A is genuinely incorporated into chromatin. They could do it for instance using some biochemical techniques such as chromatin salt extraction. The chromosome spreads featured in the later figures and the cell fractionation (chromatin/soluble) are insufficient to determine a genuine incorporation of exogenous CENP-A.

We agree with the reviewer's comment for additional data to show that exogenous CENP-A can stably bind to the chromatin and that high salt extraction techniques would be required to show this. Hence, we analyzed metaphase chromosome spreads prepared from HeLa YFP-CENP-A Low cells pretreated with high salt buffer (0.5 M NaCl) for 20 minutes. This is based on a previous study in *Drosophila* cells that showed CENP-A stably bound to the chromatin can resist high salt (500 mM NaCl) extraction compared to weakly associated CENP-A (Bobkov *et al*, 2018). Immunostaining was done with anti-GFP antibody to exclusively detect the exogenous YFP-CENP-A along with CENP-C co-staining to define the centromeric region. Our data showed that YFP-CENP-A mainly localized at the centromere, with minimal non-centromeric localization in control (siNeg) cells. Depletion of DNAJC9 resulted in higher YFP-CENP-A levels at the non-centromeric region (4.3-fold increase) relative to siNeg, suggesting that the mislocalized exogenous CENP-A is stably associated with the non-centromeric chromatin under high salt conditions. These results have been added to the revised manuscript in Fig EV4 A and B.

To further support our results from cell fractionation and chromosome spread experiments showing mislocalization of CENP-A, we examined the genome-wide binding profile of CENP-A in siDNAJC9 and siNeg cells using CENP-A CUT&RUN Seq. These experiments were done using stable near diploid model RPE1-Tet-GFP-CENP-A induced with Doxycycline which shows mislocalization of CENP-A on chromosome spreads upon DNAJC9 depletion (Fig EV10C in revised manuscript). The rationale for using RPE1 cells as we stated in our first submission is to support the cell biology and biochemical data from HeLa cells in an independent model which is not aneuploid. The CUT&RUN Seq data with two independent repeats showed increased localization of CENP-A at non-centromeric regions genome-wide in siDNAJC9 cells. We identified 17,341 significantly enriched non-centromeric peaks in both replicates of siDNAJC9-treated cells when compared to siNeg control. The new data (Fig 8 and Fig EV10D,E in the

revised manuscript) further strengthens our conclusion that DNAJC9 depletion contributes to mislocalization of CENP-A to non-centromeric regions.

2. If CENP-A does not get incorporated the entire screen and the function of DNAJC9 in CENP-A localization needs to be questions. In this context, it is also important to address whether the tagged version of CENP-A rescues endogenous CENP-A depletion phenotypes.

Please refer to our response to comment #1 above for new data showing: 1. Stable binding of exogenous YFP-CENP-A to non-centromeric regions using chromosome spreads with and without DNAJC9 depletion (Fig EV4A,B); and 2. Genome-wide localization of CENP-A in cells depleted for DNAJC9 using CENP-A CUT&RUN Seq (Fig 8 and Fig EV10D,E in the revised manuscript).

For the functionality of the tagged version of CENP-A, there are several published studies (Black *et al*, 2004; Fachinetti *et al*, 2013; Foltz *et al*, 2009; Kops *et al*, 2004; Nechemia-Arbely *et al*, 2017) that have used the same construct (YFP-CENP-A) and importantly, the tagged version of CENP-A used in our studies has been shown to localize to the centromere and support long term cell viability (Nechemia-Arbely *et al.*, 2017).

3. I assume that CENP-A is also in the genome wide screen, is then the tagged version designed in a way that it does not get degraded by the siRNA used for CENP-A?

The tagged version (YFP-CENP-A) is not siRNA resistant and can be targeted by CENP-A siRNA which is included in the genome-wide screen.

4. Figure 1D does not seem to be the same magnifications and there is no scale bar shown on the figure. Furthermore, the authors should explicitly state whether all three siRNAs targeting candidate genes yielded similar phenotypes. Table 1 lacks clarity and it remains unclear which parts represent different aspects of the study. Especially, the authors should highlight, which candidates only exhibited CENP-A localization effects with only one or two siRNAs. Generally, the extended data may be presented in a more user-friendly manner.

We thank the reviewer for pointing the omission of the scale bar in Fig 1D and details in Table 1. Figure 1D is revised to include the scale bar. The revised Table 1 includes additional columns showing the number of siRNAs tested and the number of siRNAs showing CENP-A signal > 2-folds.

5. Figure 2: How the depletion of DNAJC9 impact the cell cycle is important and needs more carefully analysis for reasons the authors mention in the manuscript (Nechemia-Arbely et al, 2019). In EV3 b/c quantifications are missing and should be included especially because it looks like there is a shift towards G2/M.

As requested by the reviewer we have now included quantifications for the EdU-based cell cycle binning in Fig. EV3 C in the revised manuscript. We have also performed FACS analysis of propidium iodide-stained asynchronous cells with and without DNAJC9 depletion (Fig EV3 D,E). Importantly, both these analyses revealed non-significant changes in the cell cycle profile upon DNAJC9 depletion. Furthermore, results from Figure 2D show that cells depleted for DNAJC9 exhibit increased nuclear CENP-A levels at G1, S and G2 stages.

6. Figure 3F: Previous publications on CENP-A overexpression reported an ectopic localization with clear foci forming at certain chromatin domains. Here, the ectopic signal is very uniform as a haze over all of

the chromatin (and the surrounding). How do the authors explain this difference?

The nature of ectopic CENP-A signal depends on the context that promotes mislocalization. Upon depletion of an H3-H4 chaperone such as DNAJC9, a genome-wide broad distribution of CENP-A is expected, which appears as a uniform signal on mitotic chromosome spreads as shown in Fig 3F (Fig EV10C in revised manuscript). We do not observe signal outside the DAPI stained region and our quantifications include background correction to exclude signal outside the chromatin region. In addition, our CUT&RUN data (Fig 8) showing broad distribution of the non-centromeric peaks enriched in siDNAJC9 condition complements the uniform pattern of ectopic CENP-A signal observed in our chromosome spreads. We note that previous studies report foci-like signal of ectopic CENP-A under unperturbed conditions where specific gene depletions were not performed (Athwal *et al*, 2015; Lacoste *et al.*, 2014).

7. In Figure 4, the increase in mitotic defects seems minimal, it also seems very low in HeLa cells in general, 'wt' HeLa usually have more than the reported percentage of missegregation.

Our results with parental HeLa cells exhibiting a relatively low range of CIN phenotypes (micronuclei, chromosome segregation defect) (Figure EV4E) are similar to those reported in our previous studies (Shrestha *et al*, 2017; Shrestha *et al.*, 2023). This is consistent with other studies for low levels of CIN in HeLa parental cells (Piette *et al*, 2021). In Figure 3F and G, we presented data showing significant elevation of CIN phenotypes in HeLa YFP-CENP-A Low cells upon depletion of DNAJC9. Similar to results with CHAF1B depletion (Shrestha *et al.*, 2023), the low levels of CENP-A mislocalization upon DNAJC9 depletion in the parental HeLa cell line (2.7-fold; Fig EV4C,D) are insufficient to induce CIN phenotypes (Fig EV4E). We hope this clarifies the reviewer's concern.

8. Figure 5 D: The authors have taken H2B as loading control for chromatin, which is fine. However, according to their model, H3-H4 is misfolded in DNAJC9 depleted cells and presumably then reduced to make space for more CENP-A (as visible on their blot). This hypothesis should be tested by for instance western blotting.

The reviewer raises a good point. The concern with western blot analysis of chromatin-bound H3 is that misfolded H3-H4 can spuriously interact with DNA (Hammond *et al*, 2021), and western blots would not be able to distinguish between productive and non-productive association of H3-H4 with the chromatin. Alternatively, this hypothesis could be tested by examining the localization of CENP-A using mitotic chromosome spreads from control and H3-depleted cells. To address this, we have now included data in the revised manuscript showing that H3.3-depleted cells exhibit higher levels of non-centromeric CENP-A (Fig EV9 and page #17 of revised manuscript). Also, please see response to comment #11 below.

9. The authors show in 5D that endogenous levels of CENP-A are also stabilized in DNAJC9 depleted cells, which is a very interesting finding because it suggests that no misexpression/mutation of CENP-A is required for more CENP-A. Would this suffice to see mitotic errors? In other words, are there mitotic errors in DNAJC9 depleted cells without exogenous CENP-A and could this be rescued by a reduction of CENP-A (for instance a partial depletion)? This would help to come up with nice hypothesis for cancer cells with excess CENP-A.

We have reported CIN phenotypes in HeLa parental cells upon depletion of DNAJC9 in Figure EV4E. Our results do not show a significant increase in mitotic errors in these cells as explained in response to comment #7 above.

10. In Figure 7, I cannot follow their conclusion that enhanced interaction of MCM2 with CENP-A contributes to CENP-A mislocalization in DNAJC9-depleted cells. They show that increased CENP-A levels in siDNAJC9.3-transfected cells at both centromeric and non-centromeric regions when compared to siNeg-transfected cells (Fig 7B and C). MCM2 depletion alone resulted in reduced centromeric CENP-A levels relative to control cells, consistent with previous observation for a role of MCM2 in centromeric association of CENP-A (Zasadzinska et al., 2018). Cells co-depleted for both MCM2 and DNAJC9 showed reduced levels of CENP-A at both centromeric and non-centromeric regions (Fig 7B and C). My first conclusion would be that MCM2 is upstream of DNAJC9's effects on CENP-A. From the results shown, I am not convinced that MCM2 contributes to CENP-A mislocalization in DNAJC9-depleted cells.

Our conclusion that MCM2 is responsible for CENP-A mislocalization upon DNAJC9 loss is supported by the following complementary lines of strong evidence:

1) MCM2 is highly enriched with soluble CENP-A upon DNAJC9 depletion - Figure 5C:

We have further highlighted MCM2 in the revised panel:

2) CENP-A mislocalization induced by DNAJC9 depletion is rescued by the co-depletion of MCM2. Figure 6C:

In addition, we note that it is not possible for MCM2 to be upstream of DNAJC9 in the CENP-A deposition pathway as DNAJC9 does not bind CENP-A (Hammond *et al.*, 2021), also supported by our IP-MS datasets where we do not specifically enrich DNAJC9 with CENP-A (Fig EV5D, 6D). Our data demonstrate that MCM2 aberrantly binds soluble CENP-A upon DNAJC9 loss and that this is required for CENP-A mislocalization. In our opinion, the most parsimonious conclusion to these findings is that MCM2 contributes to CENP-A mislocalization upon DNAJC9 loss.

See data in (Hammond *et al.*, 2021), DNAJC9 does not bind CENP-A:

11. Figure 8: the conclusions and the subtitle cannot be drawn from the data shown since the authors have not examined H3 at all and only conclude this from published data that were performed with a different experimental set up (for instance the CENP-A overexpression).

We appreciate the reviewer's point for additional data to support our conclusion that a blockage in H3-H4 supply chain provides an opportunity for CENP-A mislocalization. DNAJC9 is a co-chaperone for the major isoforms of H3 (H3.1/2/3) (Hammond *et al.*, 2021), therefore blockage of H3-H4 supply would require robust H3 depletion through siRNA-mediated targeting of these H3 isoforms. As efficient

depletion of the canonical H3.1/2 isoforms might be less efficient due to their cellular abundance, we tested our hypothesis in cells transfected with siRNA targeting one of the two H3.3-encoding genes (*H3F3A*). Even with a partial reduction of total H3.3 levels based on western blot analysis (Fig EV9A), we observed higher non-centromeric CENP-A levels (2.66-fold) in siH3F3A-transfected cells with non-significant changes in centromeric CENP-A levels (1.01-fold; Fig EV9B, C). This data, included in page #17 of revised manuscript, strengthens our model that defective H3-H4 supply contributes to CENP-A mislocalization to non-centromeric regions.

12. There are many questions that remain unanswered by the study, such as the function of DNAJC9 depletion on endogenous CENP-A and its implications in cancer, particularly when CENP-A is misexpressed. Without this information, our understanding of the molecular mechanisms of CIN in cancer cells and the role of replication in controlling CENP-A levels does not significantly increase from what we already know (Zasadzinska et al 2018; Nechemia-Arbely et al, 2019), especially if we don't know whether ectopically expressed CENP-A is chromatin incorporated or chromatin associated and whether the used construct is functional/rescues CENP-A depletion.

We agree with the reviewer that our work opens exciting new research avenues, and the translation of our findings to specific cancer drivers and types is one of them. However, we politely disagree that our study, which describes the functional and mechanistic basis of DNAJC9's role in protecting cells from CENP-A mislocalization and CIN, has not gone beyond the current state of the art. Neither Zasadzinska *et al.* 2018 nor Nechemia-Arbely *et al.* 2019 describe the role of DNAJC9 in restricting CENP-A mislocalization and that this mislocalization is a consequence of loss of fidelity of H3–H4 supply pathways that drive more CENP-A into the chromosomal arms via the MCM2 pathway. We have addressed the functionality of the tagged version and incorporation into chromatin in responses to comments #1 and #2 above.

13. Their model is interesting but again experiments are missing to prove this hypothesis. The authors claim that they 'demonstrate that DNAJC9 prevents the promiscuous entry of CENP-A into H3-H4 deposition pathways to prevent CIN, which is a known hallmark of many cancers'. However, CENP-A can get ectopically incorporated without any deleterious effects on mitosis as long as it is taken out again in a timely fashion. Wouldn't be the model than that DNAJC9 is involved in CENP-A's accurate removal? As suggested by the reviewer the CENP-A mislocalization phenotype could be explained by either increased incorporation of CENP-A at ectopic sites or defects in the removal of mislocalized CENP-A. However, our high depth interactome analysis shows that DNAJC9 does not physically associate with CENP-A consistent with previous findings (Hammond *et al.*, 2021), arguing against a model where DNAJC9 plays a direct role in evicting mislocalized CENP-A.

14. It would also have been interesting to model CENP-A to the structure of DNAJC9, MCM2 and histones that has been already solved by some of the coauthors (Hammond et al 2021). We have shown previously in (Hammond *et al.*, 2021) that DNAJC9 does not bind CENP-A.

15. Importantly, the authors may want to consider ways to address the physiological role of the interaction of CENP-A with DNAJC9 and MCM2. Is it similar to what has been proposed for H3-H4? As noted above in comment #14, DNAJC9 does not bind CENP-A-H4. It has been shown previously that CENP-A–H4 can bind MCM2 (Huang *et al.*, 2015).

16. The authors discuss CIN throughout the manuscript but do not address this extensively experimentally.

We have characterized and described the known readouts of CIN, namely micronuclei and mitotic segregation defects reported in our previous studies with CENP-A overexpressing cells (Shrestha *et al.*, 2017; Shrestha *et al.*, 2021). The CIN phenotypes were assessed in the context of both endogenous (Fig EV4E) and moderately overexpressed levels of CENP-A (Fig 3F,G) upon DNAJC9 depletion. The revised manuscript also includes new data to show that DNAJC9-depleted cells show increased mitotic chromosome segregation errors in HeLa YFP-CENP-A High cells in the context of mislocalized CENP-A (Fig EV4 F-H). In-depth characterization of CIN phenotypes using techniques such as SKY (Spectral Karyotyping) or WGS (Whole Genome Sequencing) will be pursued in future in the near diploid RPE1 cell line model.

17. There are data bases of different cancer entities that allow the correlation of DNAJC9 and MCM2 expression with CENP-A misexpression and perhaps there is even a cancer model where the authors could test their hypothesis directly.

We agree with the reviewer for examining our hypothesis in a cancer model. We propose that the loss of DNAJC9 function promotes CENP-A mislocalization. Our hypothesis is thus ideally best tested in a cancer model with CENP-A overexpression and downregulation of DNAJC9. Therefore, translating our findings to a cancer model would require detailed characterization of histone chaperone expression levels and basal level of chromosomal instability relative to matched non-cancerous control groups in those models. We believe that such analysis although relevant is currently beyond the scope of the study presented in this manuscript and deserves in-depth exploration in a separate study.

18. In summary, the authors report on an interesting finding, however, their conclusions are not backed up entirely by their experiments which seem in part premature.

We have used multiple orthogonal approaches to define a novel role for DNAJC9 in preventing CENP-A mislocalization. Our results highlight the importance of H3-H4 supply and the contribution of MCM2 in mislocalization of CENP-A in cells depleted for DNAJC9. Furthermore, in response to the various concerns raised by the reviewer, the revised manuscript includes additional experimental evidence key to reinforce the major findings presented in the study. These include: 1. Stable binding of exogenous YFP-CENP-A to the chromatin with and without DNAJC9 depletion under high salt conditions (Fig EV4A,B); 2. Genome-wide study for chromosomal localization of CENP-A in RPE1 cells with siDNAJC9 and control cells using CENP-A CUT&RUN Sequencing (Fig 8); and 3. Mislocalization of CENP-A in H3.3-depleted cells (Fig EV9).

Referee #3 (Report for Author)

This study by Balachandra et al., by employing a genome-wide RNAi screen, identified several regulators of CENP-A localisation, including DNAJC9, a member of the J-domain containing heat shock protein HSP40 family discovered recently as a co-chaperone for H3/H4. DNAJC9 depletion was found to mislocalise CENP-A and increase nuclear CENP-A throughout the cell cycle. In addition, DNAJC9-depleted cells exhibited chromosome instability (CIN) phenotypes, such as increased micronuclei and defective chromosome segregation. Interactome analysis revealed that DNAJC9 depletion disrupts the balance of histone chaperones, particularly affecting the association of CENP-A with MCM2, a protein involved in DNA replication and nucleosome assembly. Co-depletion of MCM2 and DNAJC9 reduces CENP-A mislocalization, suggesting MCM2's involvement in this process. Based on these key observations, the authors conclude that DNAJC9 is crucial for preventing CENP-A mislocalization and maintaining chromosome stability, and it does so by regulating the supply chain of histone H3-H4.

Overall, it is a solid work that convincingly highlights how disrupting the function of a DNAJC9, which is H3/H4 chaperone, can lead to CENP-A mislocalization and chromosome instability. However, the way DNAJC9's contribution is phrased in this study (including the title) appears to suggest an active role for DNAJC9 in preventing CENP-A mislocalization. In contrast, the data presented here indicate that DNAJC9's contribution to correct CENP-A localisation is rather indirect.

We thank the reviewer for the positive appraisal of our work. As suggested, we have revised the manuscript title to - "DNAJC9 prevents CENP-A mislocalization and chromosomal instability by maintaining the fidelity of histone supply chains".

Concerns:

1. While DNAJC9 depletion increases both centromeric and non-centromeric CENP-A levels, only the non-centromeric CENP-C levels are affected. It would be helpful if authors could explicitly comment on this.

We thank the reviewer for this point. We note that HeLa YFP-CENP-A Low cells express around 3-fold higher CENP-A levels (Fig EV1) in the context of endogenous levels of CENP-C. Upon DNAJC9 depletion, the moderate increase in CENP-A levels at the centromeric region (1.3-fold; Fig 3A,B) is not sufficient to affect the centromeric CENP-C levels. Thus, the non-stoichiometric nature of CENP-A-CENP-C expression levels might promote higher association of CENP-C at the non-centromeric regions with high CENP-A mislocalization. The revised manuscript includes an explanation for this as suggested by the reviewer on Page # 11.

2. The interactome analysis shows that DNAJC9 depletion reduces the association of CENP-A deposition factors in the chromatin fraction: while the text mentions the reduction of Mis18a, figure 6C does not show Mis18a, only Mis18b, Mis18BP1 and HJURP are shown. It is important to clarify if Mis18a was seen and if not, the authors need to comment on this observation.

We thank the reviewer for pointing this out and apologize for the error. We observed a significant loss of MIS18B and MIS18BP1 with chromatin bound CENP-A as shown in Fig 5C (revised manuscript). However, in this cellular fraction, we did not have sufficient observations to calculate any fold change

data for MIS18A as it was detected in only siCTRL condition in 3 out of 5 replicates analyzed. We have now corrected MIS18A to MIS18B in the text and commented on this observation (Page #14).

3. Experiments with DNAJC9 show that only the DNAJC9 J mutant (lacking enzymatic activity) increases CENP-A levels. Either the H3/H4 binding deficient mutant or the J mutant in combination with the H3/H4 binding defective mutant does not affect CENP -A levels. Possible explanations for these observations (why H3/H4 binding defective or the catalytically inactive J mutant, while in combination with H3/H4 binding defective mutant does not affect the CENP-A levels) are not adequately provided and need to be included in the revised manuscript. Details of the mutations (residue numbers) are not provided anywhere in the main text. This is crucial information and needs to be included in the revised manuscript. We agree with the reviewer that it is important to explain our observation with each of the DNAJC9 mutants. Importantly, these data demonstrate that the DNAJC9 catalytic J mutant only has a dominant negative effect when DNAJC9 binds and traps the histone cargo in an unproductive intermediate. The experiment is conducted in the presence of endogenous DNAJC9; thus, the histone-binding mutants are not expected to have an effect. We have clarified this in the revised manuscript (Page#17) and the details of the mutated residues are included in the Materials and Methods section (Page # 24).

REFERENCES

- Athwal RK, Walkiewicz MP, Baek S, Fu S, Bui M, Camps J, Ried T, Sung MH, Dalal Y (2015) CENP-A nucleosomes localize to transcription factor hotspots and subtelomeric sites in human cancer cells. *Epigenetics Chromatin* 8: 2
- Black BE, Foltz DR, Chakravarthy S, Luger K, Woods VL, Jr., Cleveland DW (2004) Structural determinants for generating centromeric chromatin. *Nature* 430: 578-582
- Bobkov GOM, Gilbert N, Heun P (2018) Centromere transcription allows CENP-A to transit from chromatin association to stable incorporation. *J Cell Biol* 217: 1957-1972
- Ciftci-Yilmaz S, Au WC, Mishra PK, Eisenstatt JR, Chang J, Dawson AR, Zhu I, Rahman M, Bilke S, Costanzo M *et al* (2018) A Genome-Wide Screen Reveals a Role for the HIR Histone Chaperone Complex in Preventing Mislocalization of Budding Yeast CENP-A. *Genetics* 210: 203-218
- Fachinetti D, Folco HD, Nechemia-Arbely Y, Valente LP, Nguyen K, Wong AJ, Zhu Q, Holland AJ, Desai A, Jansen LE, Cleveland DW (2013) A two-step mechanism for epigenetic specification of centromere identity and function. *Nat Cell Biol* 15: 1056-1066
- Filipescu D, Naughtin M, Podsypanina K, Lejour V, Wilson L, Gurard-Levin ZA, Orsi GA, Simeonova I, Toufektchan E, Attardi LD *et al* (2017) Essential role for centromeric factors following p53 loss and oncogenic transformation. *Genes Dev* 31: 463-480
- Foltz DR, Jansen LE, Bailey AO, Yates JR, 3rd, Bassett EA, Wood S, Black BE, Cleveland DW (2009) Centromere-specific assembly of CENP-a nucleosomes is mediated by HJURP. *Cell* 137: 472-484
- Hammond CM, Bao H, Hendriks IA, Carraro M, Garcia-Nieto A, Liu Y, Reveron-Gomez N, Spanos C, Chen L, Rappsilber J *et al* (2021) DNAJC9 integrates heat shock molecular chaperones into the histone chaperone network. *Mol Cell* 81: 2533-2548 e2539
- Huang H, Stromme CB, Saredi G, Hodl M, Strandsby A, Gonzalez-Aguilera C, Chen S, Groth A, Patel DJ (2015) A unique binding mode enables MCM2 to chaperone histones H3-H4 at replication forks. *Nat Struct Mol Biol* 22: 618-626
- Kops GJ, Foltz DR, Cleveland DW (2004) Lethality to human cancer cells through massive chromosome loss by inhibition of the mitotic checkpoint. *Proc Natl Acad Sci U S A* 101: 8699-8704
- Lacoste N, Woolfe A, Tachiwana H, Garea AV, Barth T, Cantaloube S, Kurumizaka H, Imhof A, Almouzni G (2014) Mislocalization of the centromeric histone variant CenH3/CENP-A in human cells depends on the chaperone DAXX. *Mol Cell* 53: 631-644

Nechemia-Arbely Y, Fachinetti D, Miga KH, Sekulic N, Soni GV, Kim DH, Wong AK, Lee AY, Nguyen K, Dekker C *et al* (2017) Human centromeric CENP-A chromatin is a homotypic, octameric nucleosome at all cell cycle points. *J Cell Biol* 216: 607-621

Nye J, Sturgill D, Athwal R, Dalal Y (2018) HJURP antagonizes CENP-A mislocalization driven by the H3.3 chaperones HIRA and DAXX. *PLoS One* 13: e0205948

Piette BL, Alerasool N, Lin ZY, Lacoste J, Lam MHY, Qian WW, Tran S, Larsen B, Campos E, Peng J *et al* (2021) Comprehensive interactome profiling of the human Hsp70 network highlights functional differentiation of J domains. *Mol Cell* 81: 2549-2565 e2548

Shrestha RL, Ahn GS, Staples MI, Sathyan KM, Karpova TS, Foltz DR, Basrai MA (2017) Mislocalization of centromeric histone H3 variant CENP-A contributes to chromosomal instability (CIN) in human cells. *Oncotarget* 8: 46781-46800

Shrestha RL, Balachandra V, Kim JH, Rossi A, Vadlamani P, Sethi SC, Ozbun L, Lin S, Cheng KC, Chari R *et al* (2023) The histone H3/H4 chaperone CHAF1B prevents the mislocalization of CENP-A for chromosomal stability. *J Cell Sci* 136

Shrestha RL, Rossi A, Wangsa D, Hogan AK, Zaldana KS, Suva E, Chung YJ, Sanders CL, Difilippantonio S, Karpova TS *et al* (2021) CENP-A overexpression promotes aneuploidy with karyotypic heterogeneity. *J Cell Biol* 220

Swartz SZ, McKay LS, Su KC, Bury L, Padeganeh A, Maddox PS, Knouse KA, Cheeseman IM (2019) Quiescent Cells Actively Replenish CENP-A Nucleosomes to Maintain Centromere Identity and Proliferative Potential. *Dev Cell* 51: 35-48 e37

Dr. Munira A. Basrai
NCI, NIH
Genetics Branch
41 Medlars Drive
Room B624
Bethesda, MD 20892

1st Mar 2024

Re: EMBOJ-2023-115742R

DNAJC9 prevents CENP-A mislocalization and chromosomal instability by maintaining the fidelity of histone supply chains

Dear Munira,

Thank you again for submitting your revised manuscript to The EMBO Journal. All three original referees have now looked at it again and were generally satisfied with your revisions. Since they are now supportive of publication, we shall be happy to accept the study after a final round of minor revision to incorporate a few the remaining presentational points of the referees, and to address the following editorial issues:

- Please provide an up-to-date email address for co-author Lu Chen, so that acknowledgement emails from our office can be delivered to them.
- Please rename the DATA AND CODE AVAILABILITY statement into "Data Availability" as specified in our Guide to Authors.
- Please rename the conflict of interest statement into "Disclosure and competing interests statement" as specified in our Guide to Authors.
- As we are switching from a free-text author contribution statement towards a more formal statement based on Contributor Role Taxonomy (CRediT) terms, please remove the present Author Contribution section and instead specify each author's contribution(s) directly in the Author Information page of our submission system during upload of the final manuscript. See <https://casrai.org/credit/> for more information.
- Please place the main and expand figure legends after the reference section.
- Tables S1-S4 should be renamed into "Dataset EV1-EV4" and referenced as such; their legends should be removed from the manuscript file, and instead included as a separate tab in each Excel file (as already done for some).
- Since we can only accommodate up to 5 Expanded View Figures that will be directly included in the HTML version, some of the currently 10 EV Figures have to be turned into "Appendix Figures". They need to be called "Appendix Figure S1/2/3..." and combined in a single "Appendix" PDF headed by a brief Table of Contents. Their legends should be moved from the main text and for each figure be placed directly under it in the Appendix.
- During our routine pre-acceptance checks, our data editors have raised the following queries regarding figures, data, and legends:
 - * Please note that a separate 'Data Information' section is required in the legends of figures 3b, d; EV 4b, d, g.
 - * Please indicate the statistical test used for data analysis in the legends of figures 5b-c; EV 5b, d-e; EV 6b, d-e.
 - * Please note that the white arrowheads are not defined in the legend of figure 3f-g. This needs to be rectified.
- Finally, please provide suggestions for a short 'blurb' text prefacing and summing up the conceptual aspect of the study in two sentences (max. 250 characters), followed by 3-5 one-sentence 'bullet points' with brief factual statements of key results of the paper; they will form the basis of an editor-written 'Synopsis' accompanying the online version of the article. Please also upload a synopsis image, which can be used as a "visual title" for the synopsis section of your paper. The image (maybe a compacted/simplified version of Figure 9?) should be in PNG or JPG format, and please make sure that it remains in the modest dimensions of (exactly) 550 pixels wide and 300-600 pixels high.

I am therefore returning the manuscript to you for a final round of minor revision, to allow you to make these adjustments and upload all modified files. Once we will have received them, we should be ready to swiftly proceed with formal acceptance and production of the manuscript.

With kind regards,

Hartmut

*** PLEASE NOTE: All revised manuscripts are subject to initial checks for completeness and adherence to our formatting guidelines. Revisions may be returned to the authors and delayed in their editorial re-evaluation if they fail to comply to the following requirements (see also our Guide to Authors for further information):

9) Digital image enhancement is acceptable practice, as long as it accurately represents the original data and conforms to community standards. If a figure has been subjected to significant electronic manipulation, this must be clearly noted in the figure legend and/or the 'Materials and Methods' section. The editors reserve the right to request original versions of figures and the original images that were used to assemble the figure. Finally, we generally encourage uploading of numerical as well as gel/blot image source data; for details see: embopress.org/page/journal/14602075/authorguide#sourcedata

At EMBO Press, we ask authors to provide source data for the main manuscript figures. Our source data coordinator will contact you to discuss which figure panels we would need source data for and will also provide you with helpful tips on how to upload and organize the files.

In the interest of ensuring the conceptual advance provided by the work, we recommend submitting a revision within 3 months (30th May 2024). Please discuss the revision progress ahead of this time with the editor if you require more time to complete the

revisions. Use the link below to submit your revision:

Link Not Available

Referee #1:

We thank the authors for their clarity and their detailed answers. Again, we wish to highlight how this study can contribute to the field with two major points:

- (1) The identification of factors potentially involved in CENP-A mislocalization and CIN: an important question with impact for prognosis and treatment of CENP-A overexpressing cancers.
- (2) The illustration of how multiple pathways underly CENP-A mislocalization, confirming that the balance in the proportions of soluble histone variants is key for the proper localization of CENP-A in human cells. Indeed, they reinforce previous studies with the fact that overexpression of CENP-A and/or a change in the levels of H3-H4 dimers available leads to CENP-A mislocalization (and subsequent CIN phenotypes).

Overall, the authors made a significant effort to address our concerns to better support their conclusions. They added controls and quantifications as asked and performed new experiments (as asked by the reviewers) that strengthen their conclusions. The additional data confirm that DNAJC9 depletion promotes CENP-A mislocalization, and that CENP-A is stably bound to the chromatin. Their new data also confirm that the levels of CENP-A mislocalization and CIN due to DNAJC9 depletion depend on the levels of CENP-A itself, as expected.

The manuscript can thus be finalized following minor revisions to correct some remaining inaccuracies:

- "Cancer cells over-expressing CENP-A have a proliferative advantage when treated with ionizing radiation and DNA-damaging agents (Jeffery et al., 2021; Lacoste et al., 2014)." This sentence should be rephrased. In Lacoste et al., cells overexpressing CENP-A can indeed better cope with Camptothecin. However, in Jeffery et al., CENP-A overexpression could give a proliferative advantage in a certain context, when p53 is defective, while it would rather promote senescence and radiosensitivity when p53 is functional.
- Fig.1D: The authors added a scale bar but did not indicate the magnification of the images. The authors should clarify whether it is the same between the two conditions or provide the parameters used for each image displayed (e.g., range of pixel values displayed...)
- New figure 8 added: For fig. 8A, ideally, the authors could indicate the number of genes plotted. The legend of the figure should mention how the counts were normalized.

Referee #2:

Comments to the revised manuscript previously entitled "The histone H3-H4 co-chaperone DNAJC9 prevents CENP-A mislocalization and chromosomal instability" now entitled "DNAJC9 prevents CENP-A mislocalization and chromosomal instability by maintaining the fidelity of histone supply chains "by Balachandra et al.

The authors have significantly improved their data and conclusions of their manuscript. Also, the new title is a better fit to their study. Even though they have not answered all my questions, the manuscript is solid and interesting and I support publication. Below I will only comment on the points raised by me (Referee #2) initially.

1. Is CENP-A genuinely incorporated into chromatin? The authors used high salt extraction on metaphase spreads (instead of more commonly used Western blot) following a protocol from a paper that has been under scrutiny in the field. I appreciated their efforts but would have preferred to see a simple westernblot of a technique re-produced by many labs that can be more accurately quantified. Alternatively, the author should show that their technique is removing proteins that are supposed to be removed from mitotic chromosomes in 0,5 M NaCl as a positive control. However, in combination with their CUT&RUN analysis, I assume this will be sufficient (even though also indirect).
2. The authors avoided my question but fair enough, if everybody uses the construct they should be allowed to (but none of the cited papers ever showed that the tagged construct is rescuing a CENP-A depleted background)
3. Would be nice to included in the data as a control.
4. The Pdf version of Table 1 is not opening for me, the Excel works and is more informative now.
5. Thank you for the analysis.
6. Fine, thank you.
7. Fine, thank you.
8. Also here, a high salt extracted Western blot would have solved the problem. The H3.3 experiments do not answer my question.
9. Ok
10. I am still not fully convinced that your data allow the conclusions you draw but at least it can be one option.

11. OK!
12. OK
13. OK
14. OK
15. OK
16. As said above, the manuscript improved significantly and seems more suitable for publication now.

Referee #3:

The authors have revised the manuscript with a significant amount of additional data and several textual changes in response to reviewers' concerns. These additions/changes significantly improve the overall quality of the manuscript.

All editorial and formatting issues were resolved by the authors.

Dr. Munira A. Basrai
NCI, NIH
Genetics Branch
41 Medlars Drive
Room B624
Bethesda, MD 20892

25th Mar 2024

Re: EMBOJ-2023-115742R1

Histone chaperone DNAJC9 prevents CENP-A mislocalization and chromosomal instability by maintaining the fidelity of histone supply chains

Dear Munira,

Thank you for submitting your final revised manuscript for our consideration. I am pleased to inform you that we have now accepted it for publication in The EMBO Journal, after having gone through all the files in the final submission.

Please note that I changed two final things:

- Added the words "histone chaperone" to the title, to make it more explicit, even if slightly overlength
- Removed the co-second author tags - while we do allow co-corresponding authors and up to 4 co-first authors, our policies do not envision co-contribution of certain middle authors to be additionally specified. Such a distinction becomes too granular, and the specific contribution is much better spelt out using the Contributor Role Taxonomy (CRediT) terms, together with their free-text boxes.

You may qualify for financial assistance for your publication charges - either via a Springer Nature fully open access agreement or an EMBO initiative. Check your eligibility: <https://www.embojournal.org/page/journal/14602075/authorguide#chargesguide>

With kind regards,

Hartmut
